Resource

# A multi-kingdom collection of 33,804 reference genomes for the human vaginal microbiome

Liansha Huang[1,12], Ruochun Guo [2,12], Shenghui Li [2,3,12] ✉, Xiaoling Wu[1], Yue Zhang [2], Shumin Guo[1], Ying Lv[1], Zhen Xiao[4], Jian Kang[3], Jinxin Meng[2], Peng Zhou[5], Jie Ma[6], Wei You[7], Yan Zhang[8], Hailong Yu[2], Jixin Zhao[2], Guangrong Huang[9], Zuzhen Duan[9], Qiulong Yan [3] ✉ & Wen Sun [10,11] ✉

The human vagina harbours diverse microorganisms—bacteria, viruses and fungi—with profound implications for women's health. Genome-level analysis of the vaginal microbiome across multiple kingdoms remains limited. Here we utilize metagenomic sequencing data and fungal cultivation to establish the Vaginal Microbial Genome Collection (VMGC), comprising 33,804 microbial genomes spanning 786 prokaryotic species, 11 fungal species and 4,263 viral operational taxonomic units. Notably, over 25% of prokaryotic species and 85% of viral operational taxonomic units remain uncultured. This collection significantly enriches genomic diversity, especially for prevalent vaginal pathogens such as BVAB1 (an uncultured bacterial vaginosis-associated bacterium) and *Amygdalobacter* spp. (BVAB2 and related species). Leveraging VMGC, we characterize functional traits of prokaryotes, notably Saccharofermentanales (an underexplored yet prevalent order), along with prokaryotic and eukaryotic viruses, offering insights into their niche adaptation and potential roles in the vagina. VMGC serves as a valuable resource for studying vaginal microbiota and its impact on vaginal health.

The human vagina is a diverse ecosystem hosting bacteria, viruses, fungi and other micro-eukaryotes which are crucial for maintaining women's and fetal health[1,2]. In healthy women, *Lactobacillus* species dominate the vaginal microbiota, producing lactic acid to sustain an acidic environment and prevent the proliferation of harmful microorganisms[3,4]. Conversely, dysbiosis of the vaginal microbiota often leads to various health issues, including bacterial vaginosis characterized by the overgrowth of anaerobic bacteria such as *Bifidobacterium vaginalis* (formerly *Gardnerella vaginalis*), *Fannyhessea vaginae* and *Prevotella* species[5,6].

Early research on the vaginal microbiota predominantly relied on culture-based or real-time PCR methods, posing inherent limitations in capturing the complete microbial diversity. The advent of next-generation sequencing, particularly amplicon sequencing targeting 16S ribosomal RNA gene or internal transcribed spacer (ITS) regions, has significantly advanced our understanding, uncovering uncultivated microorganisms associated with vaginal health, such as bacterial vaginosis-associated bacterium (BVAB) 1 to 3[7]. However, while amplicon sequencing reveals taxonomic information, it falls short in exploring the functional characteristics and genome variations (for example, for lactobacilli[8]) of the microbial community. Studies using whole metagenomic sequencing have elucidated the compositional and functional diversity of the vagina microbiome[9–11]. Utilizing this technology, the Human Microbiome Project (HMP) has provided an overview of microbial and functional profiles across various body sites, including the vagina[12]. The study by Jie et al.[11] has uncovered

**Fig. 1 | The construction and quality assessment of the VMGC. a**, The construction flowchart of VMGC. The leftmost panels represent the sources of the genomes included in the VMGC. The middle image shows the number of prokaryotic, fungal and viral genomes. The rightmost pie chart indicates the proportions of prokaryotic, fungal and viral genomes within the VMGC. **b**, The CheckM2-estimated completeness and contamination of 19,542 prokaryotic genomes. The genome quality classification refers to the MAGs; we referred to the revised MIMAG standard. **c,d**, The distribution of the N50 length (**c**) and genome size (**d**) of 19,542 prokaryotic genomes, including 1,017 near-complete, 8,397 high-quality and 10,127 medium-quality genomes. In the boxplot, the centre line represents the median, the box limits show the upper and lower

quartiles, the whiskers extend to 1.5 times the IQR, and points outside the whiskers are considered outliers. **e**, The BUSCO-estimated completeness and contamination of 38 fungal (blue dots) and 4 parasitic (orange dots) genomes. **f,g**, The CheckV-estimated completeness (**f**) and contamination (**g**) of 14,224 viral genomes. Genomes with >50% completeness are categorized as medium-quality, >90% completeness as high-quality and 100% completeness as complete. Genomes with <10% contamination are considered low-contamination. **h**, The distribution of the genome size for viral genomes. In the boxplot, the centre line represents the median, the box limits show the upper and lower quartiles, the whiskers extend to 1.5 times the IQR and points outside the whiskers are considered outliers.

associations between the vaginal microbiota and life history, while the work of Fettweis et al. linked the vaginal microbiota to preterm birth[9]. A pivotal contribution to the reference database comes from the human vaginal non-redundant gene catalogue (VIRGO)[10], which includes a total of 0.95 million genes compiled from metagenomic and bacterial genomic data. Although these findings confirm the presence of a diverse microbial community in the vagina, our knowledge of the full spectrum of microorganisms remains limited, primarily due to the lack of vaginal reference genomes.

In recent years, significant progress has been made in applying the metagenome-assembled genome (MAG) binning methodology to obtain a vast amount of genomic information from both cultivated and uncultivated microorganisms. This approach has shown efficacy in exploring microbiomes across various human body sites, including the gut[13], oral cavity[14] and skin[15]. In particular, Pasolli et al.[16] have performed a comprehensive collection of over 150,000 MAGs representing the human microbiota. However, this collection encompasses merely 151 genomes from the vagina[16]. Consequently, there exists an urgent need to undertake metagenomic-based efforts aimed at establishing

an extended reference genome database specifically tailored for the vaginal microbiota.

Herein, we have taken a comprehensive approach by integrating available metagenomic datasets and microbial genomes and engaging in in-house cultivation of vaginal fungi to construct the Vaginal Microbial Genome Collection (VMGC). The VMGC comprises 19,542 prokaryotic genomes, 38 fungal genomes and 14,224 viral genomes, spanning across 786 prokaryotic species, 11 fungal species and 4,263 species-level viral operational taxonomic units (vOTUs). Based on the VMGC, our analyses have revealed unexplored microbial species, genomic variations and functional characteristics within the human vaginal microbiome.

## Results

### Construction of the VMGC

The flowchart of the construction of VMGC is shown in Extended Data Fig. 1a. We collected a total of 4,472 publicly available vaginal metagenomic samples sourced from the human vagina, spanning 32 studies across the United States (n = 2,741 samples), France (749 samples),

China (581 samples) and 11 other transcontinental countries (Supplementary Tables 1 and 2). Applying quality control, metagenomic assembly and both single- and multi-coverage metagenomic binning processes in each sample generated a total of 63,654 preliminary MAGs with a minimum length of 200 kbp. Subsequent dereplication and quality screening yielded 18,570 prokaryotic genomes with ≥50% completeness and <5% contamination (Supplementary Table 3) (see Methods for quality criteria). In addition, a search for eukaryotes from the preliminary MAGs identified 17 eukaryotic genomes with a minimum genome size of 4.46 Mbp, including 13 fungal genomes, 2 genomes of *Trichomonas vaginalis* and 2 genomes of *Theileria parva* (Supplementary Table 4).

We complemented our collection with publicly available prokaryotic and fungal genomes sourced from the National Center of Biotechnology Information (NCBI) RefSeq genome database and obtained 1,189 prokaryotic and 18 fungi genomes that were previously isolated from the female vagina. After quality filtering, we retained 972 high-quality prokaryotic and 17 fungal genomes for further analysis (Fig. 1a and Supplementary Tables 3 and 4). In addition, we performed fungal cultivation from the vaginal swabs of healthy women and obtained 8 cultured fungi included in our genome collection.

Among 19,542 prokaryotic genomes (that is, 18,570 MAGs and 972 isolated genomes), 10,127, 8,397 and 1,017 were classified as medium-quality, high-quality and near-complete genomes, respectively (Fig. 1b), following the criteria revised from the MIMAG (Minimum Information about a Metagenome-Assembled Genome) standard[17] (Methods). The median N50 length and genome size of all prokaryotic genomes were 38.0 kbp (interquartile range (IQR) = 14.5–81.4 kbp) and 1.36 Mbp (IQR = 1.11–1.62 Mbp), respectively, with a downward trend from near-complete to medium-quality genomes (Fig. 1c,d). Of the 38 fungal genomes (that is, 13 MAGs and 25 isolated genomes), the medium completeness and contamination were 92.7% and 0.4%, respectively, and the medium genome size was 12.7 Mbp (IQR = 11.9–14.6 Mbp) (Fig. 1e).

Finally, to further expand the horizon of the viral community, we performed virus identification on the metagenome-assembled contigs using a combined feature- and homology-based approach, resulting in 14,224 viral sequences achieving at least 50% completeness (Fig. 1a and Supplementary Table 5). Based on the CheckV algorithm[18], 10.3% of viral sequences were estimated as complete viral genomes, 29.3% were of high quality (completeness ≥90%) and 60.4% were of medium quality (Fig. 1f). In addition, 42.3% of viral sequences were recognized as integrated proviruses by CheckV, and the viral regions were extracted. Most viral genomes (98.9%) showed a low degree of genomic contamination (<10%) (Fig. 1g), and analysis of the host-to-virus gene ratio also revealed low microbial host contamination (Extended Data Fig. 1b). The genome size of viral sequences ranged from 5.0 kbp to 417.5 kbp, with a median length of 33.5 kbp (IQR = 24.0–41.1 kbp). High-quality viruses had a larger genome size than medium-quality viruses; however, the complete viruses seemed to have the smallest genomes (Fig. 1h).

## Taxonomic landscape of vaginal prokaryotic species

We conducted a genome-wide clustering analysis on 19,542 prokaryotic genomes, resulting in 786 species-level genome bins (SGBs, referred to

as 'species' thereafter) based on the 95% nucleotide similarity threshold[19]. Rarefaction analysis revealed that species richness did not reach saturation (Extended Data Fig. 2a), suggesting the existence of additional undiscovered species. However, these species are likely to be predominantly rare, as the number of species approaches a plateau when considering only those with at least two conspecific genomes. About 41.8% (*n* = 329) of the 786 species had one or more genomes previously isolated from the human vagina, while the remaining 58.1% (*n* = 457) species had only MAGs available (Fig. 2a,b). We compared the prokaryotic species to NCBI RefSeq isolated genomes to extend the search to species from other environments. This analysis identified an additional 249 species (31.7% of all prokaryotic species) showing high homology to the genomes previously cultured from other habitats such as the human gastrointestinal tract, oral cavity or natural environments, leaving 208 species (26.5%) currently uncultured (Fig. 2b and Supplementary Table 6).

Taxonomic analysis of vaginal prokaryotic species revealed the presence of 15 phyla, 18 classes, 43 orders, 87 families and 239 genera. Dominant phyla included Actinomycetota (consisting of 28.5% of all species), Bacillota_A (21.2%), Bacteroidota (16.3%) and Bacillota (16.0%), followed by Pseudomonadota (6.5%) and Bacillota_C (4.5%) (Fig. 2a and Extended Data Fig. 2b). At the lower taxonomic levels, some taxa such as Bacteroidales (15.0% of all species; mainly consisting of *Prevotella* spp.), Actinomycetales (15.0%; mainly *Bifidobacterium* and Actinomycetaceae spp.), Lactobacillales (12.6%; mainly *Lactobacillus* and *Streptococcus* spp.), Tissierellales (9.2%; mainly *Anaerococcus* and *Peptoniphilus* spp.), Coriobacteriales (6.4%; mainly *Fannyhessea* spp.) and Mycobacteriales (6.1%; mainly *Corynebacterium* spp.) were of the major clades. Unlike the human gut microbiota which consisted of many species belonging to the genus *Bacteroides* and the order Lachnospirales (for example, *Clostridium*, *Ruminococcus* and *Eubacterium* spp.)[13], the vaginal microbiota had rare representatives of these, with only 0.6% (*n* = 5) of species being Bacteroides and 5.3% (*n* = 42) being Lachnospirales.

The uncultured species were widely distributed across different phyla, especially *Patescibacteria*, *Campylobacterota*, *Spirochaetota* and *Bacillota_A* (Fig. 2a). We further calculated the phylogenetic diversity based on phylogenetic trees for nine phyla with more than three uncultured species. This analysis showed that the uncultured species accounted for, on average, 52.8% (ranging from 34.3% to 83.0%) of phylogenetic diversity in the trees for these phyla, with the highest proportion of phylogenetic diversity in *Patescibacteria* and *Campylobacterota* (Fig. 2c). In particular, despite that there are an average of 25.9% of species in 4 dominated vaginal phyla being uncultured, they contributed an average of 51.8% of phylogenetic diversity.

Some uncultured species, including SGB010 (known as BVAB1, a typical uncultured bacterial vaginosis-associated bacterium[7], consisting of 461 genomes), SGB013 (391 genomes) and SGB015 (374 genomes), contained a large number of genomes. Importantly, these genomes were assembled from vaginal metagenomes from various countries (Extended Data Fig. 2c), suggesting their widespread presence in the human population. Furthermore, we profiled the relative abundances of all VMGC prokaryotic species in metagenomic samples. Consistent with previous studies[10,20], species that belonged to *Lactobacillus* (for example, *Lactobacillus iners*, *Lactobacillus crispatus* and *Lactobacillus*

**Fig. 2 | The 786 prokaryotic species in the VMGC. a**, Taxonomic classification of prokaryotic species. The innermost sunburst plot shows the taxonomic hierarchy of the species, with the size of the sector representing the number of species assigned to the corresponding taxon. The middle ring indicates whether the species are cultured or uncultured, along with their respective isolation sources (see Supplementary Table 6 for more details). A species is considered a cultured species if any genome clustered into the species shares >95% ANI with at least one isolate in the NCBI. The outermost bar plot shows the number of genomes clustered into each species. **b**, The number of uncultured species, species cultured from the vagina and species cultured from non-vaginal sites.

**c**, The proportions of uncultured species, species cultured from the vagina and species cultured from non-vaginal sites within different phyla, as well as the proportion of phylogenetic diversity occupied by uncultured species. **d**, The dominated genera across 4,429 vaginal metagenomes. The species in brackets represent the dominant members of each genus. **e**, The weighted abundances of different functional modules for each order in vaginal samples. The upper bar plot shows the average relative abundance of each order across vaginal samples. The weighted abundances in the heat map are standardized as row *Z*-scores. The order with the highest weighted abundance of each functional module is indicated by a dotted box. SCFAs, short-chain fatty acids.

*jensenii*), *Bifidobacterium* (*Bifidobacterium vaginale*), *Prevotella* (*Prevotella bivia* and *Prevotella timonensis*) and *Fannyhessea* (*F. vaginae*) were the main components of the samples. Several unclassified bacteria including BVAB1 (SGB010), *Bifidobacterium* sp. (SGB042 and SGB058) and *Prevotella* sp. (SGB013) were also among the high-abundance species (Fig. 2d and Extended Data Fig. 2c).

## Functional configuration of vaginal prokaryotic species

We annotated the functions of vaginal prokaryotic species. Approximately 67.4% of protein-coding genes could be accommodated in the Kyoto Encyclopedia of Genes and Genomes (KEGG) database[21], mostly in metabolism and genetic information processing genes (Extended Data Fig. 3a). In addition, 5.2% of genes were annotated

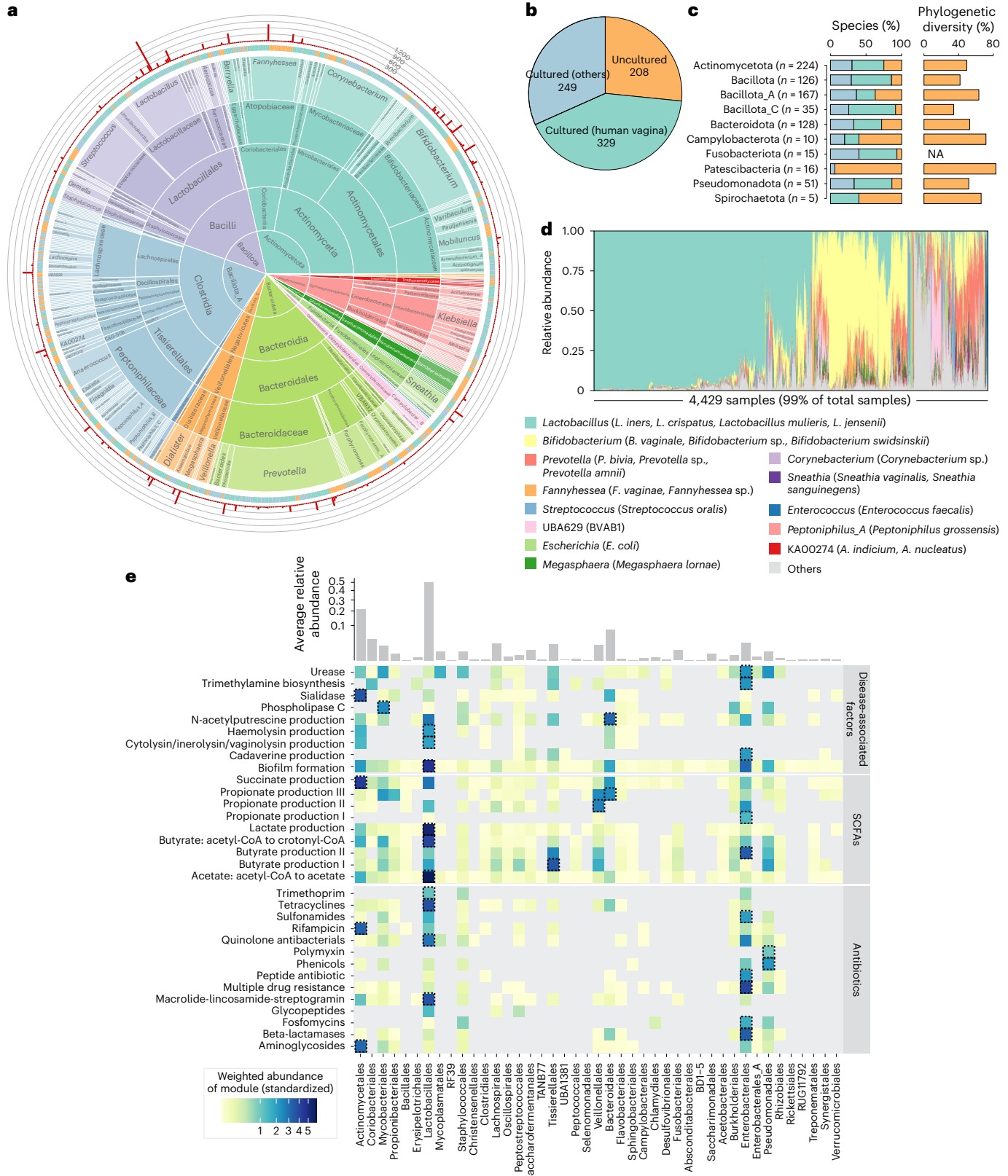

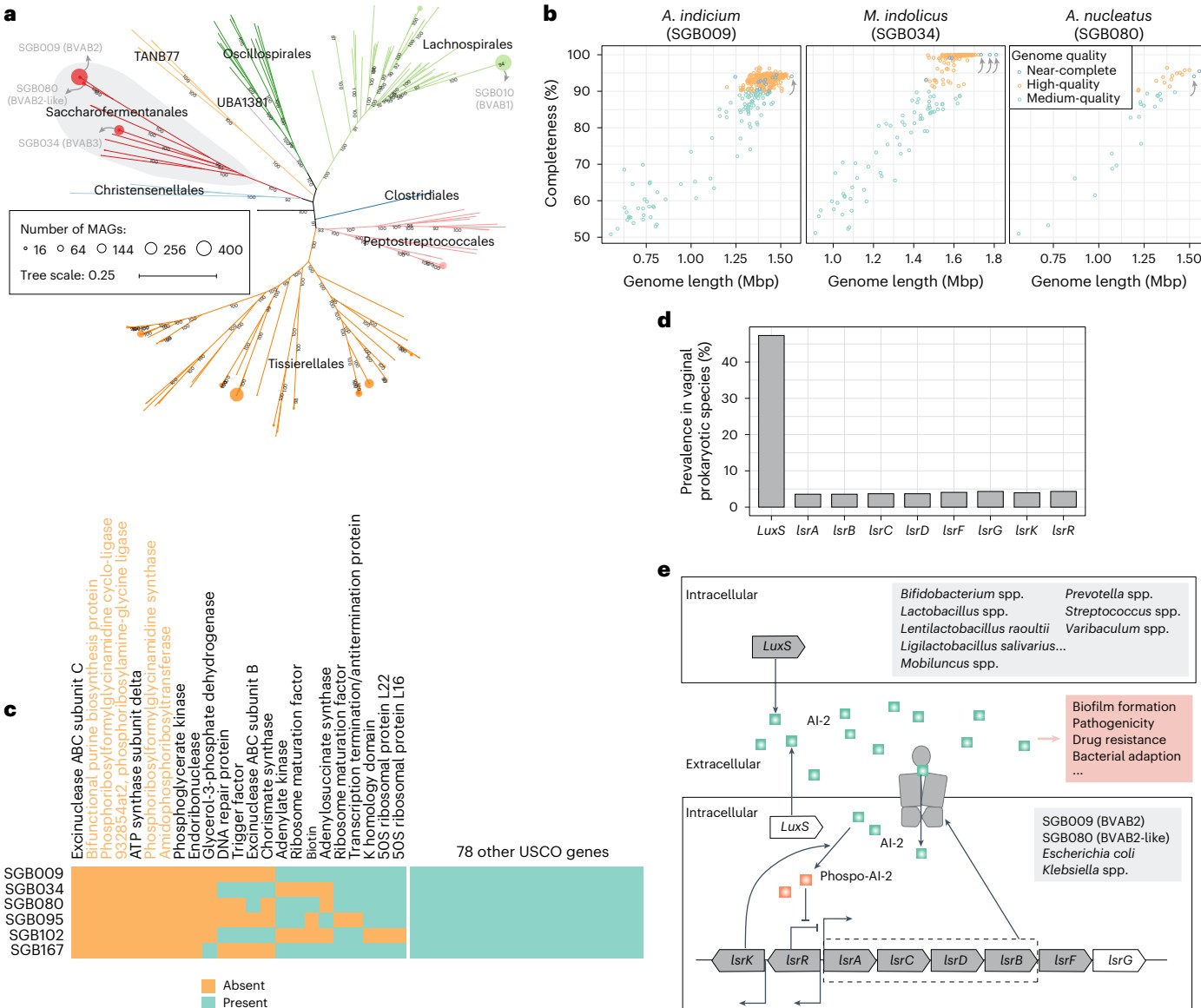

**Fig. 3 | Genomic characteristics of Saccharofermentanales members in the VMGC. a**, A phylogenetic tree for all species from the class Clostridia in the VMGC. The tree branches are coloured to represent different orders. The numbers on the branches represent bootstrap values, and the size of the point on the tip of each branch is positively correlated with the number of genomes clustered into the respective species. **b**, Completeness scores and genome sizes of genomes from three dominant members of Saccharofermentanales based on the CheckM2 algorithm. The arrows represent a single no-gapped genome available in the NCBI RefSeq database. **c**, Presence of USCO genes in Saccharofermentanales members. The five enzymes involved in de novo purine biosynthesis are coloured yellow. **d**, The prevalence of genes related to the Lsr-type autoinducer-2 (AI-2) transport system among all SGBs in the VMGC. **e**, Schematic diagram of the Lsr-type AI-2 transport system present in vaginal bacteria. It is noteworthy that the genes associated with *LuxS* and *lsrG* are absent in SGB009 (BVAB2) and SGB080.

into the Carbohydrate-Active enZYmes (CAZy) database[22], while 1.2% and 0.2% were identified as virulence factors and antibiotic resistance genes, respectively.

Our comprehensive vaginal genome collection may enable a high-resolution functional analysis of the vaginal microbiome. To demonstrate this, we focused on nine functional modules that had been reported to be associated with vaginal diseases or metabolic abnormalities[6,23] (Supplementary Table 7). We found that Lactobacillales (mainly *L. crispatus* and *L. iners*) and Enterobacterales (primarily *Escherichia coli*) significantly contributed to gene abundance in biofilm formation synthesis, followed by several potential pathogens such as *Pseudomonas aeruginosa* and BVAB1 (Fig. 2e and Extended Data Fig. 3b). Genes involved in the biosynthesis of sialidase, an enzyme capable of disrupting the vaginal mucosal barrier[23,24], were predominantly

encoded by Actinomycetales (mainly *Bifidobacterium piotii* and *B. vaginale*) (Extended Data Fig. 3c). Lactobacillales (mainly *L. iners*) and *Bifidobacterium* spp. showed the highest gene abundances related to the synthesis of cytolysins (for example, inerolysin and vaginolysin) and haemolysin (Extended Data Fig. 3d); these toxins pose a high risk for inflammation and the disruption of the vaginal epithelial barrier[25,26]. Mycobacteriales and Enterobacterales showed the highest gene abundance in the synthesis of phospholipase C and urease, while *Ureaplasma parvum* and *Pseudomonas* spp. also encoded a high gene abundance for urease. Several biogenic amines, including cadaverine, N-acetylputrescine and trimethylamine, are major amine components of vaginal fluid and are implicated in vaginitis and the activation of proinflammatory responses[27-29]. For these, Enterobacterales (mainly *E. coli*) and Saccharofermentanales (mainly *Amygdalobacter indicium* and

*Mageeibacillus indolicus*) encoded the major gene abundance responsible to produce cadaverine and trimethylamine, while various beneficial (for example, *Lactobacillus gasseri* and *Lactobacillus johnsonii*) and harmful species (for example, *E. coli* and *P. bivia*) harboured genes associated with N-acetylputrescine production (Extended Data Fig. 3e).

Similarly, we quantified the functions of short-chain fatty acids and antibiotic resistance genes synthetization in vaginal metagenomes. This analysis showed that Lactobacillales encoded the highest gene abundances associated with the biosynthesis of acetate and lactate; Actinomycetales was identified as a potential primary producer of succinate, while Veillonellales and Tissierellales played significant roles in producing propionate and butyrate, respectively (Fig. 2e). Besides, Lactobacillales, Actinomycetales, Pseudomonadales and Enterobacterales showed the highest abundances of genes for antibiotic resistance.

### Description of Saccharofermentanales in the human vagina

Saccharofermentanales, a prevalent vaginal order seldom found in other parts of the human body, contained several bacterial vaginosis-associated species, including SGB009 (*A. indicium*, previously known as BVAB2), SGB034 (*M. indolicus*, known as BVAB3) and SGB080 (*Amygdalobacter nucleatus*, a BVAB2-like species), in VMGC (Fig. 3a). *M. indolicus* has been previously isolated and sequenced in a limited number (*n* = 3) of available genomes, while the genomes of *A. indicium* and *A. nucleatus* were first sequenced during the preparation of this study (each with one complete genome in NCBI)[30]. By contrast, our MAGs, specifically SGB009, SGB034 and SGB080, have yielded 12, 3 and 1 near-complete genomes, along with a substantial number of high-quality genomes (360, 98 and 13, respectively). Interestingly, we observed that the completeness of *A. indicium* and *A. nucleatus* might be underestimated in the current CheckM2 algorithm[31], as their complete genomes remain at 94.1% estimated completeness, respectively (Fig. 3b). Similar assessment results were also revealed by other algorithms such as CheckM and BUSCO (based on 124 universal single-copy orthologs (USCOs) for prokaryotes[32]) (Extended Data Fig. 4a). To elucidate this, we analysed the 124 USCOs for each species and found that they individually lack 14 USCO genes (Fig. 3c and Extended Data Fig. 4b), indicating that their genomes may have lost certain key genes during evolution, leading to a decrease in their 'completeness'. This phenomenon was also observed in SGB034 and other Saccharofermentanales species, with nine USCO genes seemingly absent in all Saccharofermentanales species, likely accompanying marked genome reduction in the human vagina (Extended Data Fig. 4c). In addition, five of the nine missing USCO genes were involved in de novo purine biosynthesis (Fig. 3c); the lack of this pathway has yet to be reported in other reproductive tract pathogens, such as *Treponema pallidum*, *Chlamydia trachomatis* and *Mycoplasma pneumoniae*[33].

Next, we investigated the functional characterization of three prevalent Saccharofermentanales species. Nearly half (49.5%) of KEGG functional orthologs overlapped among three species, with some of these being rare in other vaginal species (Extended Data Fig. 5a). Among these, hexokinase (K00844) was only detected in Saccharofermentanales species and a few other taxa (six Veillonellales, three Bacteroidales and two Selenomonadales species), suggesting a high hexose (for example, glucose, mannose and fructose) utilization capacity for them. Conversely, some functions varied significantly between different Saccharofermentanales species. For example, SGB034 harboured more KEGG functional orthologs involved in the metabolism of thiamine, glycerophospholipid and glycerolipid, while the capacity for cadaverine production was only found in SGB009 and SGB080 (Extended Data Fig. 5b). Specifically, we found that the species SGB009 and SGB080 encoded Lsr-type autoinducer-2 (AI-2) receptors, which are virtually absent in other vaginal bacteria (Fig. 3d). As an important signalling molecule for interspecies communication, AI-2 plays a crucial role in regulating multiple bacterial behaviours (for example, survival, biofilm formation and virulence-related gene expression)

via AI-2 receptors[34,35]. In the vaginal microbiome, AI-2 was catalytically synthesized by the *LuxS* gene, which is encoded by numerous bacteria including *Bifidobacterium*, *Prevotella*, *Lactobacillus* and *Streptococcus* species (Fig. 3d,e). Research indicated that bacteria possessing AI-2 delivery and internalization pathways can not only regulate the formation of their biofilms through AI-2 but also internalize exogenous AI-2 to disrupt the biofilm formation of other competing bacteria, even if they lack the *LuxS* gene[36–38]. This suggests that the species SGB009 and SGB080 can interfere with quorum sensing and the growth of other vaginal bacteria by internalizing AI-2, potentially giving them an advantage in competition.

### Characteristics of vaginal fungal species

We clustered the 38 fungal genomes in VMGC into 11 species with a 95% genome similarity threshold (Supplementary Table 4). Taxonomic classification revealed nine known fungal species, whereas the other two species (all derived from vaginal fungal cultivation in this study) were not represented in the NCBI RefSeq database (Extended Data Fig. 6a). Two species, *Clavispora lusitaniae* and *Pichia kudriavzevii*, were represented only by MAGs without isolated genomes, and the presence of these fungi in the woman's vagina had been previously reported[39]. In addition, we calculated the mapping rates and prevalence rates of the 11 fungal species in large-scale vaginal metagenome datasets. This analysis revealed that *Candida albicans*, *Nakaseomyces glabratus* and *Saccharomyces cerevisiae* were the most dominant fungi in the human vagina, followed by *Candida parapsilosis*, *P. kudriavzevii* and *C. lusitaniae* (Extended Data Fig. 6b). Besides, *C. albicans*, *Aspergillus chevalieri*, *C. parapsilosis* and *S. cerevisiae* were the most prevalent fungal species, each with prevalent rates exceeding 10% in the vaginal mycobiome.

### Taxonomic and functional diversity of the vaginal viruses

To explore the taxonomic content of the vaginal virome, we deduplicated the viral genomes of VMGC into 4,263 species-level vOTUs at 95% nucleotide identity (Supplementary Table 8). Rarefaction analysis showed that the accumulation curve of vOTUs was not reaching a plateau at the current number of viral genomes, despite the non-singleton vOTUs (consisting of 35.5% of all vOTUs) reaching saturation (Fig. 4a). In addition, we conducted a clustering analysis on VMGC vOTUs and all viral species from five large-scale virome databases including Gut Virome Database[40], Gut Phage Database[41], Metagenomic Gut Virus catalogue[42], Oral Virus Database[43] and all NCBI RefSeq viral genomes. The result revealed that 85.8% of vOTUs in the VMGC were not found in any of the other virome databases (Fig. 4b). These findings indicate a significant unexplored diversity of viral content in the human vagina as well as in VMGC.

Taxonomic classification revealed that 66.0% (2,814 out of 4,263) of the vOTUs could be robustly assigned to known viral families, including 2,744 vOTUs assigned to 13 prokaryotic viral families and 70 vOTUs assigned to 7 eukaryotic viral families (Fig. 4c). The most frequently observed families in VMGC were Siphoviridae (*n* = 2,134 vOTUs) and Myoviridae (*n* = 461), followed by Papillomaviridae (*n* = 61), Rountreeviridae (*n* = 39), Podoviridae (*n* = 36), Quimbyviridae (*n* = 27), Autographiviridae (*n* = 14), p-crAss-like (*n* = 13) and other families. Siphoviridae and Myoviridae were the most dominant viral families in human body sites including the gut and oral cavity[40–43]; however, another viral family that predominated in the human gut, Microviridae, was rarely found in the vagina (*n* = 4 vOTUs in VMGC). In addition, 86.4% of vOTUs in VMGC had at least one predicted prokaryotic host (Fig. 4c). At the phylum level, the predicted hosts of vaginal viruses were dominated by Actinomycetota, Bacteroidota, Bacillota and Bacillota_A.

We constructed a phylogenetic tree of 4,263 vOTUs based on their genome-wide similarity at the protein level (Fig. 4d). The tree revealed that the viruses tend to cluster by both family-level taxonomies and potential host affiliations. This finding largely agreed with that obtained from the oral viral catalogue and suggested that host adaptation was

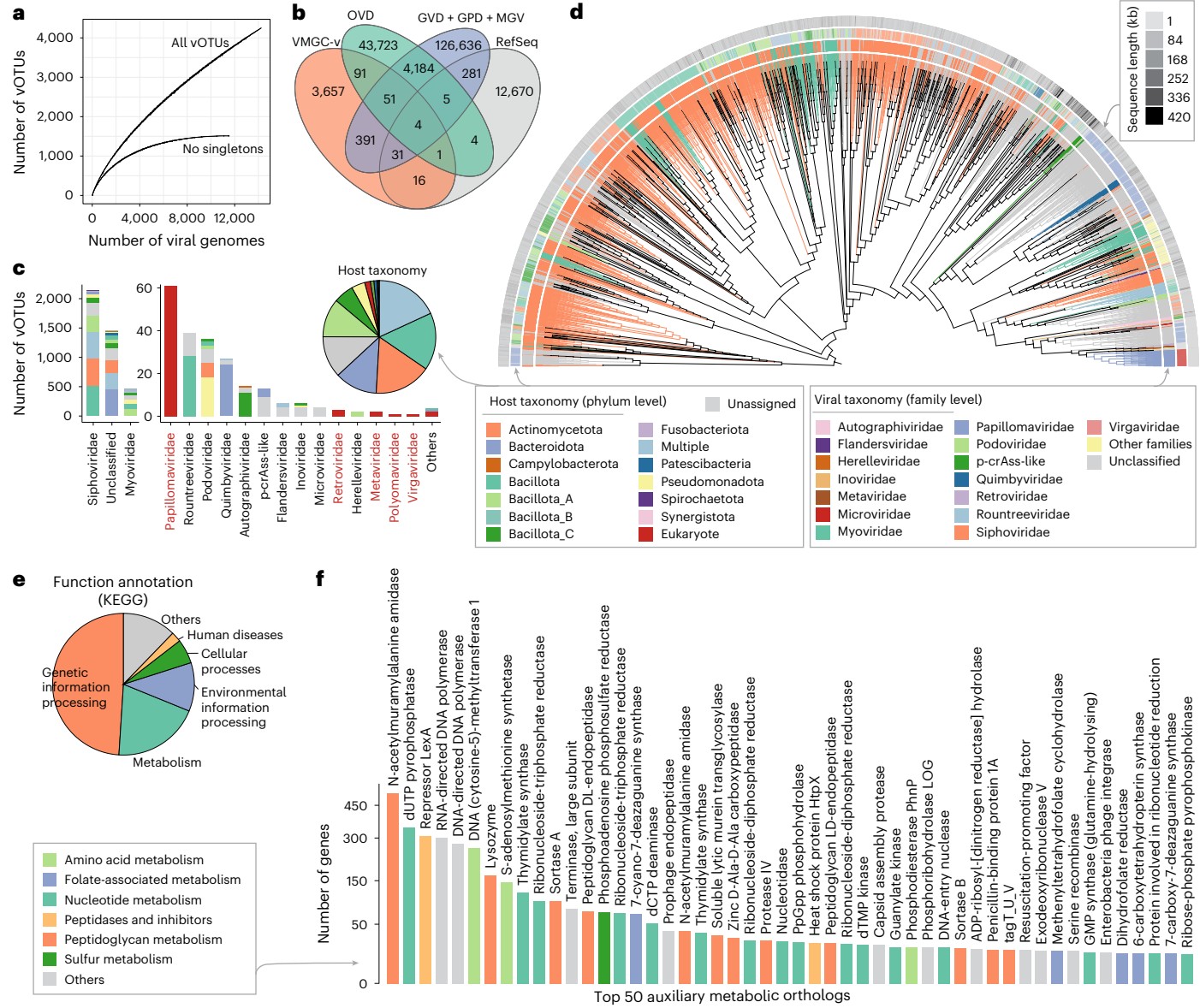

**Fig. 4 | Characteristics of viral populations in the VMGC. a**, The accumulation curve of vOTUs as the number of viral genomes increases. **b**, The overlap of viral species among several large-scale viral genome catalogues. **c**, Distribution of host phyla for 4,263 vOTUs. The bar plot shows the distribution of host phyla for different viral families, with the eukaryotic viral families coloured in red, while the pie plot displays the distribution of host phyla across all vOTUs. **d**, Proteomic tree showing the relationships among 4,263 vOTUs. **e**, The functional distribution of KEGG-annotated genes for 4,263 vOTUs. **f**, The number of genes from the top 50 auxiliary metabolic orthologs for 4,263 vOTUs. An auxiliary metabolic ortholog was defined as a KEGG functional ortholog associated with the KEGG metabolism pathway.

an important factor driving genomic evolution in human-associated viruses[43]. The Siphoviridae vOTUs were predicted to infect Actinomycetota, Bacillota, Bacillota_A and other bacteria (Fig. 4c). Strikingly, Siphoviridae contained numerous viruses (n = 440; corresponding to 20.6% of all Siphoviridae viruses) that were predicted to infect hosts across multiple prokaryotic phyla; however, this phenomenon was rarely observed in viruses of other families (average 2.4% for 12 prokaryotic viral families) (Supplementary Table 8), suggesting a relatively broad host range of vaginal Siphoviridae members. Myoviridae were predicted to infect Bacillota_A, Bacillota, Pseudomonadota and others, but Rountreeviridae and Autographiviridae seemed to be specific to the infection of Bacillota and Bacillota_C, respectively. Notably, the Bacteroidota phages were remarkably concentrated in the phylogenetic tree (Fig. 4d). Specifically, out of 520 phages infecting Bacteroidota, the majority (86.2%) were unclassified viruses, indicating the existence of many previously unknown taxa of Bacteroidetes phages in the human vagina.

To elaborate on function profile of the vaginal virome, we annotated the functions of all vOTUs using the KEGG database, resulting in 13.9% of viral genes with KEGG orthology annotations. Among these annotated genes, 49.0% were involved in genetic information processing, while 19.9% were viral auxiliary metabolic genes (Fig. 4e). Focusing on the most prevalent auxiliary metabolic genes in VMGC, many of them were related to the metabolism of peptidoglycan, nucleotide, amino acids, folate and sulfur (Fig. 4f). Further CAZy annotation revealed that the vaginal viruses encoded numerous peptidoglycan lyase/hydrolase-related enzymes (Extended Data Fig. 7), likely facilitating bacterial cell wall degradation during viral infection[44].

## Characteristics of vaginal eukaryotic viruses
Of the 70 eukaryotic vOTUs identified, we specifically focused on the Papillomaviridae viruses (n = 61), considering the wide distribution of human papillomaviruses (HPVs) in the vagina and their implication in

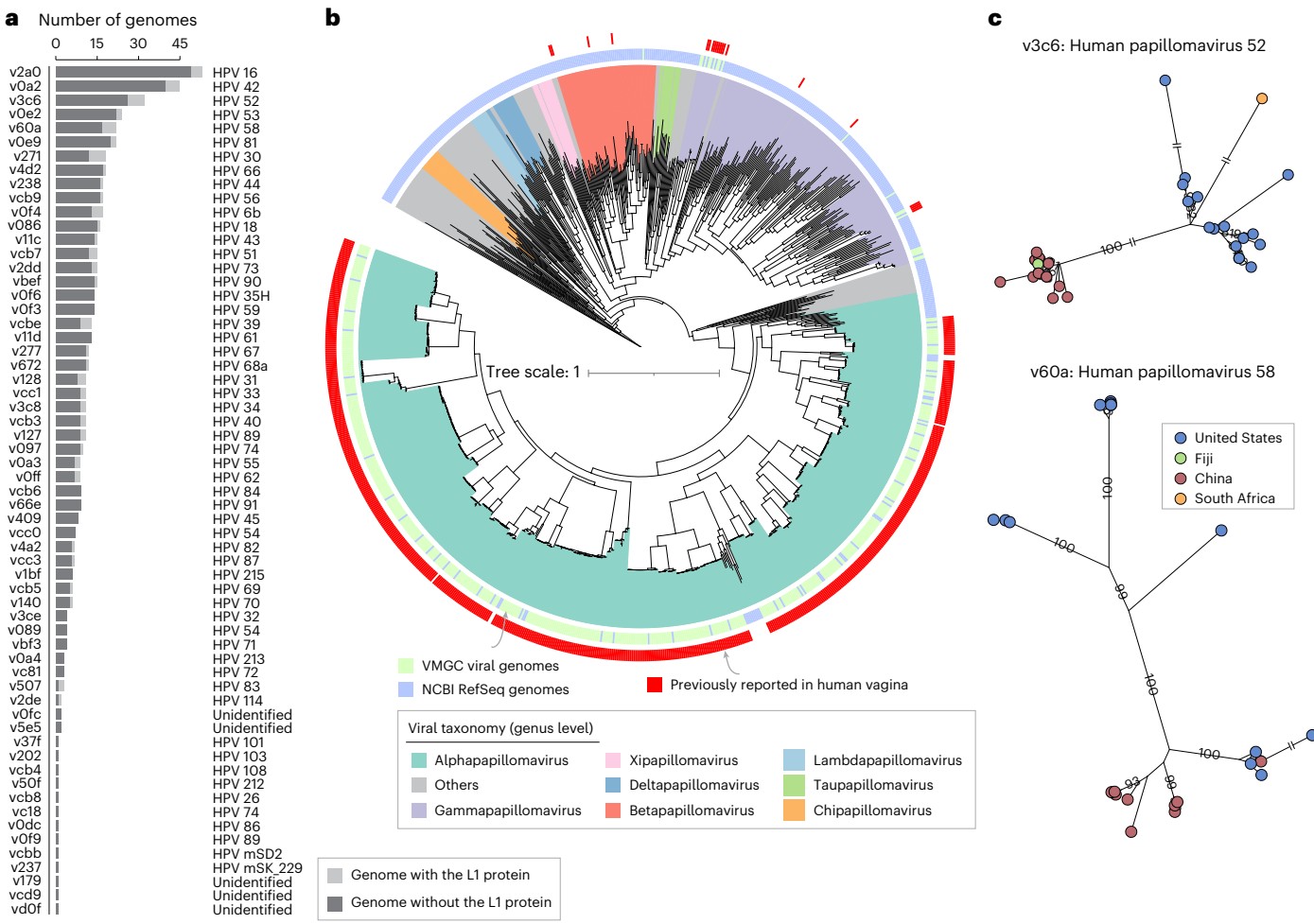

**Fig. 5 | Characteristics of Papillomaviridae members in the VMGC. a**, The number of genome and HPV typing for 61 vOTUs annotated with Papillomaviridae. **b**, Phylogenetic tree based on the L1 proteins of all Papillomaviridae genomes in the VMGC and NCBI RefSeq database. **c**, Phylogenetic trees based on the genomes from HPV52 and HPV58 in the VMGC.

diseases such as cervical cancer[45]. Almost all papillomaviruses in VMGC had complete or high-quality genomes, mostly containing complete *L1* gene (encoding the major capsid protein) for HPV typing (Fig. 5a and Supplementary Table 9). We assigned 61 Papillomaviridae vOTUs into 58 HPV types (including 5 unidentified types) based on their *L1* genes, with 52 types previously discovered in the woman's vagina and 6 types previously found in human penile and skin swabs. Likewise, a phylogenetic analysis of 61 Papillomaviridae vOTUs (*n* = 627 genomes) and the NCBI RefSeq Papillomaviridae viruses (*n* = 475 non-redundant genomes) revealed that our vOTUs represented the major phylogenetic clades of the human vaginal papillomaviruses (Fig. 5b and Extended Data Fig. 8a). Furthermore, based on our large number of genomes, we found that for certain HPV types, their genome evolutionary relationships had obvious regional stratification (Fig. 5c for the examples of HPV types 52 and 58). Compositional analysis of vaginal metagenomes based on VMGC showcased its potential in identifying enrichment of certain HPV types in patients with cervical lesions or cancer (*q* < 0.05; Extended Data Fig. 8b). Collectively, these findings highlighted the value of VMGC Papillomaviridae viruses as a representable reference catalogue for subsequent HPV research.

Besides, three eukaryotic vOTUs, v0116 (Virgaviridae virus), v00f5 and v06d5 (Retroviridae viruses), had not been reported in the human vagina. The Virgaviridae v0116 genome was closely related to the pepper mild mottle virus, a plant virus that had also been found in the human gut, with a 99.8% nucleotide similarity, likely due to dietary ingestion[46]. The Retroviridae v00f5 and v06d5 had a high nucleotide

similarity (99.9% and 97.0%) with a xenotropic murine leukaemia virus-related viruses that had previously been detected in human prostate tumour[47].

**Vaginal microbial gene catalogue and comparison with VIRGO**
To explore the gene content of VMGC, we clustered approximately 26 million protein-coding genes from all prokaryotic, fungal and viral genomes at 50% (VMGC-50), 90% (VMGC-90) and 95% (VMGC-95) amino acid identity, generating three catalogues with 595,219, 1,415,799 and 1,786,695 million non-redundant genes, respectively. Rarefaction analysis revealed that the accumulation curves of VMGC-95 and VMGC-90 were not yet reaching a plateau, whereas the curve of VMGC-50 approached saturation (Fig. 6a), suggesting that VMGC represented an insufficient gene space but relatively saturated gene family space of the human vaginal microbiome. In addition, we compared the gene content of VMGC-90 with VIRGO[10], which represented 0.62 million genes at 90% amino acid identity (referred to as VIRGO-90; Fig. 6b). About 70.6% of genes in VIRGO-90 were covered by VMGC-90, while only 30.8% VMGC-90 genes were also included in VIRGO-90. The remaining genes in VIRGO-90 might be there because this catalogue contained numerous genes from low-abundance microbes or other sequences (for example, plasmids, accessory genes) not binned into MAGs, or high intraspecies diversity in the human vaginal microbiome (Extended Data Fig. 9a)[10]. Nevertheless, VMGC provided an increase of 162% in coverage of the vaginal microbiome protein space over the VIRGO (from 0.62 to a total

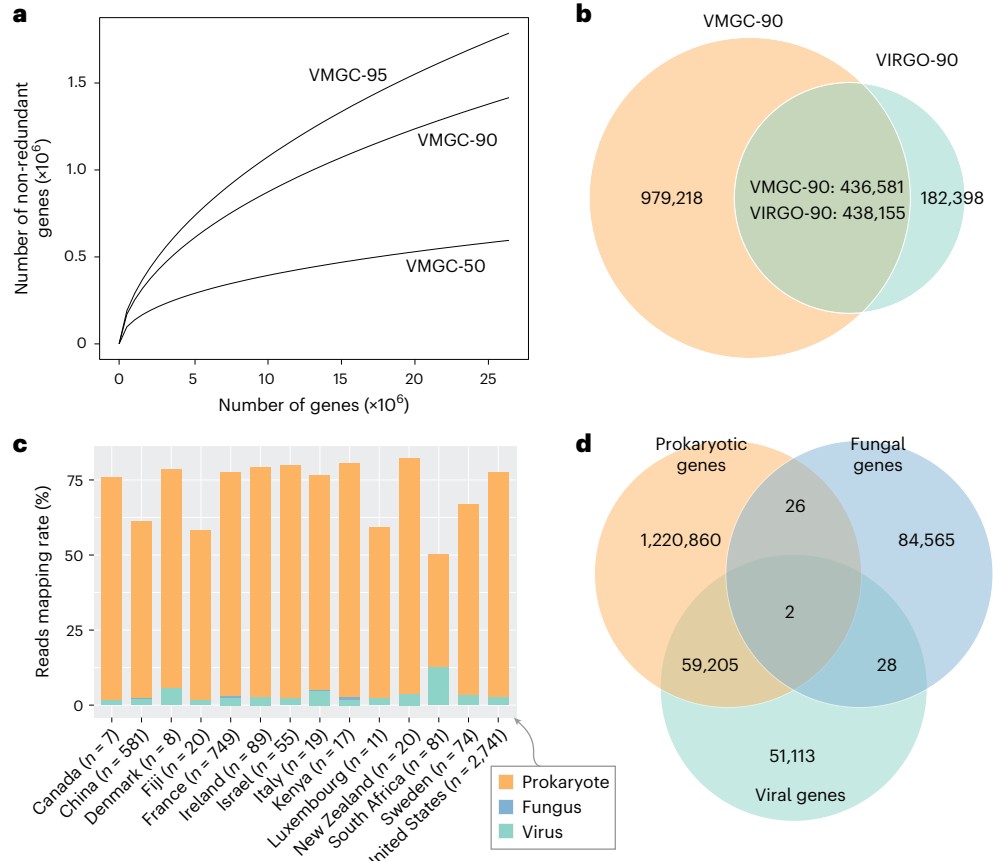

**Fig. 6 | Characteristics of the microbial genes in the VMGC. a**, The accumulation curve of non-redundant proteins as the number of sampled proteins increases. The VMGC-50, VMGC-90 and VMGC-95 represent catalogues with non-redundant proteins clustered at 50%, 90% and 95% amino acid similarity, respectively. **b**, The overlap of proteins between the VMGC-90 and VIRGO-90. The VIRGO-90 contains the non-redundant proteins that were clustered at 90% amino acid similarity from the proteins catalogue corresponding to the public human vaginal non-redundant gene catalogue (VIRGO). **c**, The performance of the gene catalogue corresponding to the VMGC-90 in the recruiting clean reads across vaginal samples. **d**, The overlap of prokaryotic, fungal and viral proteins in the VMGC-90.

of 1.63 million protein clusters). Comparison with other databases also revealed that VMGC-90 significantly expands the protein content of the vaginal microbiota (Extended Data Fig. 9b). Furthermore, mapping of vaginal metagenomes against the vaginal microbial gene and genome catalogues showed an average mapping rate of 74.5% and 83.8%, respectively, in samples from 14 different countries, slightly higher than that of the VIRGO (average 71.7%) (Fig. 6c and Extended Data Fig. 9c,d). The mapping rate was higher in samples from Europe or North America than in those from China, Fiji or South Africa ($p \ll 0.001$), suggesting a relatively low representativeness of VMGC genes in the latter regions.

We further compared the gene content of prokaryotes, fungi and viruses at the VMGC-90 level. Over a half (53.7%) of viral genes were shared with the prokaryotes (Fig. 6d and Extended Data Fig. 10a,b), likely due to the frequent exchange of genetic materials in these two kingdoms, whereas only a few genes were shared between fungi and prokaryotes/viruses. We also compared the functions of prokaryote-specific, fungus-specific and virus-specific genes. As expected, the virus-specific genes mainly encoded the typically viral enzymes involving genetic information processing (for example, DNA replication and repair, and transcription) and prokaryotic defence system, while the prokaryote- and fungus-specific genes had more metabolism-associated genes (Extended Data Fig. 10c).

## Discussion

The VMGC is a large-scale reference sequence resource of the human vaginal microbiome with 33,804 genomes across three microbial kingdoms. A considerable proportion of species in VMGC, including over 25% of prokaryotic and 85% of viral species, have not been successfully cultured. In addition, surpassing the VIRGO database[10], VMGC offers an extensive expansion of microbial gene sequences, enhancing our ability to study the functional potential of the vaginal microbiota.

Functionally, our genome-level analysis found that *L. iners* and *B. vaginale* were the primary contributors of genes associated with pore-forming toxins such as haemolysin and inerolysin, consistent with the previous study[23]. In addition to the common microbial contributors to urease and phospholipase C production in the human vagina such as *Ureaplasma* spp.[23], we have observed frequent encoding of these enzymes in the genomes of *Ralstonia pickettii*, *P. aeruginosa* and *Staphylococcus epidermidis*. For the underexplored prokaryotes, we observed that members of Saccharofermentanales, such as BVAB2, contributed to many genes related to cadaverine production, which has not been reported previously. In addition, Saccharofermentanales members were deficient in the de novo purine biosynthesis pathway, and it is known that similar microorganisms rely on host-derived purines for their growth[33]. Interestingly, the evolutionary trajectory of Saccharofermentanales members appears to be associated with genome reduction, a characteristic feature observed in obligate intracellular parasites of humans[48], providing hints about the nutritional requirements and environmental preferences of these bacteria. These results may provide a reference value for the functional contributions of these bacteria to the vaginal microecosystem and their potential implications in disease development.

The VMGC includes 14,224 viral genomes across 4,263 species, with a significant proportion predicted to infect prokaryotic organisms in the vagina. The exploration of these viruses, particularly those that specifically target vaginal pathogens such as *B. vaginale*, *F. vaginae*, BVABs and so on, offers potential insights into phage therapy against vaginal microbial infections. The VMGC also served as a valuable resource for studying the intra-species diversity of viruses within the vaginal environment and investigating the geographical heterogeneity of their transmission.

In summary, the VMGC significantly advanced our knowledge of the vaginal microbiome, aiding in the development of targeted interventions, diagnostics and personalized approaches for vaginal health. However, it is worth noting some limitations of the VMGC. First is that the microbial sequences captured by the VMGC are not complete, especially for vaginal samples from the non-western population. The second is that the VMGC lacked fungal genomes from members of *Cladosporium* and *Malassezia*, among others which were observed in the vaginal samples using the ITS sequencing approach[49]. Third, many phages in the VMGC are unknown, and their functional exploration based on existing databases remains limited. Further isolation, cultivation and physiological studies are necessary to gain a deeper understanding of their genomic information and potential contributions to vaginal health. Another concern is contamination from experimental operations or kits in public vaginal metagenome datasets used for constructing the VMGC. Although our assessment indicated a low level of contamination (Supplementary Fig. 1), caution is still warranted when utilizing the VMGC.

## Methods

### The retrieval of prokaryotic and fungal genomes in the NCBI database

We searched for human vaginal microbial genomes from approximately 520,000 prokaryotic and 13,000 fungal genomes recorded in the NCBI genome database (https://www.ncbi.nlm.nih.gov/genome/browse/) until April 2023. We first excluded genomes with any mention of the following keywords in their NCBI BioSample information (https://ftp.ncbi.nlm.nih.gov/biosample/): 'uncult*', 'MAG' and 'bin'. Subsequently, we extracted genomes that included any mention of the following keywords in their NCBI BioSample information: 'ovary', 'vagin*', 'endomet*', 'genita*', 'cervi*', 'uterus', 'reprodu*', 'posterior', 'douglas' and 'fallopian'. This process yielded approximately 2,100 candidate genomes. To further refine our selection, we manually checked and extracted genomes of cultured isolates from the candidate genomes based on the metadata records in their BioSample information, Genomes Online Database information (https://gold.jgi.doe.gov/) or original studies. This refinement resulted in 1,189 prokaryotic and 18 fungal isolate genomes. These genomes then underwent genome quality filtering. For prokaryotic genomes, we assessed their quality using CheckM2 v1.0.1[31] and GUNC v1.0.5 tools[50]. We retained 972 genomes meeting the criteria of ≥50% completeness, <5% contamination, a genome quality score of ≥50 (defined as %completeness − 5×%contamination) and a clade separation score of <0.45 (Supplementary Table 3). In terms of fungal isolate genomes, the quality assessment was performed using BUSCO v5.4.2 with the parameter '-m genome' based on the fungi_odb10.2019–11–20 database[32]. We retained 17 fungal genomes with ≥50% completeness and <5% contamination (Supplementary Table 4).

### Collection of vaginal metagenomic samples

We performed an extensive search in the NCBI database until October 2023, targeting human vaginal metagenomic samples annotated as vagin*, cervi*, endomet* and so on. The search yielded a collection of 3,123 vaginal metagenomes from 32 different studies (Supplementary Table 1). Furthermore, we obtained additional 1,366 vaginal metagenomic samples from the VIRGO website (https://virgo.igs.umaryland.edu/)[10], after removing duplicates found in the NCBI samples. The accession ID of each sample is provided in Supplementary Table 2.

### Fungal culture and shotgun genome sequencing

The study received approval from the Ethics Committee of the Second Affiliated Hospital of Dalian Medical University (number DMU20210082). Informed consent was obtained from all volunteers. Fungal cultivation was performed based on fresh specimens of genital tract secretions from three healthy women: subject 1 (21 years old), subject 2 (36 years old) and subject 3 (47 years old). The vaginal samples were collected by a gynaecologist from vaginal fornix with a cotton swab. Before the examination, the recruited women were asked to avoid vaginal douching, sexual intercourse and other vaginal manipulations for 3 days. Women with vaginal bleeding or vaginal and cervical inflammation were excluded from the research.

A total of 1 ml of suspension was inoculated immediately onto each agar plate containing fungal culture medium, which was incubated until colonies were observed under aerobic and anaerobic conditions (32 °C). After incubation for 3–14 days, various fungal colonies were observed on the cultured agar plates, while no fungal colonies were observed on the agar plates for the culture of the sterile diluent buffer as the negative control. From each incubated agar plate, phenotypically distinct colonies were picked onto the corresponding fresh medium for further purification. Then, a single colony was picked and restreaked on the corresponding medium to purify the fungal strains. The purified fungal strains for identification and storage were inoculated in a 5 ml modified Martin broth medium under the same culturing conditions (32 °C, 130 r.p.m. shaking). Shotgun genome sequencing of the cultured vaginal fungi was performed based on the Illumina NovaSeq platform for 2 × 150 bp paired-end sequencing. Sequencing libraries were prepared by using the NEBNext Ultra DNA Library Prep Kit following the manufacturer's recommendations.

### Preprocessing and assembly

To ensure data quality, raw reads from each sample were subjected to quality control using fastp v0.20.1[51] with the parameters '-u 30 -q 20 -l 30 -y --trim_poly_g' for samples with read lengths of 75 bp or less and '-u 30 -q 20 -l 60 -y --trim_poly_g' for samples with read lengths greater than 75 bp. A secondary complexity filter was applied to each sample using bbduk with the parameters 'entropy=0.6 entropywindow=50 entropyk=5' from the BBTools suite v39.00[52]. Subsequently, the reads deriving from human and *Escherichia* phage phiX174 were removed by mapping the quality-filtered reads against the CHM13v2.0 and phiX174 (NCBI accession NC_001422.1) genomes using Bowtie2 v2.4.1 with the parameters '--end-to-end --fast'[53]. The remaining reads were considered clean reads for each sample. As a result of this preprocessing, a total of 4,472 vaginal metagenomic samples, representing approximately 5.0 Tbp high-quality non-human metagenomic data, were available for further analysis (Supplementary Table 2). The clean reads were de novo assembled into contigs for each sample using MEGAHIT v1.2.9[54] with the parameters '--k-list 21,41,61,81,101,121', resulting in over 4.1 million long contigs (minimum length of 2 kbp; total length of 38.3 Gbp; with 38.9% of these contigs longer than 5 kbp).

### Genome analysis for prokaryotic populations

**MAGs.** Only assembly files containing contigs with lengths greater than 2,000 bp were used to recover MAGs. Referring to recent studies[55,56], we developed a 'mash-based multi-coverage binning approach' based on Mash[57] to select a specific number of samples for multi-sample read alignment and MAG recovering. The specific approach is outlined as follows: (1) calculate the Mash distance among the assembled contig sets of all sample pairs using Mash v2.3 with the parameters '-s 100000 -k 32'[57]; (2) for each sample, we selected the *N* most similar samples based on Mash distance, mapping their reads to the contigs of the target samples, and calculated the contigs' sequencing depth in the most similar samples using the jgi_summarize_bam_contig_depths tool[58]; (3) for each target sample, we integrated the

depth files of its $N$ most similar samples using combine.pl (https://github.com/WatsonLab/single_and_multiple_binning/blob/main/scripts/combine.pl) to obtain the contigs' multi-sample depth. Based on this integrated depth file, we performed multi-coverage metagenomic binning for the contigs using MetaBAT 2 with the parameters '-m 2000 -s 200000 --seed 2020'[58], ultimately obtaining the raw bins for each sample. To determine the optimal value for $N$, representing the number of most similar samples used for calculating sequencing depth, we randomly selected 80 vaginal samples and conducted multi-coverage binning processes for each sample using the top 10–60 closest samples (from 4,472 samples) based on Mash distance. This test revealed that, compared to single-coverage binning, all multi-coverage binning approaches, regardless of selecting 10–60 samples, yielded a significantly higher number of MAGs and a lower proportion of heterozygous MAGs (Supplementary Fig. 2). After $N$ reached 20, the trend of increasing MAG quantity approached a plateau, indicating that selecting the 20 closest samples for multi-coverage binning is appropriate for vaginal samples. Using the described approach, we applied multi-coverage binning to all vaginal metagenomic samples. In addition, we performed single-coverage binning for each sample. Subsequently, using dRep v3.4.0 with the parameters '-pa 0.9 -sa 0.99 -nc 0.3 --S_algorithm fastANI'[59], we conducted strain-level de-redundancy (99% average nucleotide identity (ANI)) for quality-controlled (see Quality assessment of prokaryotic genomes section) MAGs obtained from both multi-coverage and single-coverage binning within each sample, resulting in a total of quality-controlled 18,570 MAGs (where 4,713 and 13,857 come from single-coverage and multi-coverage binning, respectively).

**Quality assessment of prokaryotic genomes.** The completeness and contamination of prokaryotic genomes (for example, MAGs and isolated genomes) were estimated using CheckM2 v1.0.1[31], and the genome chimerism of prokaryotic genomes was evaluated by GUNC v1.0.5[50]. Genomes with <50% completeness, ≥5% contamination, quality score of <50 or clade separation score of ≥0.45 were removed for further analyses. During the process of categorizing the integrity of prokaryotic genomes, we referred to the revised MIMAG standard[17]. In general, the medium-quality MAGs were characterized by ≥50% completeness and <5% contamination; the high-quality MAGs showed ≥90% completeness and <5% contamination; the near-complete MAGs were defined as the high-quality MAGs with the presence of 5S, 16S and 23S rRNA genes, as well as at least 18 types of transfer RNA. The identification of rRNA within MAGs was performed using cmsearch in INFERNAL v1.1.4[60], while the identification of tRNAs was conducted using tRNAscan-SE v2.0.11[61] with the parameters '-L -B'.

**Clustering, taxonomic and phylogenetic analysis.** All prokaryotic MAGs ($n = 18,570$) and isolate genomes ($n = 972$) were clustered at the species level by dRep v3.4.0[59] with the parameters '-pa 0.9 -sa 0.95 -nc 0.3 --S_algorithm fastANI', resulting in 786 SGBs. For each SGB, MAGs are sorted in the order of quality groups: near-complete, high-quality and medium-quality. Within each group, they are further sorted based on the genome quality score from high to low. The top-ranked MAG was considered as the reference genome for the corresponding SGB. Taxonomic classification of the SGBs was carried out by the classify_wf model in the Genome Taxonomy Database Toolkit (GTDB-Tk) v1.4.0 with GTDB database r214.1[62]. The phylogenetic tree created by GTDB-Tk was visualized using iTOL v6.7.4 (https://itol.embl.de/)[63]. In addition, the phylogenetic diversity of each phylum was calculated based on the aforementioned phylogenetic tree using the function 'pd' in the R package picante v1.8.2.

**Taxonomic profiles in vaginal metagenomic samples.** We used an in-house script to construct Kraken2 and Bracken databases comprising 786 prokaryotic species in the VMGC. The clean reads were mapped against these prokaryotic genomes using Kraken v2.1.3 with confidence score 0.1 and other default parameters[64]. Based on these alignment results, Bracken v2.8 was used to assess the number of reads recruited by each taxon[65]. In the Bracken output, the fraction of total reads for each taxon is considered as the relative abundance of that taxon in the sample.

**Functional annotation.** We performed gene prediction of prokaryotic genomes in VMGC using Prodigal v2.6.3[66] with the parameter '-p meta'. To characterize the function of these predictive proteins, we searched for them in KEGG[21], CAZy[22], Virulence Factor Database[67] and the combined antimicrobial resistance databases (customized in our previous study[43]) using diamond v2.0.13.151[68]. For KEGG and CAZy annotations, a protein was successfully assigned into a specific KEGG functional ortholog or CAZyme if it achieved a score of at least 60 across 50% of the sequences. For Virulence Factor Database annotations, a protein was successfully assigned into a specific virulence factor if it achieved a 60% identity across 50% of the sequences. For different types of antimicrobial resistance gene, we applied specific criteria to annotate proteins based on the method provided by a previous study[69]. These criteria included the following: (1) proteins associated with beta-lactamases were annotated based on a minimum amino acid similarity of 90%, (2) proteins associated with multiple drug resistance were annotated based on a minimum amino acid similarity of 70% and (3) for other types of antimicrobial resistance gene, a minimum amino acid similarity of 80% was required for annotation.

**Quantification of functional family.** To quantify the functional contribution of prokaryotic species to the vaginal ecosystem, we calculated the weighted abundance of each functional ortholog (for example, KEGG functional ortholog or CAZyme) within different SGBs. For SGB $i$, if it has $N_{i,j}$ genes annotated with a functional ortholog $j$, the weighted abundance of this ortholog in a sample can be calculated by $W_{i,j} = \frac{N_{i,j}}{R_i}$, where $R_i$ is the relative abundance of the SGB $i$ in this sample. In terms of each functional module (for example, lactate production), its weighted abundance was calculated by summing the weighted abundances of the functional orthologs associated with this functional module (Supplementary Table 7).

**Genomic analysis of Saccharofermentanales members.** In addition to CheckM2, we also used CheckM v1.1.3[70] and BUSCO v5.4.2[32] (with the parameter '-m genome' based on the bacteria_odb10.2019-06-26 database) to estimate the completeness and contamination of all Saccharofermentanales genomes in the VMGC. Both CheckM v1.1.3 and BUSCO showed <90% completeness of these genomes including three complete genomes (NCBI accession GCF_029101565.1, GCF_000025225.2 and GCF_029101585.1). Moreover, we built a phylogenetic tree comprising 14 Saccharofermentanales species within the VMGC, along with all Saccharofermentanales genomes ($n = 192$) recorded in GTDB database r214.1[62], using PhyloPhlAn v0.99[71].

## Genome analysis for fungal populations

**Identification of fungal MAGs.** We tried to identify fungal genomes from the pool of over 63,000 raw MAGs. To reduce the computational burden, we first extracted 371 raw MAGs that had a genomic size of over 3 Mb. Subsequently, non-eukaryotic sequences were removed from each bin using EukRep v0.6.7 with the parameter '-min 2000'[72], which generated 38 MAGs with a genomic size of over 3 Mb. The genome quality of these MAGs was assessed using BUSCO v5.4.2 with the parameter '-m genome' based on the fungi_odb10.2019-11-20 database. The 30 MAGs showed >50% completeness and <5% contamination and were considered as potential fungal genomes. These 30 MAGs further underwent intra-sample and strain-level deduplication (99% ANI) by dRep v3.4.0 with the parameters '-pa 0.9 -sa 0.99 -nc 0.3 -S_algorithm fastANI'[59], resulting in 13 fungal MAGs.

**Taxonomic classification and clustering of fungal MAGs.** We performed a preliminary taxonomic classification for 13 MAGs based on their internal transcribed spacer 1 (ITS1) sequences. The ITS1 sequences were extracted using QIIME v2021.2.0[73] with the parameters '-p-f-primer ACCTGCGGARGGATCA -p-r-primer GAGATCCRTTGYTRAAAGTT -p-max-length 300'. Subsequently, we applied QIIME with the parameters 'feature-classifier classify-sklearn' to annotate the ITS1 sequences based on the UNITE fungal ITS database version 8.3[74]. Based on ITS1 annotation, we further estimated the ANI between the MAGs and NCBI fungal genomes that shared identical taxonomic assignments at the genus level. This was performed using dRep v3.4.0 with the parameters 'dereplicate -pa 0.9 -sa 0.95 -nc 0.3 --S_algorithm fastANI'. Based on the ANI results, the MAGs were identified as known species (>95%) and genera (>80%). Likewise, all fungal genomes in the VMGC were grouped into the species-level clusters at 95% ANI.

**Phylogenetic analysis of fungal genomes.** We constructed a phylogenetic tree for 13 fungal MAGs and 25 fungal isolate genomes based on their single-copy protein markers. First, the prediction of fungal protein-coding genes was implemented using GeneMark-ES v4.68_lic[75] with the parameters '--fungus --ES --min_contig 20000', leading to 222,113 predictive proteins. All predictive protein sequences were clustered into 57,230 protein families at 50% identity level with MMseqs2 v12.113e3[76]. Among these protein families, we identified 88 protein families as single-copy protein markers. These markers had to appear only once in over 80% of fungal genomes. We also ensured that the protein sequences within each marker cluster showed a similar sequence size, with a coefficient of variation of 20%. The protein markers from each genome were combined into a concatenated sequence. Furthermore, all concatenated sequences were aligned using MAFFT v7.475[77], and a phylogenetic tree was constructed using IQ-TREE v2.1.2[78] with the parameters '-m MFP -bb 1000'.

**Taxonomic profiles in vaginal metagenomic samples.** First, the clean reads of vaginal metagenomic samples were mapped against fungal species-level genomes in the VMGC using Bowtie2[53]. To improve alignment specificity and minimize read contaminations from non-fungal populations, the clean reads mapped into fungal genomes were further aligned against the following databases: (1) ITS sequences from the UNITE database, (2) 16S ribosomal DNA sequences from the SILVA 138 99% OTUs reference database[79] and (3) all prokaryotic genomes in the VMGC. Any reads that were mapped to these databases were excluded from the taxonomic profiling process. To be considered present in a specific vaginal sample, a fungal species was required to capture at least three clean reads. The prevalence rate of a fungal species is then calculated based on this presence, determined by the number of vaginal samples with reads mapped against the species divided by the total number of samples. For the species-level profiles of each vaginal sample, the read count of each species was first normalized by dividing its genomic size. The normalized read count was then divided by the sum of all normalized read counts in the sample, defining the relative abundance of a species.

**Genome analysis for viral populations**
**Identification of virus sequences.** We carried out the virus analysis within vaginal metagenomic samples according to our previous studies[43,80]. Assembled contigs with lengths of >5,000 bp were used for virus identification. We initially evaluated contigs using CheckV v0.7.0[18], removing those with a host gene count surpassing 10 and a host gene count exceeding five times the number of viral genes. Provirus fragments were detected and extracted from the contigs by CheckV. For the remaining sequences, contigs meeting any of the following criteria are considered possible virus sequences: (1) a higher number of viral genes compared to host genes according to CheckV v0.7.0, (2) detection as viruses using the thresholds of score >0.90 and $P < 0.01$ in DeepVirFinder v1.0[81] and (3) identification as viruses by VIBRANT

v1.2.1[82] using default parameters. To reduce non-viral sequence contaminations, we identified bacterial universal single-copy orthologs (bacterial USCOs) within the possible viral genomes using hmmsearch[83] with default parameters. The bacterial USCO ratio, defined as the number of bacterial USCOs divided by the total number of genes in each viral genome, was then calculated to assess the level of potential contamination. The genomes showing ≥5% bacterial USCO ratio were subsequently eliminated from the analysis. The remaining genomes with CheckV-estimated completeness of >50% were designated as the final viral genomes in the VMGC (VMGC-v).

**Genome clustering and phylogenetic tree.** The viral genomes were clustered into a vOTU based on a nucleotide similarity threshold of 95% across 85% of the genome using BLASTn v2.12.0 with the options '-evalue 1e-10 -word_size 20'. The representative genome was defined as the largest viral sequence within each vOTU. This clustering approach was also applied to evaluate the overlap of viral sequences among different virome databases (VMGC viruses, Oral Virus Database[43], Gut Virome Database[40], Gut Phage Database[41], Metagenomic Gut Virus catalogue[42], NCBI RefSeq-viruses). To understand the phylogenetic relationships between genomes, we constructed a viral proteomic tree using the ViPTreeGen v1.1.2 with the default parameter[84].

**Taxonomic classification.** The taxonomic classification of viral sequences was carried out by searching for their protein sequences in a combined database. This database was generated using proteins from Virus-Host DB (downloaded in May 2021)[85], crAss-like proteins from Guerin's study[86] and viral proteins from Benler's study[87] and Ye's study[88]. The protein-coding sequences of viral genomes were predicted by Prodigal v2.6.3[66] with the parameter '-p meta', and blasted against the combined database using diamond v2.0.13.151 with the parameters '--id 30 --subject-cover 50 --query-cover 50 --min-score 50'. For small viral genomes with a gene count of less than 30, a viral genome was annotated into a known viral family based on the criterion that more than one-fifth of its proteins showed a match to the same family. Conversely, for large viral genomes with a gene count of 30 or more, a viral sequence was assigned to a known viral family if at least 10 of its proteins were matched to the same family.

**Virus-host prediction.** Virus-host prediction was conducted based on all prokaryotic genomes (19,542 MAGs) in the VMGC. We applied various criteria to detect potential prokaryotic hosts of viruses, including the following approaches: (1) CRISPR (clustered regularly interspaced short palindromic repeats) spacer sequences were predicted in prokaryotic MAGs using MinCED v0.4.2 with the parameter '-minNR 2'[89]. If a host CRISPR spacer sequence matched to a viral genome with a bit-score of 45 or higher using BLASTn (with the parameters '-evalue 1e-5 -word_size 8'), the virus was assigned to that host. (2) The viral sequence was blasted against host genome sequences and assigned a host if the viral sequence was matched to the host genome with ≥90% nucleotide identity and ≥30% viral coverage.

**Phylogenetic analysis of Papillomaviridae members.** We classified the Papillomaviridae members in the VMGC into traditional papillomavirus types based on their L1 structural protein. We extracted the coding sequence labelled L1 structural proteins from all Papillomaviridae members available in the NCBI database. For our Papillomaviridae genomes, we selected putative *L1* structural genes with a minimum length of 1,300 bp and shared a nucleotide identity of at least 40% with known *L1* structural genes. Based on L1 sequence similarity, the Papillomaviridae members were identified as known types (>90%), species (>70%) and genera (>60%)[90]. A phylogenetic tree was built based on available *L1* structural genes from all papillomavirus in the VMGC and NCBI database using mafft-sparsecore.rb[77] and IQ-TREE tools v2.1.2 (ref. [78]).

**Function annotation.** The viral function annotation followed the same method used for the annotation of prokaryotic protein-coding sequences as described above.

## Vaginal microbial gene catalogue

We created a comprehensive gene catalogue of the vaginal microbiome by combining all putative protein-coding sequences derived from prokaryotic, fungal and viral genomes in the VMGC. The clustering of protein sequences was implemented using the easy-linclust model in MMseqs2 v12.113e3[76]. This clustering approach was also applied to evaluate the overlap of protein sequences between the VMGC and VIRGO (a comprehensive vaginal gene catalogue) based on an amino acid similarity of 90% across 80% of the sequence. Furthermore, we compared the performance of the VMGC and VIRGO on recruiting clean reads from vaginal metagenomic samples. We mapped the clean reads from each sample against all gene sequences from VMGC and VIRGO, respectively, using Bowtie2 v2.4.1 with the parameter '--end-to-end --mm --fast'. The mapping read rate was calculated as the number of mapped reads divided by the total number of clean reads in each sample. It is worth noting that, when calculating the mapping rate based on the genome set for VMGC, this set comprises genomes from 786 prokaryotic SGBs, 11 fungal species and 4,263 vOTUs in the VMGC.

## Statistics and reproducibility

We did not use a statistical method to determine the sample size, and the vast majority of collected samples are included in the analysis, with filtered samples typically being excluded due to lack of available clean reads after data quality control or mapping to the reference database. Data collection and analysis were not conducted blindly to the experimental conditions, except for fungal cultivation. Volunteers providing vaginal samples for fungal cultivation were not randomized; specific exclusion criteria have been described in the 'Fungal culture and shotgun genome sequencing' section. Given the sparse and non-normally distributed microbial data, relevant statistical analyses were performed using non-parametric tests such as the Wilcoxon signed-rank test.

## Reporting summary

Further information on research design is available in the Nature Portfolio Reporting Summary linked to this article.

## Data availability

The data files of VMGC, including prokaryotic, eukaryotic and viral genome sequences, annotation files and the updated Kraken database, have been deposited in the Zenodo repository with the accession ID 10457006 (https://doi.org/10.5281/zenodo.10457005) (ref. 91). The assembled genomes of the cultivated fungal strains were deposited in the NCBI database with BioProject accession ID PRJNA1100704. All cultivated fungal strains in this study are maintained at Dalian Medical University and can be obtained from the corresponding authors upon request. Source data are provided with this paper.

## Code availability

The metadata, intermediate results, and analysis and visualization codes used in this study have been uploaded into the GitHub repository, accessible at https://github.com/RChGO/VMGC.

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

## Acknowledgements

This work was supported by Science and Technology Innovative Project of Bao'an, Shenzhen (grant no. 2019JD318 to L.H.); COVID-19 Research & Application Project, Shenzhen Bao'an Chinese Medicine Developing Fund (grant no. 2020KJCX-KTYJ-76 to L.H.); and National Natural Science Foundation of China (grant no. 82370563 to Q.Y.).

## Author contributions

L.H., S.L., Q.Y. and W.S. conceived the study. R.G., S.L., Yue Zhang, J.K., J. Meng, H.Y. and J.Z. performed the analyses. X.W., S.G., Y.L., Z.X., P.Z., J. Ma, W.Y., Yan Zhang, G.H. and Z.D. performed culturing and sequencing experiments. L.H. and W.S. supervised the work and provided funding. S.L. and R.G. wrote the paper. All authors read, edited and approved the paper.

## Competing interests
The authors declare no competing interests.

## Additional information
**Extended data** is available for this paper at https://doi.org/10.1038/s41564-024-01751-5.

**Correspondence and requests for materials** should be addressed to Shenghui Li, Qiulong Yan or Wen Sun.

[1]Department of Reproductive Health, Shenzhen Bao'an Chinese Medicine Hospital, Guangzhou University of Chinese Medicine, Shenzhen, China. [2]Puensum Genetech Institute, Wuhan, China. [3]Department of Microbiology, College of Basic Medical Sciences, Dalian Medical University, Dalian, China. [4]Department of Obstetrics and Gynecology, First Affiliated Hospital of Dalian Medical University, Dalian, China. [5]Department of Acupuncture, Shenzhen Bao'an Chinese Medicine Hospital, Guangzhou University of Chinese Medicine, Shenzhen, China. [6]School of Traditional Chinese Medicine, Beijing University of Chinese Medicine, Beijing, China. [7]Department of Acupuncture and Moxibustion, Beijing Hospital of Traditional Chinese Medicine, Capital Medical University, Beijing, China. [8]Department of Traditional Chinese Medicine, Beijing Friendship Hospital, Capital Medical University, Beijing, China. [9]Department of Gynecology, Shenzhen Bao'an Chinese Medicine Hospital, Guangzhou University of Chinese Medicine, Shenzhen, China. [10]Centre for Translational Medicine, Shenzhen Bao'an Chinese Medicine Hospital, Guangzhou University of Chinese Medicine, Shenzhen, China. [11]Key Laboratory of Health Cultivation of the Ministry of Education, Beijing University of Chinese Medicine, Beijing, China. [12]These authors contributed equally: Liansha Huang, Ruochun Guo, Shenghui Li. ✉e-mail: lsh2@qq.com; qiulongy1988@163.com; sunwen@bucm.edu.cn

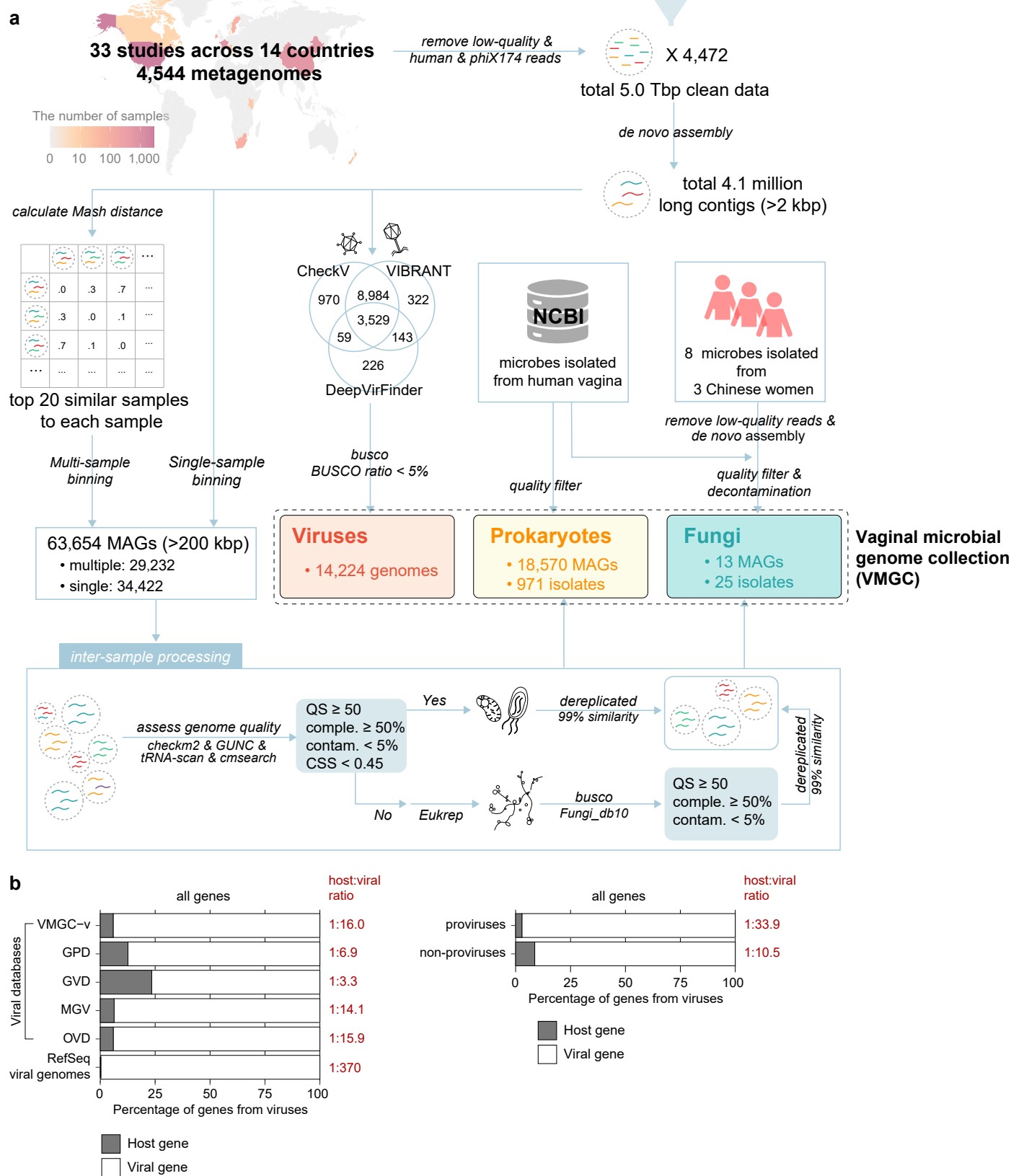

**Extended Data Fig. 1 | Supplemental figure for the construction of the VMGC.**
**(a)** Data sources and processing workflow for the construction of the VMGC. MAG, metagenome-assembled genome. QS, quality score. CSS, clade separation score. **(b)** Host:virus gene ratio analysis of the viral sequences in VMGC. For each viral sequence, the 'viral genes' and 'microbial host genes' were classified by CheckV based on a database created using known virus and prokaryotic genes, and the 'host:viral gene ratio' was calculated to estimate the potential microbial

host gene contamination for the viral sequences in VMGC. Left panel, estimated for all viruses of VMGC and compared with other existing viral databases; right panel, estimated for the proviruses and non-proviruses of VMGC. The 'host:viral gene ratio' for all VMGC viruses was 1:16.0 (1:33.9 for proviruses and 1:10.5 for non-proviruses), suggesting minimal potential for microbial host gene contamination within our viral collection.

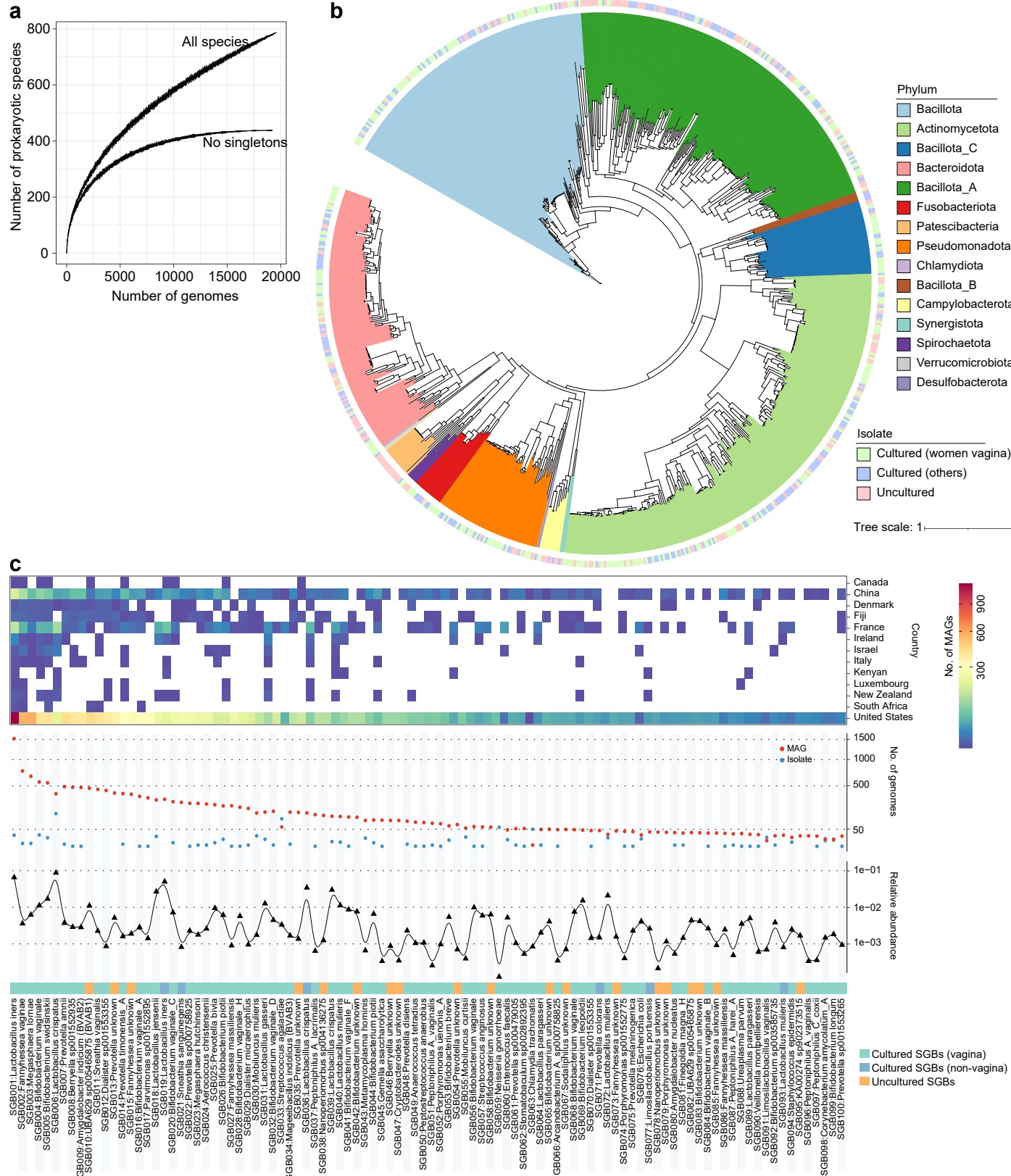

**Extended Data Fig. 2 | Summary of prokaryotic species in the VMGC.**
**(a)** Rarefaction curves of the number of species detected as a function of the number of nonredundant genomes analyzed. Curves are depicted both for all the prokaryotic species and after excluding singleton species (represented by only one genome). **(b)** Phylogenetic tree for 786 prokaryotic species. The inner and outer layers depict the phylum-level taxonomic and cultivation information of the species, respectively. **(c)** The top 100 species with the highest number of genomes in VMGC. Upper panel, heatmap showing the geographic distribution of MAGs of the prokaryotic species. Middle panel, the number of MAGs and isolated genomes of the prokaryotic species. Bottom panel, the average relative abundance of the prokaryotic species.

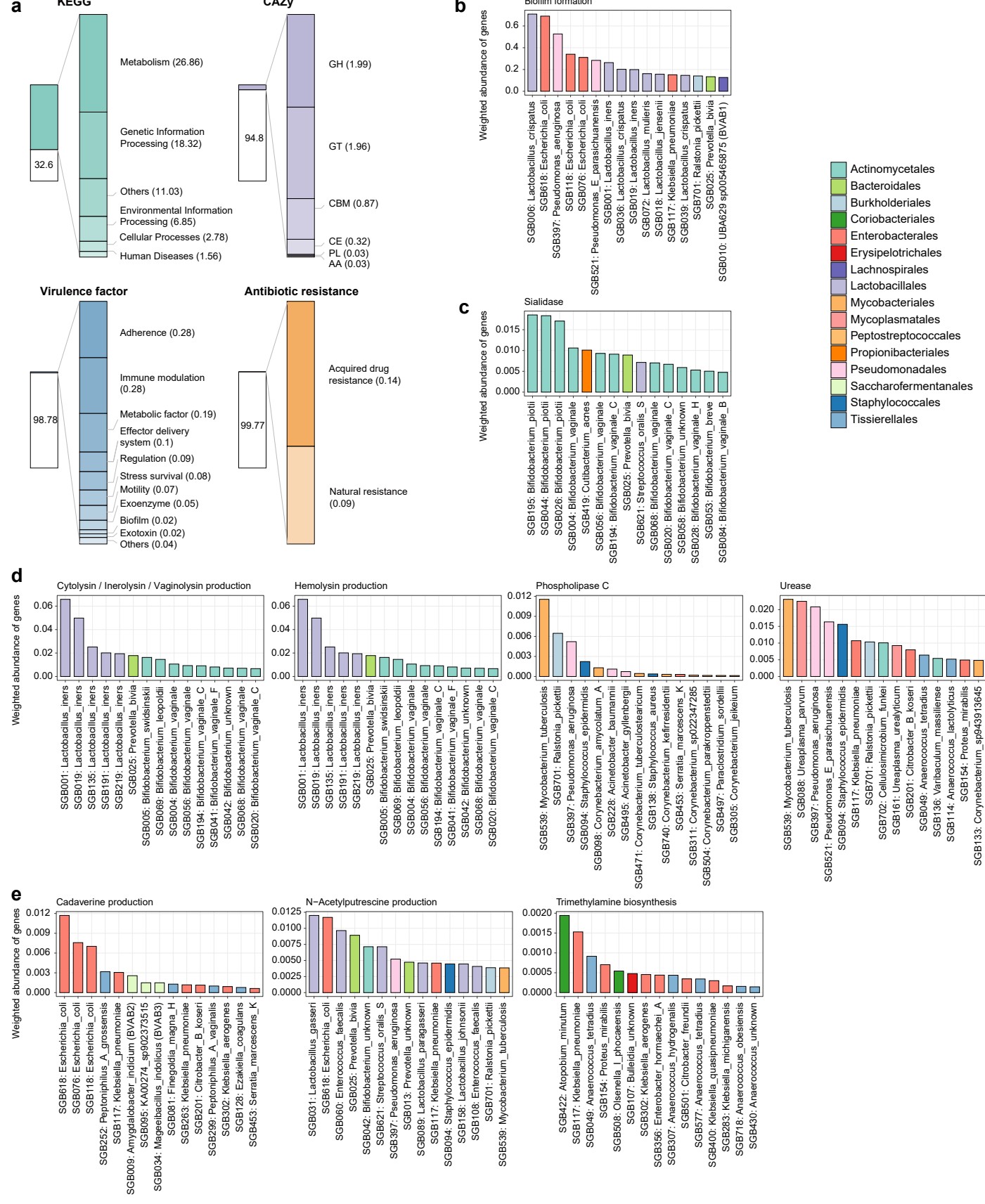

**Extended Data Fig. 3 | See next page for caption.**

**Extended Data Fig. 3 | Summary of prokaryotic functions in the VMGC.**
**(a)** Proportions of annotated proteins in all putative proteins from 786 prokaryotic species using different functional annotation databases. The left bar plot shows the proportion of unannotated proteins (white block), while the right bar plot displays the proportion of proteins annotated into different functional modules. **(b-e)** The mean weighted abundance of disease-associated functional modules in different prokaryotic species across the vaginal samples. Each bar plot presents the top 15 species with the highest abundance. The species are colored according to their order-level taxonomic classification. The results of functional modules are grouped into 4 categories: **(b)** biofilm formation (mainly to protect harmful bacteria), **(c)** sialidase (disruption of the vaginal mucosal barrier, **(d)** toxins (cytolysin, hemolysin) and enzymes (urease, phospholipase C) (disruption of the epithelial barrier and induction of inflammation), **(e)** biogenic amines (cadaverine, N-acetylputrescine, and trimethylamine) (mainly to produce unpleasant odor, elevating pH to promote the growth of harmful bacteria).

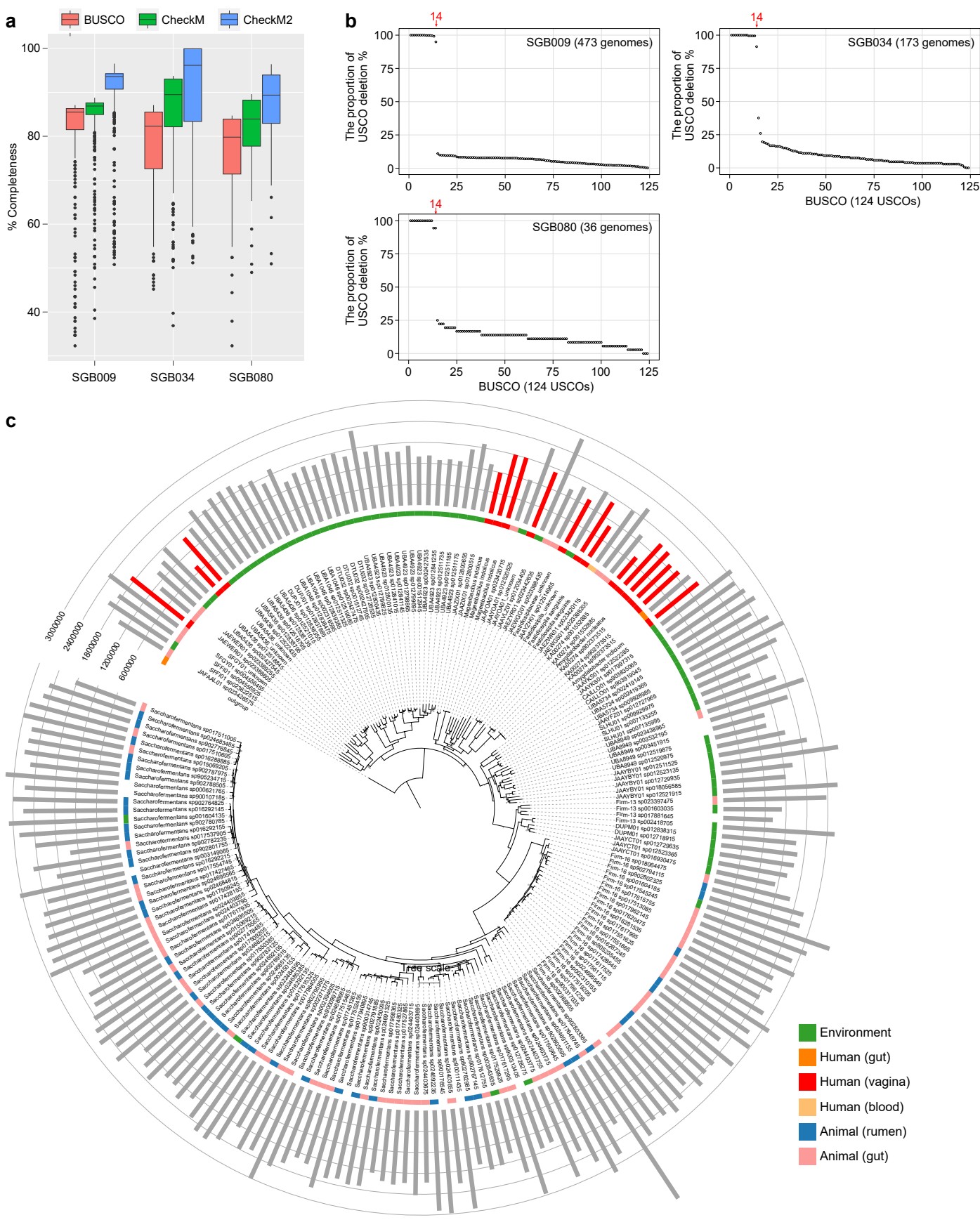

**Extended Data Fig. 4 | See next page for caption.**

**Extended Data Fig. 4 | Completeness of three dominated species of Saccharofermentanales. (a)** Boxplot showing the completeness of three species, estimated by BUSCO, CheckM, and CheckM2. In the boxplot, the center line represents the median, the box limits show the upper and lower quartiles, the whiskers extend to 1.5 times the interquartile range, and points outside the whiskers are considered outliers. **(b)** The missing proportions of 124 universal single-copy orthologs (USCOs) across the genomes of 3 species of Saccharofermentanales. Each point in the figure represents a specific USCO and is sorted based on the corresponding missing proportion. The total number of genomes within each species is shown in the bracket located in the upper right corner of the figure. **(c)** Phylogenetic tree of all Saccharofermentanales genomes in the VMGC and the GTDB-tk database. The inner and outer layers depict the sources and genome sizes of the species, respectively.

a   The overlapment of KOs among three species

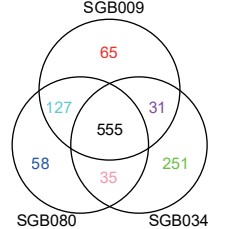

b   The overview of metabolism pathways in SGB009, SBG034, and SGB080

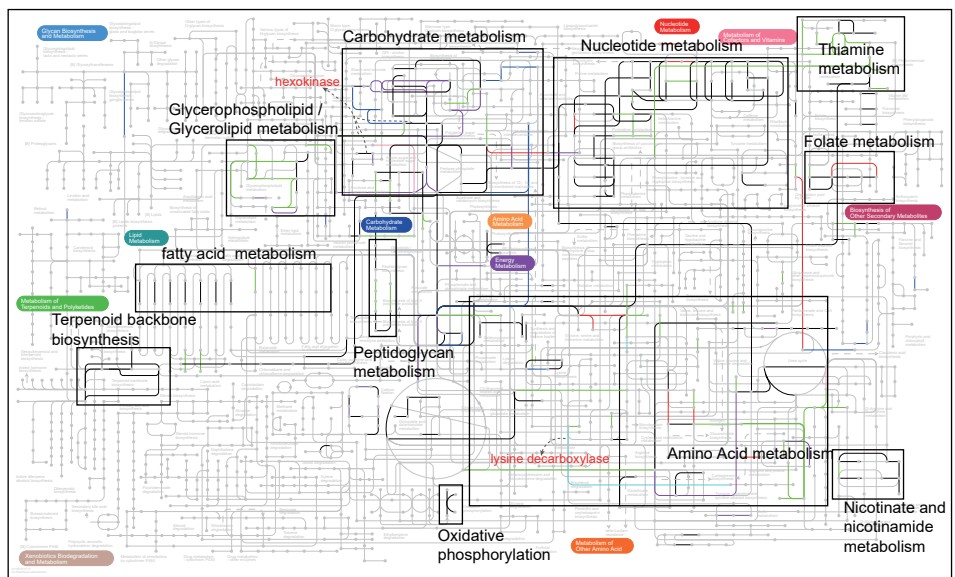

**Extended Data Fig. 5 | Functional comparison among SGB009, SGB034, and SGB080 based on the KEGG annotation. (a)** The overlap of KEGG functional orthologs (KOs) among SGB009, SGB034, and SGB080. **(b)** The overview of KEGG metabolism pathways in SGB009, SGB034, and SGB080. The figure is generated by Interactive Pathways Explorer (iPath) v3 (https://pathways.embl.de/). The line color is consistent with the color of the numbers in Figure (a).

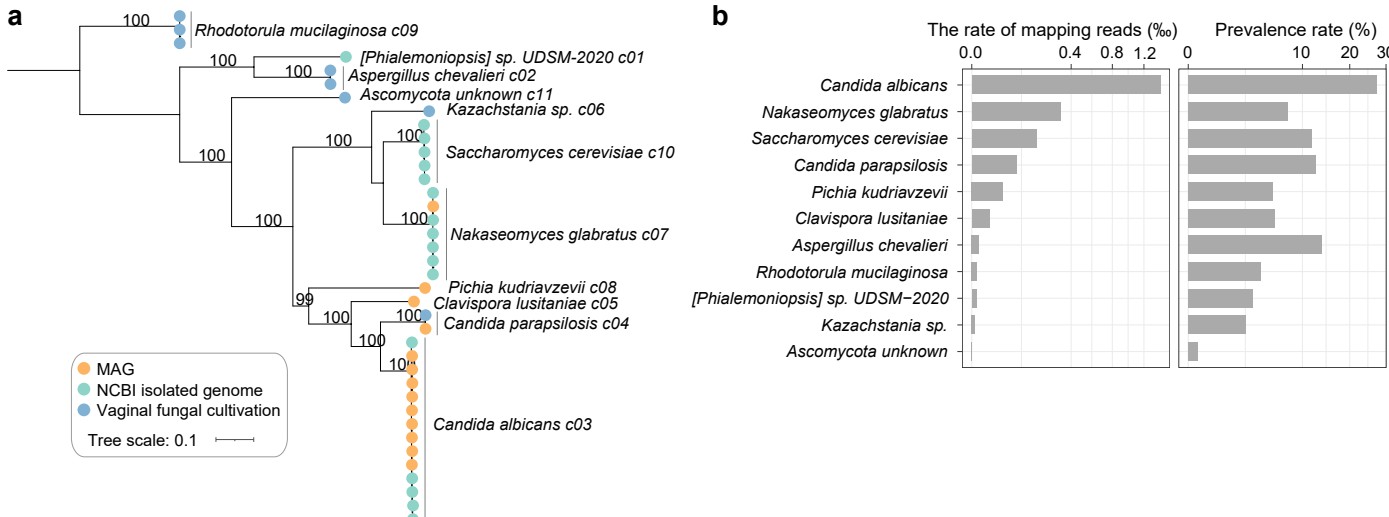

**Extended Data Fig. 6 | Characteristics of fungal populations in the VMGC.** **(a)** Phylogenetic tree illustrating the relationship between 38 fungal genomes based on their single-copy protein markers. The orange point represents the genome derived from the metagenomic binning algorithm, the green point represents the genome recorded in the NCBI genome database, and the blue point represents the genome cultivated by this study. **(b)** Mapping rates of reads and prevalence rates for 11 fungal species in the vaginal mycobiome.

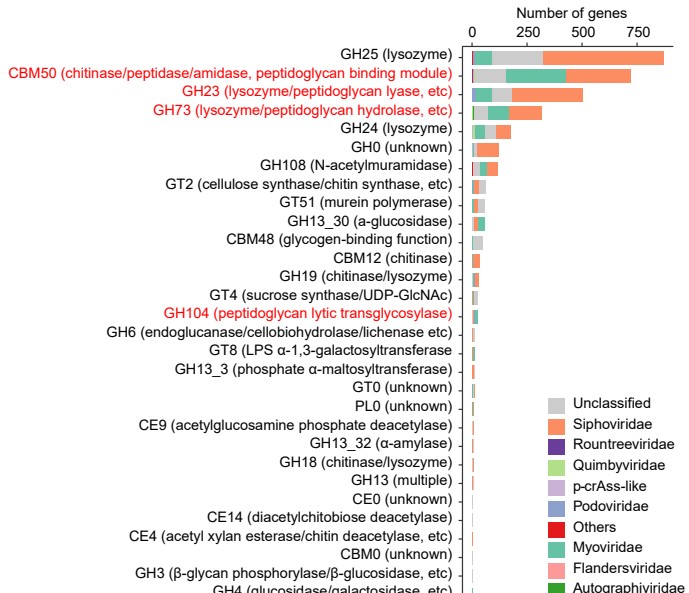

**Extended Data Fig. 7 | Distribution of carbohydrate-active enzymes (CAZymes) encoded by the vOTUs.** CBM, carbohydrate-binding module; CE, carbohydrate esterase; GH, glycoside hydrolase; GT, glycosyltransferase; PL, polysaccharide lyase.

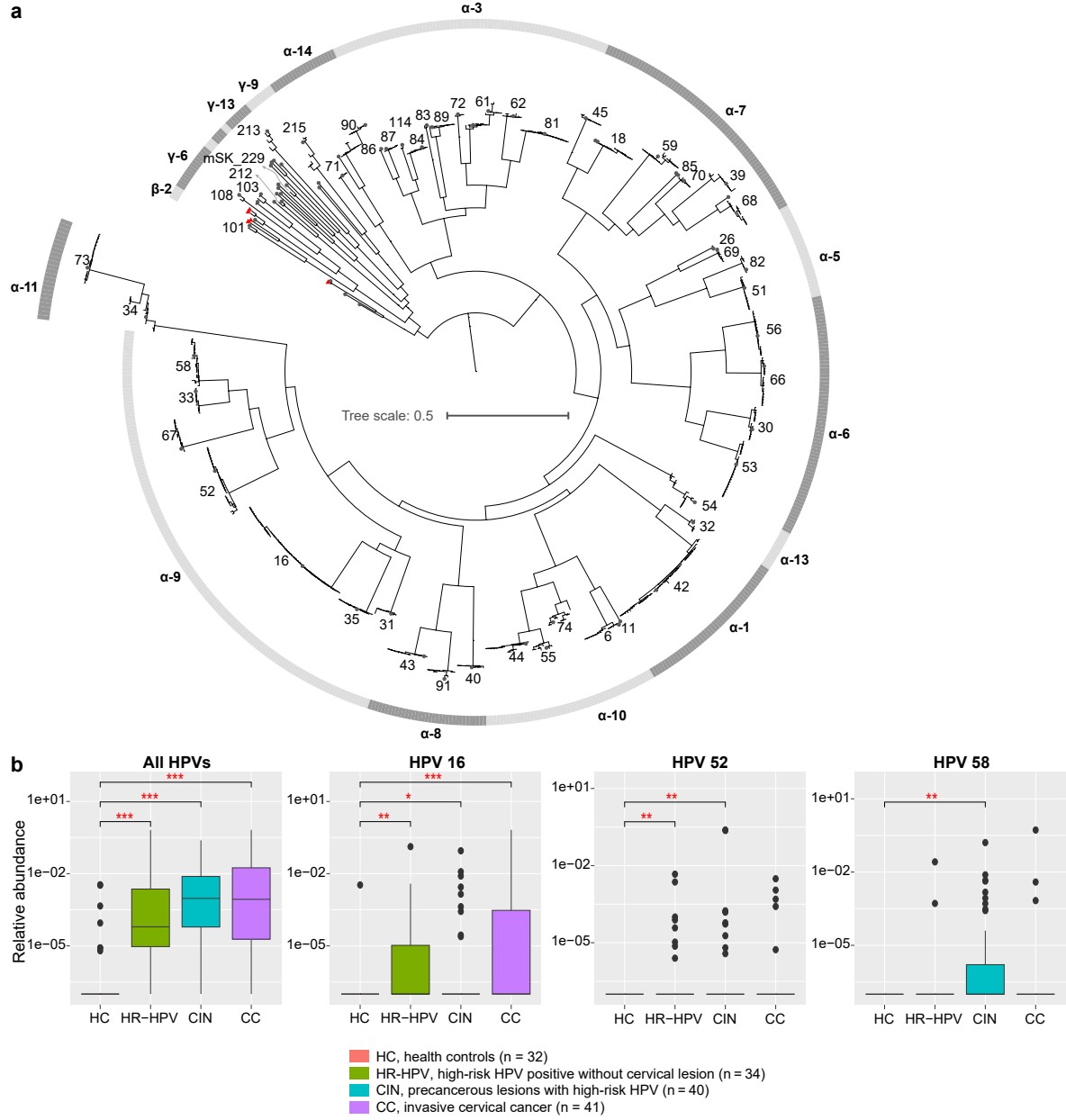

**Extended Data Fig. 8 | Phylogenetic and compositional analysis of Papillomaviridae. (a)** Phylogenetic tree constructed based on the L1 proteins of the Papillomaviridae genomes present in the VMGC and the NCBI RefSeq database. The outer ring shows the species-level classification of Papillomaviridae genomes. The numbers located at the ends of the branches correspond to the HPV types. Genomes obtained from the NCBI RefSeq database are depicted as grey points at the tips of the branches, while genomes from the VMGC that failed to classify into a specific HPV type are represented by red triangles. **(b)** Viral compositional analysis of vaginal metagenomes based on VMGC. Vaginal metagenomes were downloaded from the Liu et al.'s study [1],

and the compositional profiles of metagenomes were generated based on VMGC viral genomes. The comparison of HPV abundances is shown between health controls and cervical lesion patients. Boxplot showing the relative abundances of different groups. In the boxplot, the center line represents the median, the box limits show the upper and lower quartiles, the whiskers extend to 1.5 times the interquartile range, and points outside the whiskers are considered outliers. HC, health controls; HR-HPV, high-risk HPV positive without cervical lesion group; CIN, precancerous lesions with high-risk HPV group; CC, invasive cervical cancer group. Wilcoxon rank-sum test: *, $q < 0.05$; **, $q < 0.01$; ***, $q < 0.001$[92].

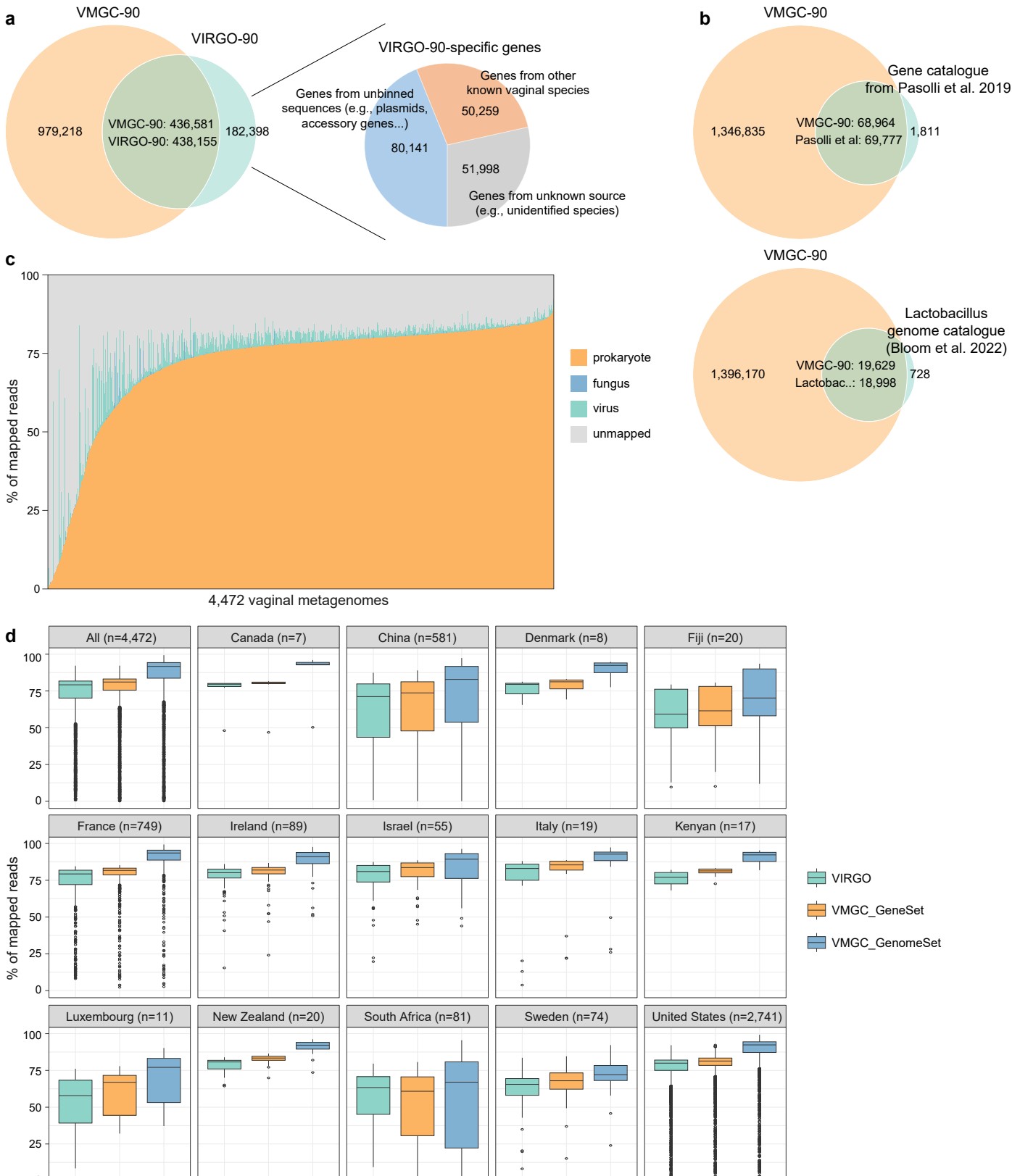

**Extended Data Fig. 9 | See next page for caption.**

**Extended Data Fig. 9 | Comparison of VMGC-90 and other vaginal gene catalogues. (a)** Source of the VIRGO-specific genes. **(b)** The overlap of proteins between the VMGC-90 and other vaginal microbial gene catalogues. Three non-redundant gene catalogues were constructed based on: 1) vaginal MAGs from t he Pasolli et al.'s study [1] (left panel), and 2) a recently published *Lactobacillus* genomic catalogue that included 1,091 previously unreported isolate genomes, partial genomes, and metagenome-assembled genomes (MAGs) [2] (right panel). These gene catalogues were constructed using the same methodology and parameters as VMGC-90. **(c)** Mapping rate of the vaginal microbial gene catalogue across all investigated samples. **(d)** Boxplot showing the mapping rates of VIRGO and VMGC genes and genomes across all samples. Samples are grouped by their countries. In the boxplot, the center line represents the median, the box limits show the upper and lower quartiles, the whiskers extend to 1.5 times the interquartile range, and points outside the whiskers are considered outliers[16,93].

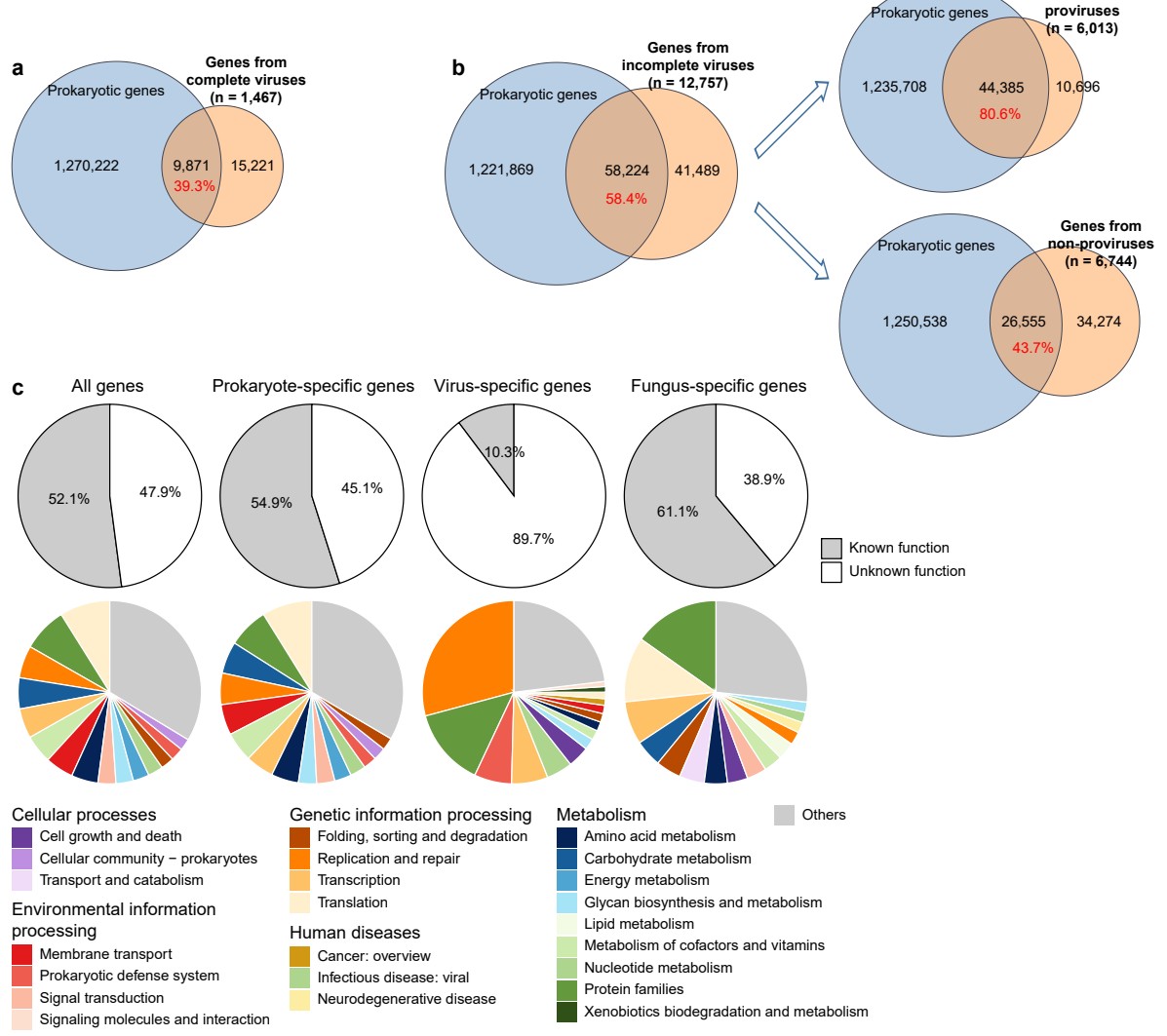

**Extended Data Fig. 10 | Comparison of VMGC-90 genes across multiple kingdoms. (a-b)** The overlap of prokaryotic and viral genes in the VMGC-90. Venn plot showing the comparisons of prokaryotic genes and genes from completeness **(a)** and incomplete viruses **(b)**. Red numbers show the percentages of viral genes covered by prokaryotic genes for each comparison. **(c)** The KEGG annotation of all microbial genes in the VMGC. Upper panels, proportions of the annotated genes; bottle panels, pie plot showing the proportions of pathways of the annotated genes at the KEGG level B.

# Reporting Summary

## Statistics

For all statistical analyses, confirm that the following items are present in the figure legend, table legend, main text, or Methods section.

| n/a | Confirmed | |
|---|---|---|
| ☐ | ☒ | The exact sample size (*n*) for each experimental group/condition, given as a discrete number and unit of measurement |
| ☐ | ☒ | A statement on whether measurements were taken from distinct samples or whether the same sample was measured repeatedly |
| ☐ | ☒ | The statistical test(s) used AND whether they are one- or two-sided *Only common tests should be described solely by name; describe more complex techniques in the Methods section.* |
| ☐ | ☒ | A description of all covariates tested |
| ☐ | ☒ | A description of any assumptions or corrections, such as tests of normality and adjustment for multiple comparisons |
| ☐ | ☒ | A full description of the statistical parameters including central tendency (e.g. means) or other basic estimates (e.g. regression coefficient) AND variation (e.g. standard deviation) or associated estimates of uncertainty (e.g. confidence intervals) |
| ☐ | ☒ | For null hypothesis testing, the test statistic (e.g. *F*, *t*, *r*) with confidence intervals, effect sizes, degrees of freedom and *P* value noted *Give P values as exact values whenever suitable.* |
| ☒ | ☐ | For Bayesian analysis, information on the choice of priors and Markov chain Monte Carlo settings |
| ☒ | ☐ | For hierarchical and complex designs, identification of the appropriate level for tests and full reporting of outcomes |
| ☐ | ☒ | Estimates of effect sizes (e.g. Cohen's *d*, Pearson's *r*), indicating how they were calculated |

*Our web collection on statistics for biologists contains articles on many of the points above.*

## Software and code

Policy information about availability of computer code

Data collection | No software was used for data collection.

Data analysis | Softwares/tools for data analysis: CheckM2 v1.0.1, CheckM2 v1.0.1, CheckM v1.1.3, GUNC v1.0.5, BUSCO v5.4.2, fastp v0.20.1, BBTools v39.00, Bowtie2 v2.4.1, METAHIT v1.2.9, Mash v1.1, dRep v3.4.0, cmsearch, INFERNAL v1.1.4, tRNAscan-SE v2.0.11, GTDB-Tk v1.4.0, iTOL v6.7.4 (https://itol.embl.de/), Kraken v2.1.3, Bracken v2.8, Prodigal v2.6.3, diamond v2.0.13.151, PhyloPhlAn v0.99, QIIME v2021.2.0, GeneMark-ES v4.68_lic, MMseqs2 v12.113e3, MAFFT v7.475, IQ-TREE v2.1.2, CheckV v0.7.0, DeepVirFinder v1.0, VIBRANT v1.2.1, hmmsearch v3.3.2, BLASTn v2.12.0, MinCED v0.4.2.

The custom analysis and visualization codes used in this study have been uploaded into the GitHub repository, accessible at: https://github.com/RChGO/VMGC.

For manuscripts utilizing custom algorithms or software that are central to the research but not yet described in published literature, software must be made available to editors and reviewers. We strongly encourage code deposition in a community repository (e.g. GitHub). See the Nature Portfolio guidelines for submitting code & software for further information.

## Data

Policy information about availability of data

All manuscripts must include a data availability statement. This statement should provide the following information, where applicable:

- Accession codes, unique identifiers, or web links for publicly available datasets
- A description of any restrictions on data availability
- For clinical datasets or third party data, please ensure that the statement adheres to our policy

Publicly available datasets used in this study: NCBI genome database (https://www.ncbi.nlm.nih.gov/genome/browse/), NCBI BioSample (https://ftp.ncbi.nlm.nih.gov/biosample/), GOLD (https://gold.jgi.doe.gov/), VIRGO website (https://virgo.igs.umaryland.edu/), fungi_odb10.2019-11-20, CHM13v2.0, GTDB database r214.1, KEGG (Kyoto Encyclopedia of Genes and Genomes), CAZy (Carbohydrate-Active EnZyme), VFDB (Virulence Factor Database), UNITE fungal ITS database version 8.3, SILVA database, Gut Virome Database (GVD), Gut Phage Database (GPD), Metagenomic Gut Virus catalogue (MGV), Oral Virus Database (OVD).

The data files of VMGC, including prokaryotic, eukaryotic, and viral genome sequences, annotation files, and the updated Kraken database, have been deposited in the Zenodo repository with the accession ID 10457006 (https://zenodo.org/records/10457006). The assembled genomes of the cultivated fungal strains were deposited in the NCBI database with BioProject accession ID PRJNA1100704.
The metadata, intermediate results, and analysis and visualization codes used in this study have been uploaded into the GitHub repository, accessible at: https://github.com/RChGO/VMGC.

## Research involving human participants, their data, or biological material

Policy information about studies with human participants or human data. See also policy information about sex, gender (identity/presentation), and sexual orientation and race, ethnicity and racism.

| | |
|---|---|
| Reporting on sex and gender | Fungal cultivation was performed based on fresh specimens of genital tract secretions from three healthy women: Subject 1 (21 years old), Subject 2 (36 years old), and Subject 3 (47 years old). |
| Reporting on race, ethnicity, or other socially relevant groupings | We collected a total of 4,472 publicly available vaginal metagenomic samples sourced from the human vagina, spanning 32 studies across the USA (n = 2,741 samples), France (749 samples), China (581 samples), and 11 other transcontinental countries. The phenotypic characteristics of the publicly metagenomic datasets were described in their respective studies. |
| Population characteristics | The population characteristics of the publicly available metagenomic datasets were described in their respective studies. Fungal cultivation was performed based on fresh specimens from three healthy women: Subject 1 (21 years old), Subject 2 (36 years old), and Subject 3 (47 years old). |
| Recruitment | For fungal cultivation, three healthy women were randomly recruited from the Dalian Medical University, and no compensation was paid to them. Informed consent was obtained from all volunteers. |
| Ethics oversight | The study received approval from the Ethics Committee of the Second Affiliated Hospital of Dalian Medical University (No. DMU20210082). Informed consent was obtained from all volunteers. |

Note that full information on the approval of the study protocol must also be provided in the manuscript.

# Field-specific reporting

Please select the one below that is the best fit for your research. If you are not sure, read the appropriate sections before making your selection.

☒ Life sciences  ☐ Behavioural & social sciences  ☐ Ecological, evolutionary & environmental sciences

For a reference copy of the document with all sections, see nature.com/documents/nr-reporting-summary-flat.pdf

# Life sciences study design

All studies must disclose on these points even when the disclosure is negative.

| | |
|---|---|
| Sample size | We performed an extensive search in the NCBI database until October 2023, targeting human vaginal metagenomic samples annotated as vagin*, cervi*, endomet*, etc. The search yielded a collection of 3,123 vaginal metagenomes from 32 different studies. We obtained additional 1,366 vaginal metagenomic samples from the VIRGO website (https://virgo.igs.umaryland.edu/), after removing duplicates found in the NCBI samples. No statistical method was used to predetermine sample size, but the sample size and amount of metagenomic sequence data are larger than currently published studies of similar nature. |
| Data exclusions | No metagenomic data was excluded in the analysis. |
| Replication | The analysis can be reproduced using the data and software described in the Methods section, since the metagenomic sequencing data and the genome resources are accessible to the public. |
| Randomization | Randomization is not applicable as this study did not involve population trials. |

| Blinding | Blinding is not applicable as this study did not involve population trials. |

# Reporting for specific materials, systems and methods

We require information from authors about some types of materials, experimental systems and methods used in many studies. Here, indicate whether each material, system or method listed is relevant to your study. If you are not sure if a list item applies to your research, read the appropriate section before selecting a response.

### Materials & experimental systems

| n/a | Involved in the study |
|-----|------------------------|
| ☒ ☐ | Antibodies |
| ☒ ☐ | Eukaryotic cell lines |
| ☒ ☐ | Palaeontology and archaeology |
| ☒ ☐ | Animals and other organisms |
| ☒ ☐ | Clinical data |
| ☒ ☐ | Dual use research of concern |
| ☒ ☐ | Plants |

### Methods

| n/a | Involved in the study |
|-----|------------------------|
| ☒ ☐ | ChIP-seq |
| ☒ ☐ | Flow cytometry |
| ☒ ☐ | MRI-based neuroimaging |

## Plants

| Seed stocks | Not applicable |

| Novel plant genotypes | Not applicable |

| Authentication | Not applicable |

