## [Peer Review File · Nature Microbiology]

Peer Review Information

Journal: Nature Microbiology

Manuscript Title: A multi-kingdom collection of 33,804 reference genomes for the human vaginal microbiome

Corresponding author name(s): Professor Wen Sun

Reviewer Comments & Decisions:

Decision Letter, initial version:

Message: 15th September 2023

Dear Professor Sun,

Thank you for your patience while your manuscript "A multi-kingdom collection of reference genomes in the human vaginal microbiome" was under peer-review at Nature Microbiology. It has now been seen by 3 referees, whose expertise and comments you will find at the end of this email. Although they find your work of some potential interest, they have raised a number of concerns that will need to be addressed before we can consider publication of the work in Nature Microbiology.

In particular, all referees have some technical concerns that will need to be addressed including additional analyses and validation of the genomes, benchmarking for atypical binning, etc. Please ensure that code and data are available to referees in a meaningful way. We will not require you to add new biological insights given the resource value of this study, but we will require you to address all other concerns in full.

Should further experimental data allow you to address these criticisms, we would be happy to look at a revised manuscript.

Please include a data availability statement as a separate section after Methods but before references, under the heading "Data Availability". This section should inform readers about the availability of the data used to support the conclusions of your study. This information

2includes accession codes to public repositories (data banks for protein, DNA or RNA sequences, microarray, proteomics data etc...), references to source data published alongside the paper, unique identifiers such as URLs to data repository entries, or data set DOIs, and any other statement about data availability. At a minimum, you should include the following statement: "The data that support the findings of this study are available from the corresponding author upon request", mentioning any restrictions on availability. If DOIs are provided, we also strongly encourage including these in the Reference list (authors, title, publisher (repository name), identifier, year). For more guidance on how to write this section please see:
<http://www.nature.com/authors/policies/data/data-availability-statements-data-citations.pdf>

* If you have not done so already we suggest that you begin to revise your manuscript so that it conforms to our Article format instructions at <http://www.nature.com/nmicrobiol/info/final-submission>. Refer also to any guidelines provided in this letter.

When submitting the revised version of your manuscript, please pay close attention to our [href="https://www.nature.com/nature-portfolio/editorial-policies/image-integrity">Digital Image Integrity Guidelines](https://www.nature.com/nature-portfolio/editorial-policies/image-integrity). and to the following points below:

2Note: This url links to your confidential homepage and associated information about manuscripts you may have submitted or be reviewing for us. If you wish to forward this e-mail to co-authors, please delete this link to your homepage first.

Nature Microbiology is committed to improving transparency in authorship. As part of our efforts in this direction, we are now requesting that all authors identified as 'corresponding author' on published papers create and link their Open Researcher and Contributor Identifier (ORCID) with their account on the Manuscript Tracking System (MTS), prior to acceptance. This applies to primary research papers only. ORCID helps the scientific community achieve unambiguous attribution of all scholarly contributions. You can create and link your ORCID from the home page of the MTS by clicking on 'Modify my Springer Nature account'. For more information please visit please visit www.springernature.com/orcid.

If you wish to submit a suitably revised manuscript we would hope to receive it within 6 months. If you cannot send it within this time, please let us know. We will be happy to consider your revision, even if a similar study has been accepted for publication at Nature Microbiology or published elsewhere (up to a maximum of 6 months).

Yours sincerely,

Reviewer Expertise:

Referee #1: microbiome, genome-resolved metagenomics

Referee #2: microbiome, genome-resolved metagenomics

Referee #3: vaginal microbiome, omics

Reviewer Comments:

Reviewer #1 (Remarks to the Author):

The manuscript from Huang et al. describes the creation of a dedicated genomic resource for the vaginal microbiome, named VMGC. This catalog, similarly to the Skin Microbial Genome Catalog, was derived from both cultured and metagenomically-derived genomes and includes representative genomes from all domains of life. This represents a very valuable collection of genomes as the authors have processed over 3,000 metagenomic samples from 26 studies, capturing a wide range of datasets across the world. The authors have also done a good job in using the most up-to-date tools and standards in the field to generate a high-quality collection of genomes from an understudied source. In addition, they analyse specific subsets of their catalog (e.g., the order Saccharofermentales) to obtain some additional biological insights into the vaginal microbiome.

3Overall I find the work of high quality and only have some minor comments and suggestions:

1) The completeness of certain taxa such as Saccharofermentales is underestimated by current tools due to their natural absence of certain housekeeping genes. Given the newest version of CheckM (v2) is now out (<https://www.nature.com/articles/s41592-023-01940-w>) and is expected to provide better quality estimates for understudied clades, I suggest the authors to assess if this tool would perform better for their specific use case.

2) The authors combined raw MAGs based on similarities in sequencing depth and GC content. Given this additional step is not common practice in the field I am a bit skeptical about the possibility of introducing contamination and the risk of combining unrelated MAGs. With this in mind the authors should present some benchmarking analyses validating their approach and justifying their choice of thresholds/parameters (e.g, sequencing depth +/- 10% and GC content +/- 2%).

3) Related with the above point, the authors used SpecI to estimate which MAGs would belong to the same species. Given this tool was published in 2013, before advances such as GTDB were developed, my feeling is that it may be difficult to assign a lot of the MAGs to a recognized/named species. What proportion of the MAGs did the authors manage to assign at the species level with SpecI before combining them?

4) MAGs can be very prone to contamination, so strictly relying on marker gene-based tools such as CheckM may not be sufficient. The authors should consider further confirming the quality of the MAGs with another tool such as GUNC (<https://genomebiology.biomedcentral.com/articles/10.1186/s13059-021-02393-0>)

5) For the viral genome filtering the authors state "we employed one or more of the following criteria to detect possible virus sequences". This vague wording suggests only certain criteria were applied to specific viruses. The authors should clearly state what was the minimum quality criteria for all viral predictions included.

6) Why was a threshold of 96% ANI used for defining a fungal species? The authors should provide some justification why this is an accepted species boundary and how robust this threshold is (i.e., what difference would using 95% or 97% ANI make).

Reviewer #3 (Remarks to the Author):

In this study, Huang and colleague introduce the Vaginal Microbiome Genome Collections, a compilation of prokaryotic, fungal, and viral MAGs and isolate genomes. I have focused on the metagenomic aspects of this study and have left comments on this studies contribution to the vaginal microbiome literature to reviewers more familiar with this topic. Overall, I found the manuscript a pleasure to read though it would benefit from further copy editing. I found the figures to be of extremely high quality and to do an excellent job of summarizing the salient properties of the VMGC and the analyses performed in this study. In terms of methodology, the manuscript meets current best practice and I have only

4

relatively minor questions and suggestions.

- The manuscript does not follow the MIMAG standard which classifies a MAG as high quality only if it is >90% complete with <5% contamination and has the 5S, 16S, and 23S rRNA genes along with 18 or more tRNAs. The definitions used for near complete, high quality, and medium quality in this manuscript (as summarized in Figure 1B) are perfectly reasonable, but the manuscript should be revised to make it clear that the adopted definitions are based on the MIMAG standard, but differ in their exact definition.
- The reported systematic underestimation of the completeness of Saccharofermentanales genomes by CheckM v1 and BUSCO is interesting. CheckM v2 (PMID: 37500759) has recently been released which uses a machine learning approach to estimate genome quality that is not based on the presence or absence of marker genes. It would be an aid to the larger scientific community if CheckM v2 could be applied to the Saccharofermentanales genomes to evaluate if it also suffers from systematically underestimating the completeness of these genomes or successfully address this limitation of CheckM v1 and BUSCO.
- Reporting on the taxonomic profiling of the ~3,000 vaginal metagenomic samples would benefit from considering the percentage of reads that remain unclassified. This is done in Figure 8C and shows that for some samples a substantial portion (>40%) of reads could not be assigned to the VMGC. I believe the results provided in Figure 2D would be more insightful if they considered the portion of unclassified read. Alternatively, I would suggest this be provided as a supplemental figure so readers can evaluate how the percentage of unclassified reads changes across the ~3,000 vaginal metagenomes.
- The methods indicate that prokaryotic MAGs with similar sequencing depth, GC content, and taxonomic assignments were merged. Were all MAGs satisfying the specified criteria merged? My concern is that merging MAGs in this way might result in a chimeric MAG with high contamination in some cases. Were such chimeras observed, as evident from greatly increased CheckM or BUSCO contamination estimates, and subsequently left as individual MAGs?
- What bowtie2 parameters were used to map reads to genome in the VMGC for the purposes of taxonomic profiling?
- I believe the statement "Vaginal samples with a relatively low proportion of potential microbial content (>400,000 clean reads) were eliminated before generating taxonomic profiles" should indicate <400,000 reads. How many of the samples failed this criterion?
- The methods section on the genomic analysis of Saccharofermentanales members references Supplementary Figure 4 as indicating the 14 marker genes that were largely absent across these genomes. Supplementary Figure 4 does not contain this information. Perhaps this was meant to reference a specific supplementary table? Are the 14 absent markers the same for SGB006, SGB028, and SGB069? As currently stated, it is not clear that this is the case.

- Why were fungal species delineated at 96% ANI while prokaryotic and viral species were delineated at 95% ANI?
- For the identification of virus sequences, the methods indicate that contigs with “fivefold the number of viral genes” were removed. Should this indicate that contigs with less than fivefold the number of viral genes were removed?
- Why was 75% alignment fraction used to delineated vOTUs? Is this the criterion used in previous studies?

Reviewer #4 (Remarks to the Author):

Generation of metagenomics in the vaginal microbiome has generally lagged behind other organ systems, and, consequently, so has the generation of large-scale reference datasets. Huang et al. attempt to address this by integrating available metagenomic data, reference microbial genomes, and very modest in-house cultivation of vaginal fungi to construct a comprehensive database of microbial genomes from the vaginal environment. This study is timely and addresses a clear gap in the field. Data acquisition efforts are extensive, resulting in over 3400 metagenomic samples from across multiple locations around the world. This reference dataset could be useful to the field, and includes elusive genomes such as that of BVAB. However - the analyses provided are highly descriptive, and in my opinion do not present particularly novel insights; I have significant methodological concerns that make me significantly doubt the validity of this database; there is little to no validation provided to support the validity of the work; and the data and code are not available in a meaningfully useful way.

1. Recent work has demonstrated substantial human contamination caused by the use of incomplete human reference genome, such as GRCh38 (Gihawi et al., bioRxiv 2023). The authors should switch to CHM13 with the most recent Y chromosome sequence and also detail the parameters used by bowtie for full replication.
2. Many newer binning methods have been published recently, including VAMB, MetaCOAG, SemiBin2. Additionally, recent work established that multi-binning should be used instead of single-binning (Mattock & Watson 2023). These advances have not been applied in this work. Can the authors validate and or justify their methodological pipeline?
3. The authors add a custom binning step – combining MAGs if they show a close sequencing depth and GC content. What is the justification for it? What validation is offered to this? Why is this step not already performed by binning? On the face of it, this step seems methodologically invalid, and while it inflates completeness of the MAGs, recent studies (Mattock & Watson 2023) show that this likely creates chimeras.
4. The method used by the authors to generate abundances is very sensitive to coverage. I believe some sort of subsampling or accounting for coverage with a relevant statistical model is necessary.
5. Estimating abundance from sequencing coverage (P13, L22-23) is fraught with issues. In generating taxonomic profiles, the authors should at the very least examine the coverage uniformity. If only a small segment of the genome is covered, it should not get any abundance at all.
6. Given the high human read % in vaginal samples (~90-95%), eliminating samples with

6<400K reads (P13, L20) corresponds to eliminating samples with less than 8M reads and will generate a bias towards samples with high bacterial fraction (potentially, high load?). This seems like an unreasonable decision.

7. Regarding missingness of USCOs (P6) – first, the authors do not show an analysis to support that these are the SAME USCOs that are missing across all MAGs – just that it's the same number. Second, the authors would need to rule out the presence of those missing genes in the un-binned fraction of the contigs or sequence reads. It's possible that those genes happen to be placed next to a variable region in the genome, which are known to not be binned properly.

8. Identification of fungal MAGs (P14, L24) is unclear and seems unvalidated and potentially non-specific. Taxonomic similarity of 80% to what? Specl was developed for prokaryotes – is it even valid for fungi? How was this validated?

9. P7, L36: 95% ANI threshold, while heavily contested in prokaryotes, is at least based on a data-driven analysis. Is there any analysis to support the use of this threshold in viruses?

10. How many phages are integrated? Are they even real phages? For how many of these were closed (circular) genomes assembled?

11. How many of these species actually come from the vaginal environments? Many of the bacterial species found (177) had been previously identified only in other environments (from other body sites mostly). Could these be contamination? What is their prevalence across the metagenomic samples? How is contamination controlled for?

12. Extensive validations are necessary here. Some suggestions could be comparison to 16S; examination of CSTs; comparisons to isolates; comparisons to other studies. In that regard, the comparison to VIRGO (P9, L33-34) is VERY alarming. With the exact same samples, there is only 58% coverage of the VIRGO dataset? If the authors have a potential explanation to this, such as L35, they should prove that this is the case.

13. How does figure 8c compare to VIRGO?

14. Are the HPV viruses assembled enriched in patients with HPV in the various studies?

15. Data and code availability is unacceptable, especially for a resource paper: (a) code should be clear, well document, and enable full replication. A clear documentation of the process and how each code file relates to it is necessary. (b) The files themselves should be deposited in an independent repository (i.e., Zenodo). (c) clear documentation of the files, what is in each one, what are the variables, etc., should be provided.

16. The introduction has a lot of information that is not really relevant to the work, e.g. the first paragraph.

17. The authors show that viruses do not fully saturate in this dataset. How does it look like for bacteria and fungi? Are 658 vaginal species across all of these samples a reasonable number? Also, why are there so few eukaryotic viruses?

18. The section "Functional configuration of vaginal prokaryotic species" suffers from extreme overinterpretations. The authors show some gene annotations. They do not show that any microbe is a "producer", "reservoir", "synthesizer" or "play a role" in anything.

19. The focus of the 9 modules (P5, last paragraph) is not justified. Supp table 7 lists some references, some from the gut.

20. P13, "Identification of BVAB2 genomes" It's great that 16 of 18 are classified as BVAB2. How many genomes outside this cluster were classified as BVAB2?

21. Page 5, L14-15 – this findings suggest novelty of species, but say nothing about importance.

22. Figure 4d – what is the meaning of the two colors in the inner ring of the genome representation

23. P3, L39 – how can *T. vaginalis*, with completeness of 38.4-47.8%, and *T. parva* with 74%, be considered “highly credible”?
24. It is not clear what is “prevalence rate” in Figure 5.

Author Rebuttal to Initial comments

Reviewer Comments:

Reviewer #1 (Remarks to the Author):

The manuscript from Huang et al. describes the creation of a dedicated genomic resource for the vaginal microbiome, named VMGC. This catalog, similarly to the Skin Microbial Genome Catalog, was derived from both cultured and metagenomically-derived genomes and includes representative genomes from all domains of life. This represents a very valuable collection of genomes as the authors have processed over 3,000 metagenomic samples from 26 studies, capturing a wide range of datasets across the world. The authors have also done a good job in using the most up-to-date tools and standards in the field to generate a high-quality collection of genomes from an understudied source. In addition, they analyse specific subsets of their catalog (e.g., the order Saccharofermentanales) to obtain some additional biological insights into the vaginal microbiome.

Overall I find the work of high quality and only have some minor comments and suggestions:

Response: We sincerely appreciate the encouragement and insightful comments from the reviewer. Please see the detailed point-by-point breakdown below for responses to each of your concerns.

1) The completeness of certain taxa such as Saccharofermentanales is underestimated by current tools due to their natural absence of certain housekeeping genes. Given the newest version of CheckM (v2) is now out (<https://www.nature.com/articles/s41592-023-01940-w>) and is expected to provide better quality estimates for understudied clades, I suggest the authors to assess if this tool would perform better for their specific use case.

Response: Thanks very much for this valuable comment. Following your suggestion, we utilized CheckM v2 to assess the quality of all prokaryotic MAGs in the revised manuscript. The MAG selection criteria were also based on the quality assessment results from CheckM v2. As noted by the reviewer, CheckM v2 indeed generated a higher completeness estimate for Saccharofermentanales MAGs in comparison to CheckM v1, with enhanced average completeness from 82.4% to 87.6% (among all 742 MAGs). Notably, CheckM v2 demonstrated a remarkable completeness assessment for BVAB3 (*Mageeibacillus indolicus*, also SGB034 in our revised manuscript), reaching the highest value of 100% (average 89.3% for

8173 MAGs). This high accuracy may be attributed to the inclusion of the complete genome of BVAB3 in its reference database (NCBI accession ID: GCF_000025225.2). For the other two major Saccharofermentales species, SGB009 (ala. BVAB2) and SGB080 (a BVAB2-like species), their completeness evaluations achieved a maximum of 96.5% (average 90.1% for 473 MAGs) and 96.4% (average 84.9% for 36 MAGs), respectively. However, this completeness was still underestimation, as two complete genomes of SGB009 and SGB090 (NCBI accession IDs GCF_029101565.1 and GCF_029101585.1, respectively, both represented as single circular no-gapped contig from a recently published study [1]) only resulted in a completeness value of 94.1% for both. Despite this, within our SGB009 and SGB080, there are still 13 and 3 near-complete genomes (>90% completeness, <5% contamination, and presence of full-length rRNA genes and ≥ 18 tRNAs), respectively, along with a substantial number of high-quality genomes ($n = 360$ and 14 , respectively, >90% completeness and <5% contamination). These genomic resources contribute significantly to advancing research on these bacteria, facilitating studies such as evolutionary and functional analyses.

References:

[1] Srinivasan S, Austin M N, Fiedler T L, et al. *Amygdalobacter indicium* gen. nov., sp. nov., and *Amygdalobacter nucleatus* sp. nov., gen. nov.: novel bacteria from the family *Oscillospiraceae* isolated from the female genital tract. *International journal of systematic and evolutionary microbiology*, 2023, 73(10): 006017.

2) The authors combined raw MAGs based on similarities in sequencing depth and GC content. Given this additional step is not common practice in the field I am a bit skeptical about the possibility of introducing contamination and the risk of combining unrelated MAGs. With this in mind the authors should present some benchmarking analyses validating their approach and justifying their choice of thresholds/parameters (e.g, sequencing depth +/- 10% and GC content +/- 2%).

Response: Thank you very much for this insightful and professional comment. Based on your suggestions and those of other reviewers, we have discontinued the practice of combining raw MAGs in the revised manuscript, to prevent potential contamination and inaccuracies. Specifically, based on the suggestions from Reviewer #3 and in light of recent methodological publications [1-2], we have implemented an updated approach that utilizes both single-coverage binning and mash-based multi-coverage binning processes for the assembled contigs to recover MAGs from metagenomic samples. This revised approach has demonstrated a significantly enhanced effectiveness in MAG reconstruction

compared to the singular single-sample approach employed in our original manuscript. For a comprehensive description of these modifications, please refer to our response to Comment #2 raised by Reviewer #3.

References:

[1] Mattock J, Watson M. A comparison of single-coverage and multi-coverage metagenomic binning reveals extensive hidden contamination. *Nature methods*, 2023, 20(8): 1170-1173.

[2] Carter M M, Olm M R, Merrill B D, et al. Ultra-deep sequencing of Hadza hunter-gatherers recovers vanishing gut microbes. *Cell*, 2023.

3) Related with the above point, the authors used Specl to estimate which MAGs would belong to the same species. Given this tool was published in 2013, before advances such as GTDB were developed, my feeling is that it may be difficult to assign a lot of the MAGs to a recognized/named species. What proportion of the MAGs did the authors manage to assign at the species level with Specl before combining them?

Response: Thanks for pointing this out. As mentioned in the above response, we have ceased the practice of combining raw MAGs, and the use of the Specl tool has been discontinued throughout our revised manuscript. Furthermore, we noted substantial genomic and taxonomic expansion in the database with recent GTDB versions (<https://gtdb.ecogenomic.org/>). Consequently, in the revised manuscript, we have transitioned from GTDB version release 202 (with 31,910 genomes; <https://data.gtdb.ecogenomic.org/releases/release202/202.0/>) to the most recent version, release 214.1 (with 85,205 genomes; <https://data.gtdb.ecogenomic.org/releases/release214/214.1/>). These updates are aimed at ensuring the latest and most comprehensive genomic information is utilized in our analysis.

4) MAGs can be very prone to contamination, so strictly relying on marker gene-based tools such as CheckM may not be sufficient. The authors should consider further confirming the quality of the MAGs with another tool such as GUNC (<https://genomebiology.biomedcentral.com/articles/10.1186/s13059-021-02393-0>)

Response: Thank you very much for this helpful comment. In the revised manuscript, based on your

suggestions, we applied quality filtering to all raw MAGs using both CheckM v2 and GUNC tools. The combination of single-coverage binning and mash-based multi-coverage binning methods yielded a total of 63,654 MAGs (>200 kbp). Among these, 33,213 MAGs were excluded due to CheckM v2 completeness being less than 50%, or contamination exceeding 5%, or a quality score ($\% \text{ completeness} - \% \text{ contamination} * 5$) less than 50. Among the remaining MAGs, 2,495 were removed because of a GUNC clade separation score (CSS) greater than 0.45. A total of 27,946 MAGs passed the aforementioned filtering criteria. These 27,946 MAGs were subsequently inter-sample dereplicated at the strain level (99% ANI), resulting in the generation of 18,570 prokaryotic MAGs in the VMGC.

5) For the viral genome filtering the authors state “we employed one or more of the following criteria to detect possible virus sequences”. This vague wording suggests only certain criteria were applied to specific viruses. The authors should clearly state what was the minimum quality criteria for all viral predictions included.

Response: Sincerely apologize for the confusion caused by our imprecise expression. In practice, during the virus identification process, our requirement was that a contig only needs to meet the criteria of any one of the three virus identification tools (e.g., CheckV, DeepVirFinder, and VIBRANT) to be classified as a potential viral sequence. Importantly, based on the identification results, (see supporting figure 1 below), we found that 24.8% of viruses were discovered by all three tools, and 89.4% of viruses were identified by no fewer than two tools, suggesting that these tools exhibited a high degree of consistency in confirming viral sequences. In the revised manuscript, we have amended the description of this process as follows:

*"For the remaining contigs, contigs meeting **any of the following criteria** are considered possible viral sequences: 1) a higher number of viral genes compared to host genes according to CheckV v0.7.0; 2) detection as viruses using the thresholds of score >0.90 and p-value <0.01 in DeepVirFinder v1.0; and 3) identification as viruses by VIBRANT v1.2.1 using default parameters."*

6) Why was a threshold of 96% ANI used for defining a fungal species? The authors should provide some justification why this is an accepted species boundary and how robust this threshold is (i.e., what difference would using 95% or 97% ANI make).

Response: Thank you very much for your valuable insights. Based on your suggestion, in the revised manuscript, we conducted a parameter exploration for defining fungal species boundaries. We utilized over 11,000 available fungal genomes from the NCBI database (downloaded in June 2023), performed pairwise average nucleotide identity (ANI) calculations for them, and conducted analyses based on known species assignments. When setting the ANI threshold at 96%, we observed that 98.76% of intra-species pairs met or exceeded this value, with only 0.14% of inter-species pairs achieving this threshold (equivalent to false positives) (see supporting figure 2 below). Similarly, at an ANI threshold of 95%, 99.09% of intra-species pairs reached this criterion, while only 0.16% of inter-species pairs did. Therefore, setting the standards at both 96% and 95% for species definition proved to be highly accurate. Considering that an ANI of 95% is widely employed as a standard in prokaryotic and viral genome studies [1-2], and in accordance with your suggestion and that from other reviewers, we have adopted the 95% threshold as the definition for fungal species in the revised manuscript. This adjustment is aimed at aligning our work with established practices in related genomic research fields.

References:

- [1] Jain C, Rodriguez-R L M, Phillippy A M, et al. High throughput ANI analysis of 90K prokaryotic genomes reveals clear species boundaries. *Nature communications*, 2018, 9(1): 5114.
- [2] Gregory A C, Zayed A A, Conceição-Neto N, et al. Marine DNA viral macro- and microdiversity from pole to pole. *Cell*, 2019, 177(5): 1109-1123. e14.

Reviewer #2 (Remarks to the Author):

In this study, Huang and colleague introduce the Vaginal Microbiome Genome Collections, a compilation of prokaryotic, fungal, and viral MAGs and isolate genomes. I have focused on the metagenomic aspects of this study and have left comments on this studies contribution to the vaginal microbiome literature to reviewers more familiar with this topic. Overall, I found the manuscript a pleasure to read though it would benefit from further copy editing. I found the figures to be of extremely high quality and to do an excellent job of summarizing the salient properties of the VMGC and the analyses performed in this study. In terms of methodology, the manuscript meets current best practice and I have only relatively minor questions and suggestions.

Response: We sincerely appreciate the reviewer's encouragement and helpful comments. Please see below our point-by-point responses to his/her concerns.

• The manuscript does not follow the MIMAG standard which classifies a MAG as high quality only if it is >90% complete with <5% contamination and has the 5S, 16S, and 23S rRNA genes along with 18 or more tRNAs. The definitions used for near complete, high quality, and medium quality in this manuscript (as summarized in Figure 1B) are perfectly reasonable, but the manuscript should be revised to make it clear that the adopted definitions are based on the MIMAG standard, but differ in their exact definition.

Response: Thanks for this professional comment. As the reviewer pointed out, there are differences between our classification criteria for MAG quality and the MIMAG standard (see supporting table 1 below). In fact, we have observed that since the introduction of the MIMAG standard, numerous genomic studies have made appropriate adjustments by referencing this standard. Particularly in recent genomic studies, there has been a trend toward a more stringent classification standard (see supporting table 1 below). After careful consideration, we have not entirely adhered to the MIMAG standard for MAG classification. In the revised manuscript, we have modified the description of our grouping definitions and underscored the distinctions between our criteria and the MIMAG standard, as follows:

*"Among the 19,542 prokaryotic genomes (i.e., 18,570 MAGs and 972 isolated genomes), 10,127 (51.8%) and 8,397 (43.0%) were classified as medium- and high-quality draft genomes, respectively (Figure 1b), based on **the criteria revising from** the MIMAG (Minimum Information about a Metagenome-Assembled Genome) standard."*

Study	Near-complete	High-quality	Medium-quality
Bowers_2017 (MIMAG) [1]	single no-gapped contig ("finished")	>90% completeness & <5% contamination & 5S-16S-23S rRNA genes & ≥18 tRNAs	≥50% completeness & <10% contamination
Pasolli_2019 [2]	>90% completeness & <5% contamination & strain heterogeneity <0.5%	≥50% completeness & <10% contamination & strain heterogeneity <0.5%	
Almeida_2021 [3]	>90% completeness & <5% contamination & 5S-16S-23S rRNA genes & ≥18 tRNAs	>90% completeness & <5% contamination	≥50% completeness & <10% contamination
Zeng_2022 [4]	>90% completeness & <5% contamination & 5S-16S-23S rRNA genes & ≥18 tRNAs & CSS < 0.45	>50% completeness & <5% contamination & QS > 75 & CSS < 0.45	>50% completeness & <5% contamination & QS 50-70 & CSS < 0.45
Saheb_2022 [5]	>90% completeness & <5% contamination & 5S-16S-23S rRNA genes & ≥18 tRNAs & CSS < 0.45	>90% completeness & <5% contamination & CSS < 0.45	≥50% completeness & <10% contamination & CSS < 0.45
This study	≥90% completeness & <5% contamination & QS ≥50 & 5S-16S-23S rRNA genes & ≥18 tRNAs	≥90% completeness & <5% contamination & QS ≥50 & CSS < 0.45	≥50% completeness & <5% contamination & QS ≥50 & CSS < 0.45

	& CSS < 0.45		
--	--------------	--	--

Supporting table 1. Classification criteria for MAG quality used in this and recently published studies.

CSS, clade separation score. QS, quality score (% completeness – 5 * % contamination).

References:

- [1] Bowers R M, Kyrpides N C, Stepanauskas R, et al. Minimum information about a single amplified genome (MISAG) and a metagenome-assembled genome (MIMAG) of bacteria and archaea. *Nature biotechnology*, 2017, 35(8): 725-731.
- [2] Pasolli E, Asnicar F, Manara S, et al. Extensive unexplored human microbiome diversity revealed by over 150,000 genomes from metagenomes spanning age, geography, and lifestyle. *Cell*, 2019, 176(3): 649-662. e20.
- [3] Almeida A, Nayfach S, Boland M, et al. A unified catalog of 204,938 reference genomes from the human gut microbiome. *Nature biotechnology*, 2021, 39(1): 105-114.
- [4] Zeng S, Patangia D, Almeida A, et al. A compendium of 32,277 metagenome-assembled genomes and over 80 million genes from the early-life human gut microbiome. *Nature communications*, 2022, 13(1): 5139.
- [5] Saheb Kashaf S, Proctor D M, Deming C, et al. Integrating cultivation and metagenomics for a multi-kingdom view of skin microbiome diversity and functions. *Nature microbiology*, 2022, 7(1): 169-179.

• *The reported systematic underestimation of the completeness of Saccharofermentanales genomes by CheckM v1 and BUSCO is interesting. CheckM v2 (PMID: 37500759) has recently been released which uses a machine learning approach to estimate genome quality that is not based on the presence or absence of marker genes. It would be an aid to the larger scientific community if CheckM v2 could be applied to the Saccharofermentanales genomes to evaluate if it also suffers from systematically underestimating the completeness of these genomes or successfully address this limitation of CheckM v1 and BUSCO.*

Response: Thanks very much for this valuable comment. Following your suggestion, we utilized CheckM v2 to assess the quality of all prokaryotic MAGs in the revised manuscript. The MAG selection criteria were also based on the quality assessment results from CheckM v2. As noted by the reviewer, CheckM

16v2 indeed generated a higher completeness estimate for Saccharofermentales MAGs in comparison to CheckM v1, with enhanced average completeness from 82.4% to 87.6% (among all 742 MAGs). Notably, CheckM v2 demonstrated a remarkable completeness assessment for BVAB3 (*Mageeibacillus indolicus*, also SGB034 in our revised manuscript), reaching the highest value of 100% (average 89.3% for 173 MAGs). This high accuracy may be attributed to the inclusion of the complete genome of BVAB3 in its reference database (NCBI accession ID: GCF_000025225.2). For the other two major Saccharofermentales species, SGB009 (ala. BVAB2) and SGB080 (a BVAB2-like species), their completeness evaluations achieved a maximum of 96.5% (average 90.1% for 473 MAGs) and 96.4% (average 84.9% for 36 MAGs), respectively. However, this completeness was still underestimation, as two finished genomes of SGB009 and SGB090 (NCBI accession IDs GCF_029101565.1 and GCF_029101585.1, respectively, both represented as single no-gapped contig from a recently published study [1]) only resulted in a completeness value of 94.1% for both. Despite this, within our SGB009 and SGB080, there are still 13 and 3 near-complete genomes (>90% completeness, <5% contamination, and presence of full-length rRNA genes and ≥ 18 tRNAs), respectively, along with a substantial number of high-quality genomes (n = 360 and 14, respectively, >90% completeness and <5% contamination). These genomic resources contribute significantly to advancing research on these bacteria, facilitating studies such as evolutionary and functional analyses.

References:

[1] Srinivasan S, Austin M N, Fiedler T L, et al. *Amygdalobacter indicium* gen. nov., sp. nov., and *Amygdalobacter nucleatus* sp. nov., gen. nov.: novel bacteria from the family *Oscillospiraceae* isolated from the female genital tract. *International journal of systematic and evolutionary microbiology*, 2023, 73(10): 006017.

• *Reporting on the taxonomic profiling of the ~3,000 vaginal metagenomic samples would benefit from considering the percentage of reads that remain unclassified. This is done in Figure 8C and shows that for some samples a substantial portion (>40%) of reads could not be assigned to the VMGC. I believe the results provided in Figure 2D would be more insightful if they considered the portion of unclassified read. Alternatively, I would suggest this be provided as a supplemental figure so readers can evaluate how the percentage of unclassified reads changes across the ~3,000 vaginal metagenomes.*

Response: Thanks very much for this constructive comment. In the revised manuscript, based on your suggestion, we have added a supplementary figure (Figure S10; also see supporting figure 3 below) to

provide a detailed presentation of unclassified reads. When we use VMGC-genes as a reference, an average of 25.5% (median = 19.9%, interquartile range [IQR] = 16.9%-24.4%) of reads are unmapped. When we use VMGC-genomes as a reference, an average of 16.2% (median = 8.3%, IQR = 5.7%-16.3%) of reads are unmapped.

• *The methods indicate that prokaryotic MAGs with similar sequencing depth, GC content, and taxonomic assignments were merged. Were all MAGs satisfying the specified criteria merged? My concern is that merging MAGs in this way might result in a chimeric MAG with high contamination in some cases. Were such chimeras observed, as evident from greatly increased CheckM or BUSCO contamination estimates, and subsequently left as individual MAGs?*

Response: Thank you very much for this insightful and professional comment. Based on your suggestions and those of other reviewers, we have discontinued the practice of combining raw MAGs in the revised manuscript, to prevent potential contamination and inaccuracies. Specifically, based on the suggestions from Reviewer #3 and in light of recent methodological publications [1-2], we have implemented an updated approach that utilizes both single-coverage binning and mash-based multi-coverage binning processes for the assembled contigs to recover MAGs from metagenomic samples. This revised approach has demonstrated a significantly enhanced effectiveness in MAG reconstruction compared to the singular single-coverage approach employed in our original manuscript. For a comprehensive description of these modifications, please refer to our response to Comment #2 raised by Reviewer #3.

References:

[1] Mattock J, Watson M. A comparison of single-coverage and multi-coverage metagenomic binning reveals extensive hidden contamination. *Nature methods*, 2023, 20(8): 1170-1173.

[2] Carter M M, Olm M R, Merrill B D, et al. Ultra-deep sequencing of Hadza hunter-gatherers recovers vanishing gut microbes. *Cell*, 2023.

• *What bowtie2 parameters were used to map reads to genome in the VMGC for the purposes of taxonomic profiling?*

Response: Thanks for pointing this out. In the revised manuscript, based on the suggestions from Reviewer #3 (Comments 4 and 5), we have ceased using Bowtie 2 for taxonomic profiling in vaginal metagenomes. Instead, we have adopted the Kraken2 and Bracken tools to achieve a more accurate determination of the taxonomic composition. Both tools were conducted with default parameters, and a

custom Kraken2 database was constructed by incorporating all microbial species from the VMGC. The updated methodology was elucidated in the Methods section of the revised manuscript.

• *I believe the statement “Vaginal samples with a relatively low proportion of potential microbial content (>400,000 clean reads) were eliminated before generating taxonomic profiles” should indicate <400,000 reads. How many of the samples failed this criterion?*

Response: Thank you for pointing this error out. Indeed, it should state, "*vaginal samples with <400,000 clean reads were eliminated for further analysis*". In the original manuscript, 85.2% (2,936/3,446) of the samples met this criterion (>400,000 clean reads) and were retained for taxonomic profiling analysis. On the other hand, as pointed out by Reviewer #3 (Comment #6), the filtering condition of "<400,000 clean reads" may introduce bias into the results. Therefore, in the revised manuscript, we have decided to discontinue the use of this filtering criterion for vaginal metagenomes. All vaginal samples are now included in our assessment.

• *The methods section on the genomic analysis of Saccharofermentanales members references Supplementary Figure 4 as indicating the 14 marker genes that were largely absent across these genomes. Supplementary Figure 4 does not contain this information. Perhaps this was meant to reference a specific supplementary table? Are the 14 absent markers the same for SGB006, SGB028, and SGB069? As currently stated, it is not clear that this is the case.*

Response: We apologize for the error in the Method section of the original manuscript, where "Supplementary Figure 4" was incorrectly referenced. The accurate references should be to "Supplementary Figure 5" and "Supplementary Table 8" in the original manuscript. In the revised manuscript, we have rectified this oversight and introduced a new figure (Figure 4c, also see supporting figure 4 below). Among the 14 absent USCO genes for SGB009, SGB034, and SGB080, 10 are shared by all three species. These 10 genes also appear to be absent in all Saccharofermentanales species in the human vagina.

- *Why were fungal species delineated at 96% ANI while prokaryotic and viral species were delineated at 95% ANI?*

Response: Thank you very much for your valuable insights. Based on your suggestion, in the revised manuscript, we conducted a parameter exploration for defining fungal species boundaries. We utilized over 11,623 available fungal genomes from the NCBI database (downloaded in June 2023), performed pairwise average nucleotide identity (ANI) calculations for them, and conducted analyses based on known species assignments. When setting the ANI threshold at 96%, we observed that 98.76% of intra-species pairs met or exceeded this value, with only 0.14% of inter-species pairs achieving this threshold (equivalent to false positives) (see supporting figure 5 below). Similarly, at an ANI threshold of 95%, 99.09% of intra-species pairs reached this criterion, while only 0.16% of inter-species pairs did. Therefore, setting the standards at both 96% and 95% for species definition proved to be highly accurate. Considering that an ANI of 95% is widely employed as a standard in prokaryotic and viral genome studies [1-2], and in accordance with your suggestion and that from other reviewers, we have adopted the 95% threshold as the definition for fungal species in the revised manuscript. This adjustment is aimed at aligning our work with established practices in related genomic research fields.

References:

- [1] Jain C, Rodriguez-R L M, Phillippy A M, et al. High throughput ANI analysis of 90K prokaryotic genomes reveals clear species boundaries. *Nature communications*, 2018, 9(1): 5114.
- [2] Gregory A C, Zayed A A, Conceição-Neto N, et al. Marine DNA viral macro- and microdiversity from pole to pole. *Cell*, 2019, 177(5): 1109-1123. e14.

• *For the identification of virus sequences, the methods indicate that contigs with “fivefold the number of viral genes” were removed. Should this indicate that contigs with less than fivefold the number of viral genes were removed?*

Response: We apologize for this confusion caused by our previous misdescription. We have revised this sentence as follows: "*We initially evaluated contigs using CheckV v0.7.0, removing those with a host gene count greater than 10 and simultaneously exceeding 5 times the number of viral genes.*"

- *Why was 75% alignment fraction used to delineated vOTUs? Is this the criterion used in previous studies?*

Response: Thank you for your valuable comment. In the original manuscript, we adopted a 75% alignment fraction for vOTU definition based on a previous study [1] and two of our previous works [2-3]. Upon your suggestion, we investigated recent literature on building viral catalogues from metagenomes, and the parameter choices from these studies are listed in supporting table 2 below. Based on these studies, we found that a threshold of 95% nucleotide similarity across 85% of the sequence alignment fraction (coverage) is the most frequently used. Particularly, in the study by Roux et al. [4], they conducted a data-driven parameter exploration based on known viruses in the NCBI RefSeq and IMG/VR databases, discovering that 95% nucleotide similarity and 85% coverage are most suitable for vOTU clustering. Therefore, in the revised manuscript, we have replaced the original 75% coverage with 85% coverage for all viral vOTU analyses. We appreciate your professional advice, which has enhanced the accuracy and generalizability of our results.

Study	% Similarity	% Coverage	
Roux_2019 [4]	95	85	Supporting table 2. Parameters for vOTU clustering used in previous studies.
Gregory_2019 [5]	95	70	
Gregory_2020 [6]	95	70	
Nayfach_2020 [7]	95	85	
Camarillo-Guerrero_2021 [1]	95	75	
Nayfach_2021 [8]	95	85	
Tisza_2021 [9]	95	85	
Li_2022 [2]	95	75	

References:

- [1] Camarillo-Guerrero L F, Almeida A, Rangel-Pineros G, et al. Massive expansion of human gut bacteriophage diversity. *Cell*, 2021, 184(4): 1098-1109. e9.
- [2] Li S, Guo R, Zhang Y, et al. A catalog of 48,425 nonredundant viruses from oral metagenomes expands the horizon of the human oral virome. *iScience*, 2022, 25(6).
- [3] Chen F, Li S, Guo R, et al. Meta-analysis of fecal viromes demonstrates high diagnostic potential of the gut viral signatures for colorectal cancer and adenoma risk assessment. *Journal of advanced research*, 2023, 49: 103-114.
- [4] Roux S, Adriaenssens E M, Dutilh B E, et al. Minimum information about an uncultivated virus genome (MIUViG). *Nature biotechnology*, 2019, 37(1): 29-37.
- [5] Gregory A C, Zayed A A, Conceição-Neto N, et al. Marine DNA viral macro- and microdiversity from pole to pole. *Cell*, 2019, 177(5): 1109-1123. e14.
- [6] Gregory A C, Zablocki O, Zayed A A, et al. The gut virome database reveals age-dependent patterns of virome diversity in the human gut. *Cell host & microbe*, 2020, 28(5): 724-740. e8.
- [7] Nayfach S, Camargo A P, Schulz F, et al. CheckV assesses the quality and completeness of metagenome-assembled viral genomes. *Nature biotechnology*, 2021, 39(5): 578-585.
- [8] Nayfach S, Páez-Espino D, Call L, et al. Metagenomic compendium of 189,680 DNA viruses from the human gut microbiome. *Nature microbiology*, 2021, 6(7): 960-970.
- [9] Tisza M J, Buck C B. A catalog of tens of thousands of viruses from human metagenomes reveals hidden associations with chronic diseases. *Proceedings of the national academy of sciences*, 2021, 118(23): e2023202118.

Reviewer #3 (Remarks to the Author):

Generation of metagenomics in the vaginal microbiome has generally lagged behind other organ systems, and, consequently, so has the generation of large-scale reference datasets. Huang et al. attempt to address this by integrating available metagenomic data, reference microbial genomes, and very modest in-house cultivation of vaginal fungi to construct a comprehensive database of microbial genomes from the vaginal environment. This study is timely and addresses a clear gap in the field. Data acquisition efforts are extensive, resulting in over 3400 metagenomic samples from across multiple locations around the world. This reference dataset could be useful to the field, and includes elusive genomes such as that of BVAB. However - the analyses provided are highly descriptive, and in my opinion do not present particularly novel insights; I have significant methodological concerns that make me significantly doubt the validity of this database; there is little to no validation provided to support the validity of the work; and the data and code are not available in a meaningfully useful way.

Response: We genuinely appreciate your thoughtful and constructive comments and suggestions. Based on your invaluable feedback, we have implemented several significant modifications in the revised manuscript as follows:

- 1) We have included a more recent set of samples (up to October 2023), adding 1029 vaginal metagenomes from six projects (increasing the total from 3443 to 4472 samples).
- 2) We adopted the strategy that combines single-coverage and multi-coverage metagenomic binning for MAG reconstruction. The previously used combined MAG approach is no longer employed to avoid potential errors. Please refer to the response to Comment #2 for a detailed description of our current methods and results.
- 3) Methodological improvements have been made based on your and other reviewers' suggestions. We incorporated several new methods/tools for analysis, such as using the latest version of the CHM13 reference genome for filtering host reads, employing CheckM v2 and GUNC for MAG selection and assessment, using Kraken2 for abundance calculation, utilizing EukRep for fungal identification, and exploring and adjusting other analysis parameters.
- 4) Regarding data and code availability, following your advice, we have uploaded the data to Zenodo (URL: <https://zenodo.org/records/10457006>) and the code to GitHub (URL: <https://github.com/RChGO/VMGC>) repositories, respectively.

Overall, based on updated samples, databases, and analysis methods, we have comprehensively revised and improved the entire manuscript. For a detailed response to each of your concerns, please refer to the point-by-point breakdown below.

251. Recent work has demonstrated substantial human contamination caused by the use of incomplete human reference genome, such as GRCh38 (Gihawi et al., *bioRxiv* 2023). The authors should switch to CHM13 with the most recent Y chromosome sequence and also detail the parameters used by bowtie for full replication.

Response: Thanks for this valuable comment. As the reviewer pointed out, CHM13 not only possesses a more complete Y chromosome sequence compared to GRCh38 but also exhibits more comprehensive sequences for other chromosomes. By mapping each vaginal metagenome against CHM13 to secondarily remove human reads, we observed that the original set of clean reads still contained a certain number of human reads. Additionally, we noticed that due to fastp's complexity filter not being designed to detect certain types of low complexity sequences, such as simple repeats (e.g., di- or tri-nucleotide repeats), a certain proportion of low complexity reads remained in our original clean reads. To address these two issues, in the revised manuscript, we took the following steps to process all vaginal metagenomes for low-quality filtering and host removal:

- 1) To ensure data quality, raw reads from each sample were subjected to quality control using fastp v0.20.165 [1] with the parameters “-u 30 -q 20 -l 30 -y --trim_poly_g” for samples with read length ≤ 75 bp, and “-u 30 -q 20 -l 60 -y --trim_poly_g” for samples with read length > 75 bp.
- 2) A secondary complexity filter was applied to each vaginal metagenome using bbduk with the parameters “entropy=0.6 entropywindow=50 entropyk=5” from the BBTools suite v39.00 [2].
- 3) The reads deriving from human and *Escherichia phage* phiX174 were eliminated by mapping the filtered reads against CHM13 v2.0 and phiX174 (NCBI accession NC_001422.1) genomes using Bowtie2 v2.4.1 [3] with the parameters “--end-to-end -fast”. In the mapping results, paired reads with a SAM flag = 77 or 141 were removed, while single reads with a SAM flag = 0 or 16 were excluded.

References:

[1] Chen S, Zhou Y, Chen Y, et al. fastp: an ultra-fast all-in-one FASTQ preprocessor. *Bioinformatics*, 2018, 34(17): i884-i890.

[2] Bushnell B. BBTools: a suite of fast, multithreaded bioinformatics tools designed for analysis of DNA and RNA sequence data. *Joint Genome Institute*. 2018.

[3] Langmead B, Salzberg S L. Fast gapped-read alignment with Bowtie 2. *Nature methods*, 2012, 9(4): 357-359.

2. Many newer binning methods have been published recently, including VAMB, MetaCOAG, SemiBin2. Additionally, recent work established that multi-binning should be used instead of single-binning (Mattock & Watson 2023). These advances have not been applied in this work. Can the authors validate and or justify their methodological pipeline?

Response: Thank you for your insightful comment. Based on your suggestion and recent methodological publications [1], in the revised manuscript, we have adopted a multi-coverage binning strategy for MAG reconstruction. Considering the substantial number of vaginal metagenomes ($n = 4,472$ samples), the conventional multi-coverage binning approach based on pairwise read alignment ($4,472 \times 4,472$) is computationally infeasible. Therefore, referring to a recent study [2], we explored an improved approach based on Mash [3] to select a specific number of samples for multi-sample read alignment and MAG clustering. The specific approach is outlined as follows:

- 1) For all 4,472 samples, we performed metagenomic assembly individually and calculated the nucleotide-level similarity (Mash distance) between the assembled contig sets of all sample pairs using Mash v2.3.
- 2) For each sample, we selected the N most similar samples based on Mash distance, mapped their reads to the contigs of the target sample, and calculated the contigs' sequencing depth in the most similar samples using the *jgi_summarize_bam_contig_depths* tool.
- 3) For each target sample, we integrated the depth files of its N most similar samples using *combine.pl* (https://github.com/WatsonLab/single_and_multiple_binning/blob/main/scripts/combine.pl) to obtain the contigs' multi-sample depth. Based on this integrated depth file, we performed multi-coverage metagenomic binning for the contigs using MetaBAT 2 [4], ultimately obtaining the raw bins for each sample.

In this approach, a critical parameter is " N ", representing the number of most similar samples used for calculating sequencing depth. To determine this parameter, we randomly selected 80 vaginal samples and conducted multi-coverage binning processes for each sample using the top 10-60 closest samples (from 4,472 samples) based on Mash distance. This test revealed that compared to single-coverage binning, all multi-coverage binning approaches, regardless of selecting 10-60 samples, yielded a significantly higher number of MAGs. The improvement ranged from 17.5% to 25.3% for medium-quality MAGs and 48.5% to 70.3% for high-quality MAGs (see supporting figure 6a below). Simultaneously, the proportion of heterozygous MAGs generated was reduced (medium-quality MAGs, 10.4% in single-coverage binning vs. an average of 2.2% in multi-coverage binning; high-quality MAGs, 8.9% vs. an average of 1.4%; see supporting figure 6b below). With the increase in N in multi-coverage

27binning, there was a moderate increase in the number of medium and high-quality MAGs. However, after N reached 20, the trend of increasing MAG quantity approached a plateau, and the proportion of heterozygous MAGs also stabilized. This result indicates that, for our vaginal samples, selecting the 20 closest samples for multi-coverage binning is appropriate.

Using the described approach, we applied multi-coverage binning to all samples, resulting in 16,518 prokaryotic MAGs that met the filtering criteria ($\geq 50\%$ completeness, $< 5\%$ contamination, quality score ≥ 50 , and clade separation score < 0.45). Additionally, we performed single-coverage binning for each sample, yielding 11,428 prokaryotic MAGs. Subsequently, we conducted strain-level de-redundancy (99% ANI) for MAGs obtained from both multi-coverage and single-coverage binning within each sample, resulting in a total of 18,570 MAGs (where 13,857 and 4,713 come from single-coverage and multi-coverage binning, respectively).

binning approach parameters. **(a)** The percentages in the chart represent the ratio of the increase (or decrease) in the number of bins/MAGs obtained through multi-coverage binning to the number of bins/MAGs obtained through single-coverage binning. **(b)** The percentages in the chart represent the proportion of bins/MAGs with GUNC CSS ≥ 0.45 (indicating heterozygous bins/MAGs) among the total number of bins/MAGs.

References:

- [1] Mattock J, Watson M. A comparison of single-coverage and multi-coverage metagenomic binning reveals extensive hidden contamination. *Nature methods*, 2023, 20(8): 1170-1173.
- [2] Carter M M, Olm M R, Merrill B D, et al. Ultra-deep sequencing of Hadza hunter-gatherers recovers vanishing gut microbes. *Cell*, 2023.
- [3] Ondov B D, Treangen T J, Melsted P, et al. Mash: fast genome and metagenome distance estimation using MinHash. *Genome biology*, 2016, 17(1): 1-14.
- [4] Kang D D, Li F, Kirton E, et al. MetaBAT 2: an adaptive binning algorithm for robust and efficient genome reconstruction from metagenome assemblies. *PeerJ*, 2019, 7: e7359.

3. The authors add a custom binning step – combining MAGs if they show a close sequencing depth and GC content. What is the justification for it? What validation is offered to this? Why is this step not already performed by binning? On the face of it, this step seems methodologically invalid, and while it inflates completeness of the MAGs, recent studies (Mattock & Watson 2023) show that this likely creates chimeras.

Response: Thank you very much for this valuable comment. Based on your suggestions and those of other reviewers, we have discontinued the practice of combining raw MAGs in the revised manuscript, to prevent potential contamination and inaccuracies. Specifically, as mentioned in the response to the previous comment, we have implemented an updated approach that utilizes both single-coverage binning and mash-based multi-coverage binning processes for the assembled contigs to recover MAGs from metagenomic samples. This revised approach has demonstrated a significantly enhanced effectiveness in MAG reconstruction compared to the singular single-coverage approach employed in our original manuscript. Furthermore, in accordance with your recommendation, we have incorporated GUNC (with a threshold set at clade separation score ≥ 0.45) in the revised manuscript to eliminate

29potential chimera MAGs. This addition ensures a more rigorous screening process, contributing to the overall reliability of our findings.

4. The method used by the authors to generate abundances is very sensitive to coverage. I believe some sort of subsampling or accounting for coverage with a relevant statistical model is necessary.

Response: Thanks for pointing this out. In the revised manuscript, based on your suggestions, we have ceased using Bowtie 2 for taxonomic profiling in vaginal metagenomes. Instead, we have adopted the Kraken2 and Bracken tools to achieve a more accurate determination of the taxonomic composition. Both tools were conducted with default parameters, and a custom Kraken2 database was constructed by incorporating all microbial species from the VMGC. The updated methodology was elucidated in the Methods section of the revised manuscript.

5. Estimating abundance from sequencing coverage (P13, L22-23) is fraught with issues. In generating taxonomic profiles, the authors should at the very least examine the coverage uniformity. If only a small segment of the genome is covered, it should not get any abundance at all.

Response: Thanks for this insightful comment. In response to the previous question, we transitioned to using Kraken2 and Bracken tools to profile the prokaryotic composition of all vaginal metagenomic samples. Considering that Kraken2 is based on the lowest common ancestor (LCA) principle for read alignment, it implies that reads mapped against a genome are typically those that have successfully undergone specific alignment. This approach significantly reduces false positives caused by non-specific alignment, thereby diminishing the need for further assessment of genome coverage.

6. Given the high human read % in vaginal samples (~90-95%), eliminating samples with <400K reads (P13, L20) corresponds to eliminating samples with less than 8M reads and will generate a bias towards samples with high bacterial fraction (potentially, high load?). This seems like an unreasonable decision.

Response: Thank you for providing this insightful comment. Based on your suggestion, we conducted profiling of the microbial composition for all vaginal samples and employed multivariate analyses to

30compare samples with read counts above and below 400,000. The results (see supporting figure 7 below), aligning with the reviewer's perspective, demonstrated subtle yet visible differences in the PCoA plot between these two sample groups. Consistently, PERMANOVA indicated the significance of this difference (*adonis* $p < 0.001$). Therefore, in the revised manuscript, to ensure a more inclusive and representative analysis, we have discontinued the use of 400,000 reads as a filtering criterion, and all vaginal samples are now incorporated into our assessment.

7. Regarding missingness of USCOs (P6) – first, the authors do not show an analysis to support that these are the SAME USCOs that are missing across all MAGs – just that it's the same number. Second, the authors would need to rule out the presence of those missing genes in the un-binned fraction of the contigs or sequence reads. It's possible that those genes happen to be placed next to a variable region in the genome, which are known to not be binned properly.

Response: Thanks very much for this constructive comment. In the original manuscript, we included Supplementary Table 8 to showcase the absence of USCOs for each species. We apologize if, due to formatting issues, the information in this table was not accessible. Specifically, the 3 primary species

belonging to Saccharofermentanales all lack 14 USCOs, with 10 being commonly absent, including 5 enzymes related to purine synthesis. This finding remains consistent in the updated manuscript (see supporting figure 8 below).

Regarding your inquiry into whether the absence of these genes is due to technical reasons, in the revised manuscript, we have included the complete genomes of these 3 species with the following NCBI accession IDs: GCF_000025225.2 (*Mageibacillus indolicus*, aga. BVAB3, also SGB034 in our revised manuscript), GCF_029101565.1 (*Amygdalobacter indicium*, aga. BVAB2, also SGB009 in our revised manuscript), and GCF_029101585.1 (*Amygdalobacter nucleatus*, a BVAB2-like species, also SGB080 in our revised manuscript). All 3 genomes consist of a single circular no-gapped contig. The genomes of *A. indicium* and *A. nucleatus* were sourced from a recent paper published in October 2023 [1], and this paper did not mention the presence of plasmids or other non-chromosomal elements in their genomes. As expected, the complete genomes of these 3 species do not contain the previously mentioned 14 USCOs, suggesting that the absence of these USCOs is indeed a natural characteristic of these species.

References:

[1] Srinivasan S, Austin M N, Fiedler T L, et al. *Amygdalobacter indicium* gen. nov., sp. nov., and *Amygdalobacter nucleatus* sp. nov., gen. nov.: novel bacteria from the family *Oscillospiraceae* isolated from the female genital tract. *International journal of systematic and evolutionary microbiology*, 2023, 73(10): 006017.

8. Identification of fungal MAGs (P14, L24) is unclear and seems unvalidated and potentially non-specific. Taxonomic similarity of 80% to what? Specl was developed for prokaryotes – is it even valid for fungi? How was this validated?

Response: Thanks for pointing this out. We sincerely apologize for the confusion caused by our previous inaccurate description. In reality, to reduce computational burden, we excluded MAGs with Specl similarities exceeding 80% (which are highly likely to be prokaryotic MAGs) before proceeding with the identification of fungal MAGs. In the revised manuscript, based on your suggestion, we have discontinued the use of Specl and the original fungal identification method. Instead, we have adopted the EukRep tool [1] to directly identify fungi from all raw MAGs, ultimately resulting in 13 fungal MAGs with completeness $\geq 50\%$.

References:

[1] West P T, Probst A J, Grigoriev I V, et al. Genome-reconstruction for eukaryotes from complex natural microbial communities. *Genome research*, 2018, 28(4): 569-580.

9. P7, L36: 95% ANI threshold, while heavily contested in prokaryotes, is at least based on a data-driven analysis. Is there any analysis to support the use of this threshold in viruses?

Response: Thank you very much for this insightful comment. In this study, our utilization of 95% ANI as the threshold for vOTU clustering is grounded in previous studies [1-6] and our prior works [7-9]. In particular, studies by Roux et al. [10] and Gregory et al. [1] have conducted parameter exploration in this context, affirming that 95% serves as the most suitable boundary for defining viral species-level population (or vOTUs). Furthermore, taking into account your suggestion, that of Reviewer #2, and the parameter exploration by Roux et al. (for details, refer to the response to Reviewer #2 Comment #10), we have adjusted another parameter for vOTU clustering, specifically the sequence alignment fraction, from 75% to 85%. All corresponding results have been updated accordingly. We appreciate your professional advice, which has contributed to the refinement of the accuracy and applicability of our findings.

References:

- [1] Gregory A C, Zayed A A, Conceição-Neto N, et al. Marine DNA viral macro- and microdiversity from pole to pole. *Cell*, 2019, 177(5): 1109-1123. e14.
- [2] Gregory A C, Zablocki O, Zayed A A, et al. The gut virome database reveals age-dependent patterns of virome diversity in the human gut. *Cell host & microbe*, 2020, 28(5): 724-740. e8.
- [3] Camarillo-Guerrero L F, Almeida A, Rangel-Pineros G, et al. Massive expansion of human gut bacteriophage diversity. *Cell*, 2021, 184(4): 1098-1109. e9.
- [4] Nayfach S, Camargo A P, Schulz F, et al. CheckV assesses the quality and completeness of metagenome-assembled viral genomes. *Nature biotechnology*, 2021, 39(5): 578-585.
- [5] Nayfach S, Páez-Espino D, Call L, et al. Metagenomic compendium of 189,680 DNA viruses from the human gut microbiome. *Nature microbiology*, 2021, 6(7): 960-970.
- [6] Tisza M J, Buck C B. A catalog of tens of thousands of viruses from human metagenomes reveals hidden associations with chronic diseases. *Proceedings of the national academy of sciences*, 2021, 118(23): e2023202118.
- [7] Li S, Guo R, Zhang Y, et al. A catalog of 48,425 nonredundant viruses from oral metagenomes expands the horizon of the human oral virome. *iScience*, 2022, 25(6).
- [8] Li S, Yan Q, Wang G, et al. Cataloguing and profiling of the gut virome in Chinese populations uncover extensive viral signatures across common diseases. *bioRxiv*, 2022: 2022.12. 27.522048.
- [9] Chen F, Li S, Guo R, et al. Meta-analysis of fecal viromes demonstrates high diagnostic potential of the gut viral signatures for colorectal cancer and adenoma risk assessment. *Journal of advanced research*, 2023, 49: 103-114.
- [10] Roux S, Adriaenssens E M, Dutilh B E, et al. Minimum information about an uncultivated virus genome (MIUViG). *Nature biotechnology*, 2019, 37(1): 29-37.

10. *How many phages are integrated? Are they even real phages? For how many of these were closed (circular) genomes assembled?*

Response: Thanks for this valuable comment. Based on your suggestion, we aligned all viral sequences (n = 14,224) against all 19,542 prokaryotic MAGs in the VMGC to assess their integrated states. This

34analysis revealed that only 3.9% (553/14,224) of viruses were successfully aligned (>95% nucleotide similarity across >85% of the sequence), indicating a possible integrated state. However, this is considered a preliminary estimate since, to our knowledge, there is currently no well-established method for distinguishing between free viral particles and integrated viruses in metagenomic samples. Furthermore, we evaluated the lifestyles of all high-quality viruses ($\geq 90\%$ completeness) using the BACPHLIP tool [1]. This analysis showed that out of the 5,637 high-quality viruses, 43.4% (2,444/5,637) are virulent viruses, while the remaining 56.6% are temperate viruses (Supplementary Table 5 in the revised manuscript). Additionally, among the 14,224 viruses in the revised manuscript, 1,467 could be assessed as complete genomes based on the CheckV algorithm, with 1,351 containing direct terminal repeats (DTRs), suggesting closed genomes. These results have been incorporated into the revised manuscript, as per your suggestion.

References:

[1] Hockenberry A J, Wilke C O. BACPHLIP: predicting bacteriophage lifestyle from conserved protein domains. *PeerJ*, 2021, 9: e11396.

11. How many of these species actually come from the vaginal environments? Many of the bacterial species found (177) had been previously identified only in other environments (from other body sites mostly). Could these be contamination? What is their prevalence across the metagenomic samples? How is contamination controlled for?

Response: Thank you very much for your thorough comment. To ensure that the reconstructed MAGs and isolated genomes originate from the female vagina, we undertook the following measures in data sourcing: 1) all vaginal metagenomic samples were carefully selected to ensure they originated from human females, primarily verified through NCBI records or corresponding published literature; 2) each isolated strain was scrutinized for publicly available metadata information (mainly from NCBI and JGI websites or relevant literature), confirming its association with human female vaginal sources.

In the revised manuscript, based on our new taxonomic profile result, almost all species (97.3%, 765/786) were found to appear in >20 metagenomic samples. These species include 246 (among 249 species) that have been previously identified only in other body sites or environments. This result strongly suggests that the presence of most species is not the result of accidental contamination.

Additionally, to avoid issues with sequence traceability due to renumbering, all our genome identifiers retain their original NCBI BioSample IDs. This practice facilitates the swift removal of related sequences from the VMGC in case of contamination concerns raised by peers. It's important to note that, despite our efforts to minimize contamination based on publicly available information, we cannot entirely eliminate the possibility of metadata errors or contamination issues introduced during experimental procedures.

12. Extensive validations are necessary here. Some suggestions could be comparison to 16S; examination of CSTs; comparisons to isolates; comparisons to other studies. In that regard, the comparison to VIRGO (P9, L33-34) is VERY alarming. With the exact same samples, there is only 58% coverage of the VIRGO dataset? If the authors have a potential explanation to this, such as L35, they should prove that this is the case.

Response: Thank you very much for the helpful comment. As the reviewer pointed out, we also noticed that VMGC still lacks a considerable proportion of genes present in VIRGO. Even in the revised manuscript, there are still 287,810 (~41%) genes from VIRGO-90 (n = 708,006) that are not included in VMGC-90 (n = 1,415,799) (see supporting figure 9 below). Following your suggestion, we investigated the sources of these VIRGO-specific genes, including:

- 1) In the selection of publicly available isolated strains, we observed that out of the 308 isolate genomes used by VIRGO, 64 genomes were not included in our collection. Further analysis revealed that 60 of them were actually isolated from other human body parts (e.g., oral cavity, male urethra, or others) or non-human hosts (see supporting table 3 below); for the remaining 4, we could not identify their sources. These 64 non-vaginal genomes contain ~86k non-redundant genes, of which ~76k are not present in VMGC-90 (see supporting figure 9 below). After excluding these genomes, VIRGO still has 211,705 (~30% of the total) genes not included in VMGC.
- 2) A possible reason is that VIRGO may contain a large number of genes from low-abundance species or other unbinned genomic elements (e.g., plasmids) that are not covered by MAGs and thus not present in VMGC. To verify this, we directly constructed a gene catalogue from the assembled contigs of all investigated samples (consisting of ~3.5 million non-redundance genes). These genes were aligned against the remaining VIRGO-specific genes, and ~79k genes were found to have matches (see supporting figure 9 below). After removing genes from unbinned sequences, there are still 133,018 genes in VIRGO (~19% of the total) without identified sources.

As VIRGO did not publicly disclose the original data of metagenome samples used to build the non-redundant gene catalogue, and only provided additional data for testing (~1,400 metagenome samples, which have all been incorporated into our study), we currently have no means to assess the origin of

36these ~133k VIRGO-specific genes. On the other hand, our rarefaction analysis reveals that the gene content of VMGC (VMGC-90) is, in fact, undersaturated under the current sample size (Figure 8a in the revised manuscript). This suggests that despite the substantial size of the current database, there are still numerous unknown genes within the reproductive tract that require exploration in future studies.

VIRGO specific isolates (n = 64)	NCBI BioSample ID	NCBI genome ID	Body Site (Host)
Brucella abortus bv 2, 86/8/59	SAMN02595278	GCF_000157735.1	Reproductive system (Bos taurus)
Brucella abortus bv 4, 292	SAMN02595276	GCF_000157695.1	Reproductive system (Bos taurus)
Brucella abortus bv 5, B3196	SAMN02595354	GCF_000163115.1	Reproductive system (Bos taurus)
Brucella abortus bv 6, 870	SAMN02770262	GCF_000740215.1	Reproductive system (Bos taurus)
Brucella abortus bv 9, C68	SAMN02769911	GCF_000740195.1	Reproductive system (Bos taurus)
Campylobacter fetus venerealis Azul-94	SAMN02471365	GCF_011600945.2	vagina ? (Bos taurus)
Campylobacter fetus venerealis bv. intermedius 99541	SAMN02469943	GCF_011600945.2	prepuce of a naturally-infected bull (Bos taurus)
Campylobacter fetus venerealis X/161/5, NCTC 10354	SAMN01918896	GCF_011600945.2	Vaginal mucus (heifer)
Campylobacter jejuni jejuni IA3902	SAMN02603425	GCF_000009085.1	aborted placenta (sheep)
Chlamydophila psittaci 02DC15	SAMN02603518	GCF_000204255.1	? (cattle)
Clostridium botulinum E3 Alaska E43	SAMN02603537	GCF_000020285.1	Salmon eggs? (Food)
Corynebacterium ciconiae DSM 44920	SAMN02256507,	GCF_000372385.1,	Trachea (Ciconia nigra)

37SAMN13404516	GCF_030440575.1	
Corynebacterium uterequi DSM 45634 Genome Sequencing 1	SAMN03480647	GCF_001021065.1	Reproductive system (Equus caballus)
Eggerthella lenta 1_60AFAA	SAMN00103544	GCF_000763035.1	Terminal ileum (Homo sapiens)
Eggerthella lenta VPI 0255, DSM 2243	SAMN00002594	GCF_000024265.1	Blood (Homo sapiens)
Eggerthella sp. 1_3_56FAA	SAMN02463797	GCF_000185625.1	Terminal ileum (Homo sapiens)
Eggerthella sp. HGA1	SAMN00116781	GCF_000191845.1	Gastrointestinal tract (Homo sapiens)
Eggerthella sp. YY7918	SAMD00060993	GCF_000270285.1	Intestinal tract (Homo sapiens)
Haemophilus parainfluenzae HK262	SAMN00761858	GCF_000259485.1	Urethra (Homo sapiens)
Lactobacillus acidophilus NCFM	SAMN02603047	GCF_000011985.1	Gastrointestinal tract (Homo sapiens)
Lactobacillus acidophilus NCFM (re- annotation)	SAMN02603047	GCF_000011985.1	Gastrointestinal tract (Homo sapiens)
Lactobacillus ruminis E 194e	SAMN00001480	GCF_000159375.2	Gastrointestinal tract (Homo sapiens)
Leptotrichia buccalis C-1013-b, DSM 1135	SAMN00002593	GCF_000023905.1	Dental plaque (Homo sapiens)
Leptotrichia goodfellowii F0264	SAMN00004572	GCF_000176335.1	oral (Homo sapiens)
Leptotrichia goodfellowii LB 57, DSM 19756	SAMN02597279	GCF_000516535.1	oral (Homo sapiens)
Leptotrichia shahii DSM 19757	SAMN02440425	GCF_000373045.1	Dental plaque (Homo sapiens)
Leptotrichia sp. oral taxon 879 F0557	SAMN02436883	GCF_000469385.1	Dental plaque (Homo sapiens)
Megasphaera cerevisiae DSM 20462	SAMN03763890	GCF_001045675.1	Food?
Megasphaera elsdenii DSM 20460	SAMN08639723	GCF_003010495.1	? (Bos taurus)
Megasphaera elsdenii J1	SAMN04488492	GCF_900142305.1	rumen?
Megasphaera elsdenii T81	SAMN02744015	GCF_000621885.1	rumen?
Megasphaera elsdenii YE34	SAMN02910401	GCF_900113775.1	rumen?
Mesorhizobium sp. WSM4349	SAMN02441380	GCF_000373125.1	? (Acmispon glaber)
Mycobacterium smegmatis JS623	SAMN02261384	GCF_000328565.1	Skin (Homo sapiens)
Mycoplasma buteonis ATCC 51371	SAMN02745656	GCF_000733865.1	Respiratory system (Birds)

Mycoplasma canis UF33	SAMN02471771	GCF_000258965.1	Vagina (Canis lupus familiaris)
Mycoplasma columbinum ATCC 29257	SAMN02744819	GCF_000712175.1	trachea (Pigeon)
Mycoplasma columborale ATCC 29258	SAMN02841178	GCF_000701845.1	trachea (Pigeon)
Mycoplasma elephantis ATCC 51980	SAMN02743892	GCF_000687815.1	genital tract (elephant)
Mycoplasma fermentans PG18	SAMD00060938	GCF_000209735.1	Urogenital tract (Homo sapiens)
Mycoplasma gallinarum DSM 19816	SAMN02743976	GCF_000621085.1	respiratory tract (fowl)
Mycoplasma genitalium G37	SAMN02363424	GCF_000167595.1	male urethra (Homo sapiens)
Mycoplasma genitalium G37 (re-annotation)	SAMN02363424	GCF_000167595.1	male urethra (Homo sapiens)
Mycoplasma genitalium M2288	SAMN02603574	GCF_000292505.1	male urethra (Homo sapiens)
Mycoplasma genitalium M2321 (Second Assembly Apr12)	SAMN02603571	GCF_000292405.1	male urethra (Homo sapiens)
Mycoplasma genitalium M6282	SAMN02603573	GCF_000292445.1	male urethra (Homo sapiens)
Mycoplasma genitalium M6320	SAMN02603572	GCF_000292485.1	male urethra (Homo sapiens)
Mycoplasma imitans ATCC 51306	SAMN02584995	GCF_000518305.1	Turbinates (duck)
Mycoplasma primatum ATCC 25948	SAMN02841175	GCF_000702785.1	urethral tissues (monkey)
Mycoplasma suis KI_3806	SAMEA3138343	GCF_000203215.1	Blood (Sus scrofa)
Pasteurella multocida 36950	SAMN02603284	GCF_000234745.1	Respiratory system (bovine)
Phenylobacterium zucineum HLK1	SAMN02603048	GCF_000017265.1	Blood (Homo sapiens)
Prevotella sp. F0323	SAMN02463812	GCF_000234115.1	oral (Homo sapiens)
Streptococcus ictaluri 707-05	SAMN02436329	GCF_000188015.2	? (catfish)
Streptococcus macacae NCTC 11558	SAMN02436328	GCF_000187995.2	dental plaque (Macaca fascicularis)
Sutterella parvirubra YIT 11816	SAMN00189160	GCF_000250875.1	Gastrointestinal tract (Homo sapiens)
Sutterella wadsworthensis 3_45B	SAMN02463869	GCF_000186505.1	Gastrointestinal tract (Homo sapiens)
Taylorella asinigenitalis MCE3	SAMN02602982	GCF_000226625.1	genital tract (Equus asinus)
Taylorella equigenitalis ATCC 35865	SAMN02604145	GCF_000276685.1	cervical swab (Equine)

Taylorella equigenitalis MCE9	SAMN02602972	GCF_000185745.1	urethral fossa (Equus caballus)
Enterococcus faecalis 402/96 (ENT14)	?	?	?
Gardnerella vaginalis 42431	?	?	?
Lactobacillus sakei sakei 23K	SAMEA3138223	GCF_000026065.1	?
Leptotrichia buccalis ATCC14201	SAMN00187933	?	?

Supporting table 3. Sources of 64 bacterial isolates in the VIRGO database.

13. How does figure 8c compare to VIRGO?

Response: Thanks for pointing this out. Following your suggestion, we have added a supplementary figure (Supplementary Figure 10b, also see supporting figure 10 below) in the revised manuscript to show the mapping rates of vaginal metagenomes based on VMGC genes, VMGC genomes, and VIRGO, respectively. The reads recruited by VIRGO constituted an average of 71.7% (median = 79.2%) of the total clean reads. In comparison, reads recruited by VMGC genes (VMGC-90) and the VMGC genome set accounted for an average of 74.5% (median = 81.1%) and 83.8% (median = 91.7%) of the total clean reads, respectively.

14. Are the HPV viruses assembled enriched in patients with HPV in the various studies?

Response: Thank you for your insightful comment. Among the studies included in this research, only one study (Liu_2022, dataset accession ID PRJNA771720) [1] provided available HPV detection results. Following your suggestion, we conducted species profiling on vaginal metagenomic samples from four different groups in this study, including health controls (HC, n = 32), high-risk HPV positive without cervical lesion group (HR-HPV, n = 34), precancerous lesions with high-risk HPV group (CIN, n = 40), and invasive cervical cancer group (CC, n = 41), based on VMGC and compared the HPV abundance across different groups. In these samples, 36 HPV types were identified and quantified. The results revealed a continuous increase in the total HPV abundances with the severity of cervical lesions, showing a distinct upward trend from HC to HR-HPV to CIN to CC (see supporting figure 11 below). Specifically, three HPV types, including HPV 16, 52, and 58, showed a significant increase in the cervical

41lesion group compared to HC. These findings indicate that VMGC-based analysis can indeed uncover certain characteristics of HPV viruses in patients with cervical lesions.

References:

[1] Liu H, Liang H, Li D, et al. Association of cervical dysbacteriosis, HPV oncogene expression, and cervical lesion progression. *Microbiology spectrum*, 2022, 10(5): e00151-22.

15. *Data and code availability is unacceptable, especially for a resource paper: (a) code should be clear, well document, and enable full replication. A clear documentation of the process and how each code file relates to it is necessary. (b) The files themselves should be deposited in an independent repository (i.e., Zenodo). (c) clear documentation of the files, what is in each one, what are the variables, etc., should be provided.*

Response: Thank you very much for this valuable comment. Based on your suggestions, we have uploaded all the metadata, intermediate results, analysis codes, and visualization codes used in the manuscript to the GitHub repository, accessible at: <https://github.com/RChGO/VMGC>. Additionally, we have provided clear and detailed descriptions for these materials; for example, supporting figure 12 below illustrates an example of our analysis workflow for the single-coverage and Mash-based multi-

coverage binning approach. Furthermore, the data files of VMGC, including prokaryotic, eukaryotic, and viral genome sequences, annotation files, and the updated Kraken database, have been deposited in the Zenodo repository with the accession ID 10457006 (<https://zenodo.org/records/10457006>, see supporting figure 13 below for screenshot). We appreciate your careful review and hope these additional resources enhance the transparency and reproducibility of our work.Single-coverage and Mash-based multiple-coverage binning

Metagenome-assembled genomes in the VMGC were obtained through a process that integrated both single-coverage binning and Mash-based multiple-coverage binning methods. The specific operational steps are outlined below.

- Required dependencies

```
Mash 2.3
GNU parallel 20201122
bwa-mem2 2.2.1
MetaBAT v2
perl 5.16.3
combine.pl
```

- Step 1: Calculate the Mash distances between the assembled files, and obtain the top 20 closest samples for each assembled sample.

```
mash sketch -p 50 -s 100000 -k 32 -o mash.sketch contigs/*.fasta

find contigs/*.fasta | parallel -k -j 10 mash dist mash.sketch.msh {} \ | sort -nk3 \ | head -n 20 \ | sed -e
"s/contigs\\//g" -e "s/.fasta//g" > mash.sketch.msh.top20
```

- Step 2: For a assembly file, clean reads from the top 20 closest samples (including its own reads) by Mash distance were used to calculate the sequencing depth of contigs.

```
find contigs/*.fasta | parallel --colsep '\t' -j 20 bwa-mem2 index -p {} {}

cat mash.sketch.msh.top20 | parallel --colsep '\t' -j 10 mkdir -p depth/{2} \ ; bwa-mem2 mem -t 10
contigs/{2} clean_reads/{1}.1.fq.gz clean_reads/{1}.2.fq.gz \ | samtools view -bS - -@ 10 \ | samtools sort -@
10 -o depth/{2}/{1}.sort.bam \&\& jgi_summarize_bam_contig_depths --outputDepth
depth/{2}/{1}.sort.bam.depth depth/{2}/{1}.sort.bam
```

- Step 3: Integrate depth calculation files using the public script combine.pl, and perform multiple-coverage binning.

```
find depth/* -type d | parallel -j 10 combine.pl {}/*.sort.bam.depth \> {}.depth

find depth/*.depth | parallel -j 10 mkdir -p bins/{/}/ \ ; metabat2 -i contigs/{/}.fasta -a {} -o bins/{/}/.mbin
-m 2000 -s 200000 --saveCls --unbinned --seed 2020
```

- Step 4: Perform single-coverage binning.

```
find depth/* -type d | parallel -j 10 metabat2 -i contigs/{/}.fasta -a depth/{/}/.depth -o bins/{/}/.sbin -
m 2000 -s 200000 --saveCls --unbinned --seed 2020
```

Supplementary figure 12. Workflow of the single-coverage and Mash-based multi-coverage binning approach used in this study.

Published January 4, 2024 | Version v1

Dataset 
VMGC - Human Vaginal Microbiome Genome Collection

Guo, ruochun 
The VMGC is a large-scale reference genome resource of the human vaginal microbiome, including over 33,000 genomes derived from 786 prokaryotes, 38 fungi, and 4,263 viruses associated with the human vagina. In terms of representation, the VMGC demonstrates high efficiency in capturing microbial sequences, with a median mapping rate of 91.7% across 4,472 vaginal metagenomic samples obtained from 14 countries.

Files

Files (11.7 GB)		
Name	Size	 Download all
VMGC_eukaryocyte.info md5:a4b0da3eee94de403f738cfe01652adb 	10.9 kB	 Download
VMGC_eukaryocyte.tar.gz md5:c88710bc4922ee2e1d0f4e51c27cefaa 	201.2 MB	 Download
VMGC_prokaryote_MAG.info md5:0b47879fb512968c0819ee7d0cb5347a 	2.4 MB	 Download
VMGC_prokaryote_MAG.tar.gz md5:657a32cebc47b383c8f5d97bcc7b76d5 	8.7 GB	 Download
VMGC_prokaryote_SGB.info md5:9cb8726f73aba745d9884b2d1b7750a0 	122.5 kB	 Download
VMGC_prokaryote_SGB.tar.gz md5:923533477f20a4d8390a3209103afa80 	460.8 MB	 Download
VMGC_prokaryote_SGB_KrakenDB.tar.gz md5:d902bfbdf1fb2d139e88c92b0714cefe 	2.2 GB	 Download
VMGC_virus.info md5:6a711cabcb72d80e54342829c4b43ada 	992.0 kB	 Download
VMGC_virus.tar.gz md5:1600dbe008aaa1d92436ce9e959b5d40 	206.3 MB	 Download

Supplementary figure 13. Screenshot of <https://zenodo.org/records/10457006> (data files of VMGC).

4516. The introduction has a lot of information that is not really relevant to the work, e.g. the first paragraph.

Response: Thank you very much for this valuable comment. In accordance with your suggestions, we have made modifications to the revised manuscript and eliminated some unnecessary content. The first part of the new introduction is as follows:

“The human vagina is a diverse ecosystem hosting bacteria¹, viruses², fungi³, and other micro-eukaryotes⁴. In healthy women, Lactobacillus species, including L. crispatus, L. jensenii, and L. gasseri, dominate the vaginal microbiota^{5,6}, producing lactic acid to maintain a crucial acidic environment (pH below 4.5)⁷. This acidity prevents the proliferation of harmful microorganisms, ensuring a balanced vaginal ecosystem⁸. Dysbiosis, marked by decreased lactobacilli and increased pathogenic microorganisms^{9,10}, can lead to various health issues, including bacterial vaginosis (BV) characterized by anaerobic bacteria (e.g., Bifidobacterium vaginalis, Fannyhessea vaginae, and Prevotella spp.) overgrowth^{11,12}. Conditions like vulvovaginal candidiasis may result from Candida overgrowth¹³, while human papillomavirus (HPV) is linked to cervical cancer¹⁴, and herpes simplex virus (HSV) and human immunodeficiency virus (HIV) may lead to sexually transmitted infections¹⁵. Emerging research suggests alterations in the vaginal microbiota could indirectly impact reproductive health outcomes¹⁶, such as infertility and preterm birth^{17,18}. In summary, the composition and balance of the vaginal microbiota are crucial for women's health.”

17. The authors show that viruses do not fully saturate in this dataset. How does it look like for bacteria and fungi? Are 658 vaginal species across all of these samples a reasonable number? Also, why are there so few eukaryotic viruses?

Response: We sincerely appreciate your insightful comments. Based on your suggestion, we conducted a rarefaction analysis for the bacterial species in VMGC. Similar to viruses, we observed that, under the current genome count, the bacterial species in VMGC remains unsaturated (see supporting figure 14a below). However, additional species are likely to be rarer members of the human vaginal microbiome, as the number of species approaches saturation when considering those with at least two conspecific genomes. Furthermore, we examined the proportion of bacterial species relative to total abundance at different genome counts. This analysis revealed that, with 10,000 genomes, approximately 590 species cover the majority (>93%) of the relative abundance (see supporting figure 14b below). This result indicates that our collection satisfactorily covers highly abundant vaginal species.

Regarding fungi, due to the limited number of fungal genomes in the vagina, the species may still be unsaturated, requiring further investigation in future studies.

Regarding viruses, in the revised manuscript, we conducted a rarefaction analysis for all viruses. Similarly, although VMGC's coverage of all viruses has not reached saturation, it is approaching an asymptote for non-singleton viruses (see supporting figure 14c below). For eukaryotic viruses, given that our study focuses on metagenomic samples from the reproductive tract, the viruses primarily comprise bacteriophages potentially infecting bacteria. On the other hand, as human-infecting eukaryotic viruses are more likely to be RNA viruses, and our work is based on DNA sequencing of metagenomic samples, the analysis of RNA viruses is not extensively covered.

18. The section “Functional configuration of vaginal prokaryotic species” suffers from extreme overinterpretations. The authors show some gene annotations. They do not show that any microbe is a “producer”, “reservoir”, “synthesizer” or “play a role” in anything.

Response: Thanks for this insightful comment. Based on your suggestion, in the revised manuscript, we have toned down the language in this entire section to avoid overstatement and enhance accuracy. For instance, the expression "producer" has been revised to "encoded the highest gene abundances involving the synthesis of...", "contributed to gene abundance in...", or "identified as a potential primary producer of...". Please refer to the section "Functional configuration of vaginal prokaryotic species" in the revised manuscript for the modified content.

19. The focus of the 9 modules (P5, last paragraph) is not justified. Supp table 7 lists some references, some from the gut.

Response: Thanks for pointing this out. The products of these nine modules include some metabolites widely reported to be associated with bacterial vaginosis [1-2]. As for Supplementary Table 7, it lists some references from the gut. Most of the bacteria present in the vagina also appear in the gut. Referring to studies on gut bacteria, the selection of key genes (KO functional orthologs) in these functional modules seems reasonable to us.

References:

[1] Onderdonk A B, Delaney M L, Fichorova R N. The human microbiome during bacterial vaginosis. *Clinical microbiology reviews*, 2016, 29(2): 223-238.

[2] Zhu B, Tao Z, Edupuganti L, et al. Roles of the microbiota of the female reproductive tract in gynecological and reproductive health. *Microbiology and molecular biology reviews*, 2022, 86(4): e00181-21.

20. P13, "Identification of BVAB2 genomes" It's great that 16 of 18 are classified as BVAB2. How many genomes outside this cluster were classified as BVAB2?

Response: Thanks for pointing this out. We sincerely apologize for the previous inaccurate description. What we intended to convey is that all 18 genomes are classified as BVAB2. Among these, 16 genomes have complete 16S rDNA sequences, while the remaining 2 genomes only have partial 16S rDNA sequences. It's worth noting that the section "Identification of BVAB2 genomes" has already been

48removed in the revised manuscript, as the genomes of BVAB2 have been recently sequenced and publicly disclosed in a recent publication [1].

References:

[1] Srinivasan S, Austin M N, Fiedler T L, et al. *Amygdalobacter indicium* gen. nov., sp. nov., and *Amygdalobacter nucleatus* sp. nov., gen. nov.: novel bacteria from the family *Oscillospiraceae* isolated from the female genital tract. *International journal of systematic and evolutionary microbiology*, 2023, 73(10): 006017.

21. Page 5, L14-15 – this findings suggest novelty of species, but say nothing about importance.

Response: Thanks for pointing this out. Based on your suggestion, we have modified this sentence in the revised manuscript as follows: “*These findings suggested the considerable **novelty** of the uncultured species in the tree of life in the human vagina.*”

22. Figure 4d – what is the meaning of the two colors in the inner ring of the genome representation

Response: Thanks for pointing this out. In the original figure, the innermost ring displays forward-strand (red) and reverse-strand (green) protein-coding genes.

23. P3, L39 – how can *T. vaginalis*, with completeness of 38.4-47.8%, and *T parva* with 74%, be considered “highly credible”?

Response: Thanks for pointing this out. We have removed the expression “highly credible” in the revised manuscript as suggested.

24. It is not clear what is "prevalence rate" in Figure 5.

Response: In the Methods section, we have supplemented the calculation method for the prevalence rate of fungi as follows: *"To be considered present in a specific vaginal sample, a fungal species was required to capture at least three clean reads. The prevalence rate of a fungal species is then calculated based on this presence, determined by the number of vaginal samples with reads mapped against the species divided by the total number of samples."*

Decision Letter, first revision:

Message: 12th February 2024

Dear Professor Sun,

Thank you for your patience while your manuscript "A multi-kingdom collection of reference genomes in the human vaginal microbiome" was under peer-review at Nature Microbiology. It has now been seen by 3 referees, whose expertise and comments you will find at the of this email. You will see from their comments below that while they find your work of interest, some important points are raised. We are very interested in the possibility of publishing your study in Nature Microbiology, but would like to consider your response to these concerns in the form of a revised manuscript before we make a final decision on publication.

In particular, you will see that referees #1 and #3 have a few remaining concerns that we will need you to address and include additional information or analyses in the paper. The rest referees' reports are clear and the remaining issues should be straightforward to address.

If you have not done so already please begin to revise your manuscript so that it conforms

50to our Article format instructions at <http://www.nature.com/nmicrobiol/info/final-submission/>

The usual length limit for a Nature Microbiology Article is six display items (figures or tables) and 3,000 words. We have some flexibility, and can allow a revised manuscript at 3,500 words, but please consider this a firm upper limit. There is a trade-off of ~250 words per display item, so if you need more space, you could move a Figure or Table to Supplementary Information.

Some reduction could be achieved by focusing any introductory material and moving it to the start of your opening 'bold' paragraph, whose function is to outline the background to your work, describe in a sentence your new observations, and explain your main conclusions. The discussion should also be limited. Methods should be described in a separate section following the discussion, we do not place a word limit on Methods.

Nature Microbiology titles should give a sense of the main new findings of a manuscript, and should not contain punctuation. Please keep in mind that we strongly discourage active verbs in titles, and that they should ideally fit within 90 characters each (including spaces).

Please include a data availability statement as a separate section after Methods but before references, under the heading "Data Availability". This section should inform readers about the availability of the data used to support the conclusions of your study. This information includes accession codes to public repositories (data banks for protein, DNA or RNA sequences, microarray, proteomics data etc...), references to source data published alongside the paper, unique identifiers such as URLs to data repository entries, or data set DOIs, and any other statement about data availability. At a minimum, you should include the following statement: "The data that support the findings of this study are available from the corresponding author upon request", mentioning any restrictions on availability. If DOIs are provided, we also strongly encourage including these in the Reference list (authors, title, publisher (repository name), identifier, year). For more guidance on how to write this section please see: <http://www.nature.com/authors/policies/data/data-availability-statements-data-citations.pdf>

To improve the accessibility of your paper to readers from other research areas, please pay

particular attention to the wording of the paper's opening bold paragraph, which serves both as an introduction and as a brief, non-technical summary in about 150 words. If, however, you require one or two extra sentences to explain your work clearly, please include them even if the paragraph is over-length as a result. The opening paragraph should not contain references. Because scientists from other sub-disciplines will be interested in your results and their implications, it is important to explain essential but specialised terms concisely. We suggest you show your summary paragraph to colleagues in other fields to uncover any problematic concepts.

If your paper is accepted for publication, we will edit your display items electronically so they conform to our house style and will reproduce clearly in print. If necessary, we will re-size figures to fit single or double column width. If your figures contain several parts, the parts should form a neat rectangle when assembled. Choosing the right electronic format at this stage will speed up the processing of your paper and give the best possible results in print. We would like the figures to be supplied as vector files - EPS, PDF, AI or postscript (PS) file formats (not raster or bitmap files), preferably generated with vector-graphics software (Adobe Illustrator for example). Please try to ensure that all figures are non-flattened and fully editable. All images should be at least 300 dpi resolution (when figures are scaled to approximately the size that they are to be printed at) and in RGB colour format. Please do not submit Jpeg or flattened TIFF files. Please see also 'Guidelines for Electronic Submission of Figures' at the end of this letter for further detail.

Figure legends must provide a brief description of the figure and the symbols used, within 350 words, including definitions of any error bars employed in the figures.

When submitting the revised version of your manuscript, please pay close attention to our [href="https://www.nature.com/nature-research/editorial-policies/image-integrity">Digital Image Integrity Guidelines](https://www.nature.com/nature-research/editorial-policies/image-integrity). and to the following points below:

Please include a statement before the acknowledgements naming the author to whom correspondence and requests for materials should be addressed.

Finally, we require authors to include a statement of their individual contributions to the paper -- such as experimental work, project planning, data analysis, etc. -- immediately after the acknowledgements. The statement should be short, and refer to authors by their initials. For details please see the Authorship section of our joint Editorial policies at http://www.nature.com/authors/editorial_policies/authorship.html

- * include a point-by-point response to any editorial suggestions and to our referees. Please include your response to the editorial suggestions in your cover letter, and please upload your response to the referees as a separate document.

- * ensure it complies with our format requirements for Letters as set out in our guide to authors at www.nature.com/nmicrobiol/info/gta/

- * state in a cover note the length of the text, methods and legends; the number of references; number and estimated final size of figures and tables

- * resubmit electronically if possible using the link below to access your home page:

*This url links to your confidential homepage and associated information about manuscripts you may have submitted or be reviewing for us. If you wish to forward this e-mail to co-authors, please delete this link to your homepage first.

Please ensure that all correspondence is marked with your Nature Microbiology reference number in the subject line.

Nature Microbiology is committed to improving transparency in authorship. As part of our efforts in this direction, we are now requesting that all authors identified as 'corresponding author' on published papers create and link their Open Researcher and Contributor Identifier (ORCID) with their account on the Manuscript Tracking System (MTS), prior to acceptance. This applies to primary research papers only. ORCID helps the scientific community achieve unambiguous attribution of all scholarly contributions. You can create and link your ORCID from the home page of the MTS by clicking on 'Modify my Springer Nature account'. For more information please visit please visit www.springernature.com/orcid.

We hope to receive your revised paper within three weeks. If you cannot send it within this time, please let us know.

Reviewer Expertise:

Referee #1: microbiome, metagenomics, bioinformatics, culture

Referee #2: microbiome, metagenomics

Referee #3: vaginal microbiome, bioinformatics

Reviewers Comments:

Reviewer #1 (Remarks to the Author):

I thank the authors for addressing all my comments and I have no other major concerns. My only remaining comment would be for the authors to be cautious about the use of Kraken2 with default parameters. Given Kraken2 is based on exact k-mer matching of 31 nt and does not use any information of genome coverage, there is a risk of obtaining unspecific mappings to species that are actually not present in the sample. This can be alleviated by setting a threshold on the confidence score (e.g., 0.1). I suggest the authors at the very least compare how their taxonomic results differ using this criteria.

Reviewer #3 (Remarks to the Author):

I thank the authors for their thorough and insightful responses to reviewer comments. In particular, I found the additional analyses on exploring fungal ANI species boundaries and multi-sample coverage binning highly informative.

Reviewer #4 (Remarks to the Author):

This is an overall extensive revision that has addressed many of my methodological concerns. I do believe that the work is overall technically robust and uses up-to-date methods. In particular, the binning approach that was very problematic in the first version of the manuscript is now, to the best of my knowledge, state of the art and well motivated. The manuscript is still very descriptive, which is perhaps to be expected from a resource paper. Its biggest weakness, however, is that it lacks any type of validation or sanity tests that convince the reader that this work results in something that makes sense. Some of the puzzling figures (high overlap between phage and bacterial genomes, low concordance with VIRGO) are still there, and some of the validations I asked to perform did not make their way into the text itself.

With respect to my previous comments:

10. I'm not really satisfied with this answer. One can expect prophages not to bin with MAGs due to sequence composition and different coverage, so mapping to MAGs does not address my comment. A way to check this directly is to map the phages to all assembled contigs (not just the binned ones), and see if there's extension beyond the phage "edges". I am concerned that the high overlap of viral and prokaryotic genes (L469) is a result of misidentification of prophage edges.

11. This concern is not adequately addressed. Common kit contaminants ("kitomes") will generate the same pattern of genomes present in multiple samples. I find the authors' efforts for controlling contamination inadequate. The authors should at least clearly specify that their database could be contaminated.

12. Only one of the validations proposed was done. Even that validation, was not entirely satisfactory, and, as far as I can tell, did not make its way to the manuscript.

14. Same – as far as I can tell, was not included in the manuscript. Also, better to show this in log space.

15. The resource aspect of this work is much better, but documentation is still lacking for specific code files. E.g., what does "TableTreat.py" do?

16,19. Not addressed.

18. This is still not addressed. A few examples I picked up on: "Key contributor" (L243), "Enterobacterales synthesized" (L250), "important producers" (L253), "important contributors" (L255), "observed to play significant roles in producing" (L262), "was catalytically synthesized" (L315; a sentence that is misphrased for other reasons as well), "involvement...in the generation of these two substances (L501-503).

Additional comment:

I think it's wrong to say that the study by Pasoli et al. constructed the field (L98). If anything, the first such studies were from Jillian Banfield.

Author Rebuttal, first revision:

Reviewers Comments:

Reviewer #1 (Remarks to the Author):

55I thank the authors for addressing all my comments and I have no other major concerns. My only remaining comment would be for the authors to be cautious about the use of Kraken2 with default parameters. Given Kraken2 is based on exact k-mer matching of 31 nt and does not use any information of genome coverage, there is a risk of obtaining unspecific mappings to species that are actually not present in the sample. This can be alleviated by setting a threshold on the confidence score (e.g., 0.1). I suggest the authors at the very least compare how their taxonomic results differ using this criteria.

Response: We sincerely appreciate the reviewer’s encouragement and helpful comments. Based on your suggestion, we have modified the Kraken2 confidence score to 0.1 and updated all corresponding results in the revised manuscript. Additionally, we have compared the results generated with confidence scores of 0 (original parameter) and 0.1. This analysis revealed that, for all species, the average relative abundance under the two parameters is nearly identical (Pearson correlation coefficient = 0.998, $p < 0.001$; see supporting figure 1a below). Similarly, the species composition for all samples under the two parameters was also almost identical (see supporting figure 1b below).

Supporting figure 1. Comparison of species composition at two Kraken2 parameters. (a)

Comparison of average relative abundance of species at Kraken2 confidence score parameters 0 and 0.1.

(b) Comparison of species composition for all 4,429 vaginal metagenomes between the two parameters.

Reviewer #2 (Remarks to the Author):

I thank the authors for their thorough and insightful responses to reviewer comments. In particular, I found the additional analyses on exploring fungal ANI species boundaries and multi-sample coverage binning highly informative.

Response: We sincerely appreciate the encouragement and professional comments from the reviewer.

Reviewer #3 (Remarks to the Author):

This is an overall extensive revision that has addressed many of my methodological concerns. I do believe that the work is overall technically robust and uses up-to-date methods. In particular, the binning approach that was very problematic in the first version of the manuscript is now, to the best of my knowledge, state of the art and well motivated.

The manuscript is still very descriptive, which is perhaps to be expected from a resource paper. Its biggest weakness, however, is that it lacks any type of validation or sanity tests that convince the reader that this work results in something that makes sense. Some of the puzzling figures (high overlap between phage and bacterial genomes, low concordance with VIRGO) are still there, and some of the validations I asked to perform did not make their way into the text itself.

Response: We sincerely appreciate the reviewer's candid evaluation and professional comments on this study. Specifically, we are grateful for your recognition of our current multi-sample binning approach, which was modified based on your helpful suggestions provided in the previous round. In the revised manuscript, in accordance with your advice, we have added relevant validations and incorporated them into the manuscript, including the addition of five supplementary figures. Other sections of the

manuscript, including main texts, methods, and supplementary tables, have also undergone corresponding modifications. Please find below our detailed responses to each of your comments.

With respect to my previous comments:

10. I'm not really satisfied with this answer. One can expect prophages not to bin with MAGs due to sequence composition and different coverage, so mapping to MAGs does not address my comment. A way to check this directly is to map the phages to all assembled contigs (not just the binned ones), and see if there's extension beyond the phage "edges". I am concerned that the high overlap of viral and prokaryotic genes (L469) is a result of misidentification of prophage edges.

Previous comment:

10. How many phages are integrated? Are they even real phages? For how many of these were closed (circular) genomes assembled?

Response: Thank you very much for your insightful comments. Regarding the concern raised by the reviewer about the potential contamination of VMGC viruses with host genes, we utilized two methods during the viral identification process to avoid this issue. Firstly, we used CheckV to conduct a provirus assessment on the identified viral sequences. Based on the assessment results, CheckV extracted the "viral region" from each provirus; this process significantly reduced the risk of host gene contamination [1]. The total number of proviruses was 6,013, accounting for 42.3% of the total 14,224 viral sequences. Secondly, to further minimize contamination from host genes in VMGC viruses, we removed sequences with a high bacterial universal single-copy orthologs (BUSCO) ratio (>5%; methodology refers to [2]). This method eliminated approximately 0.8% of the total candidate viruses.

However, it is challenging to completely avoid "host gene contamination" when predicting virus sequences from metagenomic data. To assess the proportion of this contamination, we employed CheckV to evaluate "viral genes" and "microbial host genes", which were classified based on a database created using known viral and prokaryotic genes [1]. As demonstrated by CheckV's authors (see supporting figure 2a below), the host-to-virus gene ratio in viral sequences from the IMG/VR database was approximately 1:3, indicating a potential presence of substantial host contamination in this database. In contrast, our vaginal virus catalogue yielded a ratio of 1:16.0, significantly lower than several current large-scale virus catalogues such as Gut Virome Database (GVD) (host:virus = 1:3.3) and Gut Phage Database (GPD) (host:virus = 1:6.9) (see supporting figure 2b below). Notably, the host:virus ratio for proviruses in our catalogue was only 1:33.9, while other non-proviruses had a ratio of 1:10.5 (see supporting figure 2c below). These results indicate a low proportion of host contamination in our virus sequences.

58Subsequently, to assess whether the high overlap between viral and prokaryotic genes is due to "edge host gene contamination", we conducted two analyses:

1) We divided the 14,224 viruses in VMGC into complete viruses (n = 1,467; including 1,351 circular viruses containing direct terminal repeats [DTRs] and 116 linear viruses containing inverted terminal repeats [ITRs]) and incomplete viruses (n = 12,757). These complete viruses were considered to have no host genes. Using the same de-redundancy and clustering parameters, we observed that the complete viruses have a 39.3% gene overlap with prokaryotic genes, while the incomplete viruses have a 58.4% gene overlap (see supporting figure 3a below).

2) Then, we separately dereplicated the proviruses (n = 6,013) and non-proviruses (n = 6,744) in incomplete viruses and compared their genes with prokaryotic genes. This analysis showed that 80.6% of

genes from proviruses and 43.7% of genes from non-proviruses are covered by prokaryotes (see supporting figure 3b below). Furthermore, to assess the impact of "edge host gene contamination", we progressively truncated each virus sequence's first and last 1,2,3,...15 genes and evaluated the share of genes with prokaryotes. As shown in supporting figure 3c below, for both proviruses and non-proviruses, the shared gene proportion with prokaryotes only slightly decreased after truncation, indicating minimal "edge host gene contamination".

These results indicate that 1) the complete viruses have a 39.3% shared gene proportion with prokaryotic genes, suggesting a high baseline shared proportion. 2) The high proportion of gene sharing (58.4%) in incomplete viruses is primarily due to proviruses (80.6%), and the "edge host gene contamination" is almost non-existent. Considering the low proportion of "microbial host gene" in proviruses (see supporting figure 2c above), their high sharing proportion with prokaryotic genes may be attributed to the presence of a significant number of "viral genes" in prokaryotic genomes, rather than the opposite. Overall, the mentioned high overlap (53.7%) of viral genes by prokaryotic genes in our manuscript is valid. One possible reason is that our comparison involves viruses and prokaryotes from the same ecosystem and the samples, suggesting a higher frequent gene exchange than previously thought.

In line with the above evaluation and your suggestions, we have incorporated the results and descriptions of proviruses and host:virus ratio analysis (i.e., supporting figure 2b-c, presented as Supplementary Figure 2 in manuscript) and the comparison of different types of viruses with prokaryotic genes (i.e., supporting figure 3a-b, presented as Supplementary Figure 14 in manuscript) into the revised manuscript. Additionally, the corresponding methods have been elucidated in the revised manuscript.

Supporting figure 3. The overlap of prokaryotic and viral genes in the VMGC-90. (a) Venn plot showing the comparisons of prokaryotic genes and genes from completeness (left panel) and incomplete viruses (right panel). **(b)** Venn plot showing the comparisons of prokaryotic genes and genes from proviruses (left panel) and non-proviruses (right panel). Red numbers show the percentages of viral genes covered by prokaryotic genes for each comparison. **(c)** Proportion of viral genes covered by prokaryotic genes under the different numbers (1,2,3,...15) of genes trimmed from both ends of the viral sequences.

References:

[1] Nayfach S, Camargo A P, Schulz F, et al. CheckV assesses the quality and completeness of metagenome-assembled viral genomes. *Nature biotechnology*, 2021, 39(5): 578-585.

[2] Gregory A C, Zablocki O, Zayed A A, et al. The gut virome database reveals age-dependent patterns of virome diversity in the human gut. *Cell host & microbe*, 2020, 28(5): 724-740. e8.

11. This concern is not adequately addressed. Common kit contaminants ("kitomes") will generate the same pattern of genomes present in multiple samples. I find the authors' efforts for controlling contamination inadequate. The authors should at least clearly specify that their database could be contaminated.

Previous comment:

11. How many of these species actually come from the vaginal environments? Many of the bacterial species found (177) had been previously identified only in other environments (from other body sites mostly). Could these be contamination? What is their prevalence across the metagenomic samples? How is contamination controlled for?

Response: Thank you for providing this insightful comment. We agree with your perspective that a suitable definition of the "kitome" could effectively mitigate contamination. However, given the absence of negative controls in the majority of the studies we utilized, it is currently challenging to distinguish whether a specific bacterium originates from the investigated vaginal samples or from kit contaminants. On the other hand, considering the stochastic nature of contamination, it is improbable that a "contaminant species" would simultaneously appear across a large number of different samples, studies/projects, or kit types. Based on this, we conducted a thorough assessment of the vaginal species distribution across diverse subjects (spanning over 4,000 samples), studies (spanning 31 studies), and kit types (spanning 19 kit types for DNA extraction; shown in Supplementary Table 1 in the revised manuscript). The results revealed that over 95% of prokaryotic species are present in at least 31 diverse subjects, and similarly, over 95% of species are identified in samples from 7 distinct studies or in samples from 6 different kit types (see supporting figure 4 below). While not entirely precise, these results suggest that the majority of vaginal species in the VMGC are unlikely to result from random contamination.

Furthermore, in response to your suggestion, we have added a limitation in the Discussion of the revised manuscript, addressing the potential contamination in the public vaginal metagenome datasets used to construct the VMGC. The added content is as follows:

"Moreover, it is crucial to acknowledge the potential contamination (e.g., from experimental operations or kits) in the public vaginal metagenome datasets used for constructing the VMGC. Although

62our assessment indicated a low level of contamination (Supplementary Figure 16), caution is still warranted when utilizing the VMGC."

12. Only one of the validations proposed was done. Even that validation, was not entirely satisfactory, and, as far as I can tell, did not make its way to the manuscript.

Previous comment:

12. Extensive validations are necessary here. Some suggestions could be comparison to 16S; examination of CSTs; comparisons to isolates; comparisons to other studies. In that regard, the comparison to VIRGO (P9, L33-34) is VERY alarming. With the exact same samples, there is only 58% coverage of the VIRGO dataset? If the authors have a potential explanation to this, such as L35, they should prove that this is the case.

Response: We greatly appreciate the insightful comments from the reviewer.

Regarding the comparison to VIRGO, based on your suggestions, we have updated the figures and results in the revised manuscript using the new comparison between VMGC-90 and VIRGO-90. Specifically, as mentioned in our previous responses, we found that some specific genes in VIRGO may originate from non-vaginal bacterial isolates, and here we further identified a small number ($n = 15,575$, approximately 2.3% of all) of these VIRGO-specific genes are originated from the human genome, potentially due to incomplete host decontamination. After removing these non-relevant genes, the non-redundant gene count for VIRGO-90 is 620,553, with 438,155 (70.6% of all genes) covered by our VMGC-90. This updated result is reflected in the Results (section “*Vaginal microbial gene catalogue and comparison with VIRGO*”) and Figure 8b (also see supporting figure 5a below) of the revised manuscript. Additionally, for those VIRGO-specific genes not covered by VMGC, as mentioned previously, we found that a portion of genes originated from the unbinned sequences in the vaginal metagenomes (~44%, 80,141/182,398, usually from plasmids or other accessory genes) or other known vaginal species (~28%, 50,259/182,398) (see supporting figure 5a below). This information has been added to the revised manuscript as Supplementary Figure 12a for clarification.

Regarding other validations:

1) *Comparison to 16S*. As our vaginal microbial gene catalogue relies on protein-coding genes predicted by the Prodigal software, we did not initially compare it with 16S rRNA genes. To further explore the presence of 16S rRNA genes in VMGC, we downloaded two large-scale 16S rRNA gene catalogues [1,2] and compared their sequences with the VMGC prokaryotic genomes. This analysis revealed that only 52.7%-56.5% of the 16S rRNA gene sequences could be covered by VMGC genomes (see supporting figure 5b below), indicating that the coverage of VMGC for 16S rRNA genes is evidently insufficient and needs improvement. In practice, due to the high conservation and multiple copies of 16S rRNA genes, they are challenging to be assembled into long contigs and binned into MAGs, leading to only 16.5% (3,234/19,542, Supplementary Table 3) of the VMGC prokaryotic genomes having complete 16S rRNA genes. Therefore, despite the abundance of microbial genomic data, studying the 16S rRNA gene content in the vaginal microbiome remains crucial.

2) *Comparisons to isolates*. Indeed, VMGC includes genomes and gene sets from isolated genomes in the NCBI database. To further validate the comprehensiveness of VMGC, we compared it to a recently published *Lactobacillus* genomic catalogue (spanning 1,091 previously unreported isolate genomes,

partial genomes, and MAGs [3]). This analysis showed that 18,998 (96.3%) out of 19,726 non-redundant *Lactobacillus* genes were covered by VMGC (see supporting figure 5c below), highlighting the high comprehensiveness of VMGC genes.

3) *Comparison to other studies.* While VIRGO is the current most comprehensive publicly available vaginal microbial gene catalogue, we did not compare the VMGC to other studies in the manuscript. To address the reviewer's request, we downloaded vaginal metagenome-assembled genomes (MAGs) from a previously published study [4] to construct a microbial gene catalogue encompassing 71,588 non-redundant genes. A comparative analysis with VMGC-90 revealed that 97.5% (69,777/71,588) of genes in this catalogue were covered by VMGC-90 (see supporting figure 5d below), affirming the extensive coverage of VMGC over existing genes. Additionally, we have contacted the authors of a previous study on the vaginal microbiome of Chinese women and constructed a new vaginal microbial gene catalogue based on their metagenome dataset [5]. This catalogue includes a total of 412,787 non-redundant genes from 2,720 MAGs (assembled from 1,148 metagenomes). The comparison result showed that 373,906 (86.1%) genes in this catalogue could be covered by VMGC-90 (see supporting figure 5e below), once again emphasizing the high comprehensiveness of VMGC genes.

According to your suggestion, we have incorporated the results from supporting figures 5c-e into the revised manuscript as Supplementary Figure 12b. Additionally, a description of these results has been added to the Results (section “*Vaginal microbial gene catalogue and comparison with VIRGO*”) as follows: “*Comparison with other databases also revealed that VMGC-90 significantly expands the protein content of the vaginal microbiota (Supplementary Figure 12b).*”

References:

[1] Lebeer S, Ahannach S, Gehrman T, et al. A citizen-science-enabled catalogue of the vaginal microbiome and associated factors. *Nature Microbiology*, 2023, 8(11): 2183-2195.

[2] Liu Y, Li T, Guo R, et al. The vaginal microbiota among the different status of human papillomavirus infection and bacterial vaginosis. *Journal of Medical Virology*, 2023, 95(3): e28595.

[3] Bloom S M, Mafunda N A, Woolston B M, et al. Cysteine dependence of *Lactobacillus iners* is a potential therapeutic target for vaginal microbiota modulation. *Nature Microbiology*, 2022, 7(3): 434-450.

[4] Pasolli E, Asnicar F, Manara S, et al. Extensive unexplored human microbiome diversity revealed by over 150,000 genomes from metagenomes spanning age, geography, and lifestyle. *Cell*, 2019, 176(3): 649-662. e20.

[5] Jie Z, Chen C, Hao L, et al. Life history recorded in the vagino-cervical microbiome along with multi-omes. *Genomics, Proteomics & Bioinformatics*, 2022, 20(2): 304-321.

14. *Same – as far as I can tell, was not included in the manuscript. Also, better to show this in log space.*

Previous comment:

14. *Are the HPV viruses assembled enriched in patients with HPV in the various studies?*

Response: Thanks for pointing this out. We have incorporated the HPV results into the revised manuscript as Supplementary Figure 11, and the vertical axis has been log-transformed as suggested.

15. *The resource aspect of this work is much better, but documentation is still lacking for specific code files. E.g., what does “TableTreat.py” do?*

Response: Thanks for bringing this to our attention. We have further improved the relevant documentation. For specific details, please refer to the document at:

<https://github.com/RChGO/VMGC/tree/main/Pipelines/README.md> (see supporting figure 6 below for part screenshot).

The files in this folder contain custom scripts required for constructing VMGC. Example files related to their usage are stored at <https://github.com/RChGO/VMGC/blob/main/Documents/>. Refer to the following instructions for more details.

TableTreat.py

- The Python script was designed to match and merge two tables.

```
>TableTreat.py -h
```

```
optional arguments:
-h, --help            show this help message and exit
-a A                  input table1, required
-b B                  input table2, required
-o O                  input name of outfile, required
-m {table_column,table_row,table_merge}
                    pick out the needed rows or columns according to table2, default table_row
-r                    whether to output variables that aren't in table2, default closed
-w W                  merge type [left/right/inner/outer], default left
-x X                  column for table1, default 0
-y Y                  column for table2, default 0
-i I                  whether or not table1 has header, default None, e.g. 0,1,2
-j J                  whether or not table2 has header, default None, e.g. 0,1,2
-f F                  replace this of vacant position, default NA
```

```
>TableTreat.py -a sgb.info -b sgb.seq -x 0 -m table_merge -o Temp.list
```

blast_cvq.v1.py

- In long sequence blastn alignments, it's common for the alignments between two sequences to be fragmented. This Python script was developed to merge these fragmented alignment results, providing an overall assessment of similarity and alignment length between the two sequences.

```
>blast_cvq.v1.py -h
```

```
usage: blast_cvq.v1.py [-h] in_f out_f min_id min_len

positional arguments:
in_f      blast.m8
out_f     output
min_id    ignore the identity less than min_id[default 75]
min_len   ignore the identity less than min_len[default 200]

optional arguments:
-h, --help show this help message and exit
```

```
>blast_cvq.v1.py blast\_cvq.input blast\_cvq.output 0 0
```

blast_cluster.v2.pl

- This Perl script was developed to cluster sequences based on the output of blast_cvq.v1.py.

```
>blast_cluster.v2.pl -h
```

```
perl blast\_cluster.v2.pl [in.len.sort] [in.blast.cvg] [out.fasta] [%cvg] [%iden]
```

```
>blast_cluster.v2.pl blast\_cluster.len.sort blast\_cvq.output blast\_cluster.output 85 95
```

Supporting figure 6. Part screenshot of the documentations of the VMGC GitHub website.

16,19. Not addressed.

Previous comment:

16. The introduction has a lot of information that is not really relevant to the work, e.g. the first paragraph.

19. The focus of the 9 modules (P5, last paragraph) is not justified. Supp table 7 lists some references, some from the gut.

Response: Thank you very much for your valuable comments. Here are our responses to the two comments:

Comment #16: In line with your suggestion, we have removed irrelevant content, such as the associations between fungi/viruses and vaginal diseases, from the Introduction of the revised manuscript. The first paragraph of the Introduction has been merged into the second paragraph and condensed overall to better align with the content of our work.

Comment #19: The selection of these 9 modules was based on an extensive literature review and our own summary. They all fall under widely reported bacterial vaginosis-associated virulence factors, roughly grouped into 4 categories:

Biofilm formation	Protecting harmful bacteria
Sialidase	Disruption of the vaginal mucosal barrier
Toxins (cytolysin, hemolysin) and enzymes (urease, phospholipase C)	Disruption of the epithelial barrier and induction of inflammation
Biogenic amines (cadaverine, N-acetylputrescine, and trimethylamine)	Producing unpleasant odor, elevating pH to promote the growth of harmful bacteria

For a detailed interpretation, please refer to supporting table 1 below. We have incorporated the descriptions of these 9 functional modules into Supplementary Table 7 in the revised manuscript. Furthermore, following your suggestions and Comment #18, we have rewritten the content pertaining to the 9 functional modules. The revised descriptions are organized in the sequence of the 4 aforementioned categories, enhancing the overall structure and clarity. Additionally, brief descriptions of these functionalities have been integrated into the Results to improve the readability and coherence of the study.

Supporting figure 1. Detailed descriptions of 9 functional modules.

Functional module	Description
Biofilm formation	Biofilm formation is key for the development of disease since it confers heightened antibiotic tolerance and resistance to host immune defenses, making diseases chronic and/or relapsing [1-3]. Vaginal biofilms protect bacterial vaginosis-associated taxa from clearance by lower pH and lactic acid, antibacterials/antibiotics, and the host immune system [4-5].
Sialidase	Sialidase is an enzyme that enhance the ability of microorganisms to invade and destroy tissue, which is capable of damaging host tissues, disrupting the vaginal mucosal barrier, and impairing the specific immunoglobulin A immune response against other virulence factors [6-8]. Vaginal fluid sialidase is widely highly correlated with bacterial vaginosis [9-10], and has a strong association with adverse pregnancy outcomes [11].
Cytolysin / Inerolysin / Vaginolysin production	Cholesterol-dependent cytolysins (e.g., inerolysin and vaginolysin) belong to a family of pore-forming toxins produced by vaginal pathogenic bacteria and participate in various stages of disease, promoting bacterial invasion and infection [12-13]. In the vagina, cytolysins and sialidases are often considered to work synergistically to damage the vaginal epithelium [14-16]. Sialidases have the potential to degrade the host vaginal mucosal layer's glycoprotein chains, disrupting the mucosal barrier of epithelial cells. Removing the mucin layer allows cytolysins better access to epithelial cells. Additionally, cytolysins produced by Gardnerella vaginalis (mainly vaginolysin) can cause cell death by activating the protein kinase pathway in vaginal epithelial cells [17].
Hemolysin production	Hemolysin induces inflammation and the disruption of vaginal epithelial barriers, leading to a high risk of bacterial vaginosis [18].
Urease	Urease hydrolyzes urea into carbon dioxide and ammonia, raising the pH of the vaginal environment and facilitating mixed infection with other bacteria [19]. Additionally, urease can also damage the vaginal epithelial layer and induce inflammation [20].

Phospholipase C	Phospholipase C (lecithinase or phosphatidylcholine phosphorylase) catalyzes the hydrolysis of lecithin into phosphorylcholine and 1,2-diglyceride. Bacterial production of phospholipase C can induce loss of structural integrity of epithelial cells, which may damage reproductive tract tissues by both direct and indirect mechanisms [21-24].
Cadaverine production	The generation of biogenic amines is a mechanism of acid resistance. Biogenic amines are associated with increased vaginal pH, abnormal vaginal odor or discharge, or activation of proinflammatory responses [25-26]. Biogenic amines can reduce the barrier integrity of the epithelia and consequently facilitate the outgrowth of bacterial vaginosis-associated vaginal taxa [27-28].
N-Acetylputrescine production	
Trimethylamine biosynthesis	

References:

- [1] Swidsinski A, Mendling W, Loening-Baucke V, et al. An adherent *Gardnerella vaginalis* biofilm persists on the vaginal epithelium after standard therapy with oral metronidazole. *American journal of obstetrics and gynecology*, 2008, 198(1): 97. e1-97. e6.
- [2] Patterson J L, Stull-Lane A, Girerd P H, et al. Analysis of adherence, biofilm formation and cytotoxicity suggests a greater virulence potential of *Gardnerella vaginalis* relative to other bacterial-vaginosis-associated anaerobes. *Microbiology*, 2010, 156(Pt 2): 392.
- [3] Danielsson D, Teigen P K, Moi H. The genital econiche: focus on microbiota and bacterial vaginosis. *Annals of the New York Academy of Sciences*, 2011, 1230(1): 48-58.
- [4] Onderdonk A B, Delaney M L, Fichorova R N. The human microbiome during bacterial vaginosis. *Clinical microbiology reviews*, 2016, 29(2): 223-238.
- [5] Jung H S, Ehlers M M, Lombaard H, et al. Etiology of bacterial vaginosis and polymicrobial biofilm formation. *Critical reviews in microbiology*, 2017, 43(6): 651-667.
- [6] Zhu B, Tao Z, Edupuganti L, et al. Roles of the microbiota of the female reproductive tract in gynecological and reproductive health. *Microbiology and Molecular Biology Reviews*, 2022, 86(4): e00181-21.

- [7] Lewis W G, Robinson L S, Perry J, et al. Hydrolysis of secreted sialoglycoprotein immunoglobulin A (IgA) in ex vivo and biochemical models of bacterial vaginosis. *Journal of Biological Chemistry*, 2012, 287(3): 2079-2089.
- [8] Cauci S, Driussi S, Monte R, et al. Immunoglobulin A response against *Gardnerella vaginalis* hemolysin and sialidase activity in bacterial vaginosis. *American journal of obstetrics and gynecology*, 1998, 178(3): 511-515.
- [9] Briselden A M, Moncla B J, Stevens C E, et al. Sialidases (neuraminidases) in bacterial vaginosis and bacterial vaginosis-associated microflora. *Journal of clinical microbiology*, 1992, 30(3): 663-666.
- [10] McGregor J A, French J I, Jones W, et al. Bacterial vaginosis is associated with prematurity and vaginal fluid mucinase and sialidase: results of a controlled trial of topical clindamycin cream. *American journal of obstetrics and gynecology*, 1994, 170(4): 1048-1060.
- [11] Cauci S, McGregor J, Thorsen P, et al. Combination of vaginal pH with vaginal sialidase and prolidase activities for prediction of low birth weight and preterm birth. *American journal of obstetrics and gynecology*, 2005, 192(2): 489-496.
- [12] Christie M P, Johnstone B A, Tweten R K, et al. Cholesterol-dependent cytolysins: from water-soluble state to membrane pore. *Biophysical reviews*, 2018, 10: 1337-1348.
- [13] Los F C O, Randis T M, Aroian R V, et al. Role of pore-forming toxins in bacterial infectious diseases. *Microbiology and Molecular Biology Reviews*, 2013, 77(2): 173-207.
- [14] Garcia E M, Kraskauskiene V, Koblinski J E, et al. Interaction of *Gardnerella vaginalis* and vaginolysin with the apical versus basolateral face of a three-dimensional model of vaginal epithelium. *Infection and immunity*, 2019, 87(4): 10.1128/iai.00646-18.
- [15] Ragaliauskas T, Plečkaitytė M, Jankunec M, et al. Inerolysin and vaginolysin, the cytolysins implicated in vaginal dysbiosis, differently impair molecular integrity of phospholipid membranes. *Scientific reports*, 2019, 9(1): 10606.
- [16] Plečkaityte M. Cholesterol-dependent cytolysins produced by vaginal bacteria: certainties and controversies. *Frontiers in cellular and infection microbiology*, 2020, 9: 452.
- [17] Gelber S E, Aguilar J L, Lewis K L T, et al. Functional and phylogenetic characterization of Vaginolysin, the human-specific cytolysin from *Gardnerella vaginalis*. *Journal of bacteriology*, 2008, 190(11): 3896-3903.

- [18] Cauci S, Scrimin F, Driussi S, et al. Specific immune response against Gardnerella vaginalis hemolysin in patients with bacterial vaginosis. *American journal of obstetrics and gynecology*, 1996, 175(6): 1601-1605.
- [19] Harada K, Tanaka H, Komori S, et al. Vaginal infection with Ureaplasma urealyticum accounts for preterm delivery via induction of inflammatory responses. *Microbiology and immunology*, 2008, 52(6): 297-304.
- [20] Amabebe E, Richardson L S, Bento G F C, et al. Ureaplasma parvum infection induces inflammatory changes in vaginal epithelial cells independent of sialidase. *Molecular Biology Reports*, 2023, 50(4): 3035-3043.
- [21] McGregor J A, Lawellin D, Franco-Buff A, et al. Phospholipase C activity in microorganisms associated with reproductive tract infection. *American journal of obstetrics and gynecology*, 1991, 164(2): 682-686.
- [22] De Silva N S, Quinn P A. Characterization of phospholipase A1, A2, C activity in Ureaplasma urealyticum membranes. *Molecular and cellular biochemistry*, 1999, 201: 159-167.
- [23] Udayalaxmi J, Bhat G K, Kotigadde S. Biotypes and virulence factors of Gardnerella vaginalis isolated from cases of bacterial vaginosis. *Indian Journal of Medical Microbiology*, 2011, 29(2): 165-168.
- [24] Mohammadzadeh R, Kalani B S, Kashanian M, et al. Prevalence of vaginolysin, sialidase and phospholipase genes in Gardnerella vaginalis isolates between bacterial vaginosis and healthy individuals. *Medical Journal of the Islamic Republic of Iran*, 2019, 33: 85.
- [25] Nelson T M, Borgogna J L C, Brotman R M, et al. Vaginal biogenic amines: biomarkers of bacterial vaginosis or precursors to vaginal dysbiosis? *Frontiers in physiology*, 2015, 6: 253.
- [26] Srinivasan S, Morgan M T, Fiedler T L, et al. Metabolic signatures of bacterial vaginosis. *MBio*, 2015, 6(2): 10.1128/mbio.00204-15.
- [27] Borgogna J L C, Shardell M D, Grace S G, et al. Biogenic amines increase the odds of bacterial vaginosis and affect the growth of and lactic acid production by vaginal Lactobacillus spp. *Applied and environmental microbiology*, 2021, 87(10): e03068-20.
- [28] Wolrath H, Borén H, Hallén A, et al. Trimethylamine content in vaginal secretion and its relation to bacterial vaginosis. *Apmis*, 2002, 110(11): 819-824.

18. This is still not addressed. A few examples I picked up on: “Key contributor” (L243), “Enterobacterales synthesized” (L250), “important producers” (L253), “important contributors” (L255), “observed to play significant roles in producing” (L262), “was catalytically synthesized” (L315; a sentence that is misphrased for other reasons as well), “involvement...in the generation of these two substances (L501-503).

Previous comment:

18. The section “Functional configuration of vaginal prokaryotic species” suffers from extreme overinterpretations. The authors show some gene annotations. They do not show that any microbe is a “producer”, “reservoir”, “synthesizer” or “play a role” in anything.

Response: Thank you sincerely for your thoughtful and meticulous comment. In accordance with your suggestion and Comment #19 (as mentioned above), we have rewritten this section and other related content in the revised manuscript. The new content thoroughly avoids the potential overstatements mentioned in your comment as follows:

“We found that *Lactobacillales* (mainly *Lactobacillus crispatus* and *Lactobacillus iners* at the species level) and *Enterobacterales* (primarily *Escherichia coli*) significantly contributed to gene abundance in biofilm formation synthesis, followed by several potential pathogens such as *Pseudomonas aeruginosa* and *BVAB1* (Figure 3b; Supplementary Figure 5a). Genes involved in the biosynthesis of sialidase, an enzyme capable of disrupting the vaginal mucosal barrier^{29,30}, were predominantly encoded by *Actinomycetales* (mainly *Bifidobacterium piovii* and *Bifidobacterium vaginale*) (Supplementary Figure 5b). *Lactobacillales* (mainly *Lactobacillus iners*) and *Bifidobacterium spp.* exhibited the highest gene abundances related to the synthesis of cytolytins (e.g., *inerolysin* and *vaginolysin*) and hemolysin (Supplementary Figure 5c); these toxins pose a high risk for inflammation and the disruption of vaginal epithelial barrier^{31,32}. *Mycobacteriales* and *Enterobacterales* displayed the highest gene abundance in the synthesis of phospholipase C and urease, while *Ureaplasma parvum* and *Pseudomonas spp.* also encoded a high gene abundance for urease. Several biogenic amines, including cadaverine, *N*-acetylputrescine, and trimethylamine, are major amine components of vaginal fluid and are implicated in vaginitis and the activation of proinflammatory responses³³⁻³⁵. For these, *Enterobacterales* (mainly *Escherichia coli*) and *Saccharofermentanales* (mainly *Amygdalobacter indicium* and *Mageeibacillus indolicus*) encoded the major gene abundance responsible for the production of cadaverine and trimethylamine, while various beneficial (e.g., *Lactobacillus gasseri* and *Lactobacillus johnsonii*) and harmful species (e.g., *Escherichia coli* and *Prevotella bivia*) harbored genes associated with *N*-acetylputrescine production (Supplementary Figure 5d).”

Additional comment:

I think it's wrong to say that the study by Pasoli et al. constructed the field (L98). If anything, the first such studies were from Jillian Banfield.

Response: Thanks for pointing this out. Following your suggestion, the sentence was modified into “*a recent study by Pasoli et al. has performed a comprehensive collection of over 150,000 MAGs...*” in the revised manuscript to avoid misleading.

Decision Letter, second revision:

Message: Our ref: NMICROBIOL-23071678B

22nd March 2024

Dear Dr. Sun,

Thank you for submitting your revised manuscript "A multi-kingdom collection of reference genomes in the human vaginal microbiome" (NMICROBIOL-23071678B). It has now been seen by the original referees and their comments are below. The reviewers find that the paper has improved in revision, and therefore we'll be happy in principle to publish it in Nature Microbiology, pending minor revisions to satisfy the referees' final requests and to comply with our editorial and formatting guidelines.

Thank you again for your interest in Nature Microbiology Please do not hesitate to contact me if you have any questions.

Sincerely,

75Reviewer #1 (Remarks to the Author):

Thank you for addressing the remaining comments. The authors should be commended for such thorough revisions throughout the different review stages. I have no further concerns.

Reviewer #4 (Remarks to the Author):

I thank the authors for addressing my comments. I have no further comments and I commend the authors for a very comprehensive and well-done analysis.

Final Decision Letter:

Message: 1st May 2024

Dear Professor Sun,

I am pleased to accept your Resource "A multi-kingdom collection of 33,804 reference genomes for the human vaginal microbiome" for publication in Nature Microbiology. Thank you for having chosen to submit your work to us and many congratulations.

After the grant of rights is completed, you will receive a link to your electronic proof via

76email with a request to make any corrections within 48 hours. If, when you receive your proof, you cannot meet this deadline, please inform us at rjsproduction@springernature.com immediately. You will not receive your proofs until the publishing agreement has been received through our system

Please note that *Nature Microbiology* is a Transformative Journal (TJ). Authors may publish their research with us through the traditional subscription access route or make their paper immediately open access through payment of an article-processing charge (APC). Authors will not be required to make a final decision about access to their article until it has been accepted. Find out more about Transformative Journals

We welcome the submission of potential cover material (including a short caption of around 40 words) related to your manuscript; suggestions should be sent to Nature Microbiology as electronic files (the image should be 300 dpi at 210 x 297 mm in either TIFF or JPEG format). Please note that such pictures should be selected more for their aesthetic appeal than for their scientific content, and that colour images work better than black and white or grayscale images. Please do not try to design a cover with the Nature Microbiology logo

etc., and please do not submit composites of images related to your work. I am sure you will understand that we cannot make any promise as to whether any of your suggestions might be selected for the cover of the journal.
